# Global Minimizers of $\ell^p$-Regularized Objectives Yield the Sparsest ReLU Neural Networks

**Julia Nakhleh**
Department of Computer Science
University of Wisconsin-Madison
Madison, WI
jnakhleh@wisc.edu

**Robert D. Nowak**
Department of Electrical and Computer Engineering
University of Wisconsin-Madison
Madison, WI
rdnowak@wisc.edu

## Abstract

Overparameterized neural networks can interpolate a given dataset in many different ways, prompting the fundamental question: which among these solutions should we prefer, and what explicit regularization strategies will provably yield these solutions? This paper addresses the challenge of finding the sparsest interpolating ReLU network—i.e., the network with the fewest nonzero parameters or neurons—a goal with wide-ranging implications for efficiency, generalization, interpretability, theory, and model compression. Unlike post hoc pruning approaches, we propose a continuous, almost-everywhere differentiable training objective whose global minima are guaranteed to correspond to the sparsest single-hidden-layer ReLU networks that fit the data. This result marks a conceptual advance: it recasts the combinatorial problem of sparse interpolation as a smooth optimization task, potentially enabling the use of gradient-based training methods. Our objective is based on minimizing $\ell^p$ quasinorms of the weights for $0 < p < 1$, a classical sparsity-promoting strategy in finite-dimensional settings. However, applying these ideas to neural networks presents new challenges: the function class is infinite-dimensional, and the weights are learned using a highly nonconvex objective. We prove that, under our formulation, global minimizers correspond exactly to sparsest solutions. Our work lays a foundation for understanding when and how continuous sparsity-inducing objectives can be leveraged to recover sparse networks through training.

## 1 Introduction

Highly overparameterized neural networks have become the workhorse of modern machine learning. Because these networks can interpolate a given dataset in many different ways (see e.g. Figs. 1b and 1c), explicit regularization is frequently incorporated into the training procedure to favor solutions that are, in some sense, more regular or desirable. In this work, we focus on explicit regularizers which yield *sparse* single-hidden-layer ReLU interpolating networks, which for our purposes are those with the fewest nonzero input weight/bias parameters among the active neurons.[1] Sparse models are particularly desirable for computational efficiency purposes, as they have lower storage requirements and computational overhead when deployed at inference time, and may have other attractive properties in terms of generalization, interpretability, and robustness (Mozer and Smolensky (1988); Guo et al. (2018); Liao et al. (2022); Liu et al. (2022), among many others).

Although a myriad of sparsity-inducing training schemes have been proposed in the neural network literature, almost none of them have actually been proven to yield true *sparsest* solutions, and the

---

[1]In the univariate-input case, this is equivalent to the count of active neurons.

39th Conference on Neural Information Processing Systems (NeurIPS 2025).

justifications for their use remain almost entirely heuristic and/or empirical. Furthermore, many such strategies rely on complex pruning pipelines—composed of iterative magnitude thresholding, fine-tuning, and sensitivity analyses—which are computationally costly, difficult to implement, and offer no theoretical guarantees in terms of the resulting sparsity. In contrast, we propose a simple regularization objective, based on the $\ell^p$ quasinorm of the network weights for $0 < p < 1$, whose global minimizer is *provably* a sparsest interpolating ReLU network for sufficiently small $p$. This objective is continuous and differentiable away from zero, making it compatible with gradient descent. Although $\ell^p$-norm minimization with $0 < p < 1$ has been studied in finite-dimensional linear problems (most extensively in the context of compressed sensing), where it is known to guarantee sparsity under certain assumptions on the data/measurements, its behavior in the context of neural networks—wherein the features themselves are continuously parameterized and data-adaptive—is challenging to characterize mathematically, and to our knowledge, we are the first to do so. Specifically, our contributions are the following:

1. **Sparsity, uniqueness, and width/parameter bounds for univariate $\ell^p$-regularized networks.** In Section 3, we prove that, for single-hidden-layer ReLU networks of input dimension one, minimizing the network's $\ell^p$ *path norm* (see (2)) implicitly minimizes both its $\ell^1$ path norm (i.e., the total variation of its derivative) and, for sufficiently small $p > 0$, its $\ell^0$ path norm (total knot/neuron count). We show that for *any* $0 < p < 1$, a minimum $\ell^p$ path norm interpolant of $N$ data points has no more than $N - 2$ active neurons. In contrast, $\ell^1$ path norm minimization alone is *not* guaranteed to implicitly minimize sparsity, and may yield solutions with arbitrarily many neurons (Fig. 1a). Our result follows from reframing the network training problem as an optimization over continuous piecewise linear (CPWL) functions which interpolate a dataset with minimal $p$-variation (6) of the derivative. Using this variational framework, we can explicitly describe the optimal functions' behavior based on the geometry of the data points. This characterization provides data-dependent bounds on the sparsity and weight magnitudes of such minimum-$\ell^p$ solutions, and highlights an easily-verifiable condition on the data under which $\ell^p$ minimization for *any* $0 < p < 1$ yields a sparsest interpolant ($\ell^0$ solution). Additionally, our analysis shows that the solution to the univariate $\ell^p$ minimization problem is *unique* for almost every $0 < p < 1$; in contrast, univariate $\ell^0$ and $\ell^1$ solutions are both known to be non-unique in general (Debarre et al. (2022); Hanin (2022)).

2. **Exact sparsity in arbitrary input dimensions.** In Section 4, we show for networks of arbitrary input dimension that the problem of minimizing the network's $\ell^p$ path norm can be recast as a finite-dimensional minimization of a continuous, strictly concave function over a polytope.[2] Using this reformulation, we show that there always exists some data-dependent threshold $p^*$ below which $\ell^p$ minimization recovers an $\ell^0$ (sparsest) solution, in terms of the count of nonzero parameters of the active neurons in the network. We also show that no $\ell^p$ (for any $0 < p < 1$) or $\ell^0$ solutions has more than $N$ active neurons and, if the data is in general position, any $\ell^0$ solution has $O(N)$ active input weight/bias parameters among these active neurons (Proposition 4.1).

3. **A principled, differentiable objective for sparse ReLU networks.** Our theory provides the first rigorous justification for using a smooth $\ell^p$ penalty for $0 < p < 1$ to obtain truly sparsest interpolating ReLU networks via gradient-based methods—no pruning or complex post-hoc approaches required.

## 2 Related work

**Sparsity via $\ell^p$ minimization in finite-dimensional linear models:** $\ell^p$ penalties with $0 < p \leq 1$ for linear constraint problems have been studied extensively in the compressed sensing literature, and have been shown to yield exact $\ell^0$ minimizers under certain conditions (typically involving restricted isometry and/or null-space constants) on the measurement matrix (Candes and Tao (2005); Chartrand (2007); Chartrand and Staneva (2008); Foucart and Lai (2009)). Such penalties have also been studied in the statistics literature under the name *bridge regression* (Frank and Friedman (1993); Knight and Fu (2000); Fan and Li (2001)). Existing theory in these areas is highly dependent on the fixed, finite-dimensional nature of the linear constraint, and is not readily adaptable to the neural network context, wherein the features themselves are adaptively learned.

---

[2]We use *polyhedron* to refer to an intersection of finitely many closed halfspaces, and *polytope* to refer to a bounded polyhedron. Both are necessarily convex.

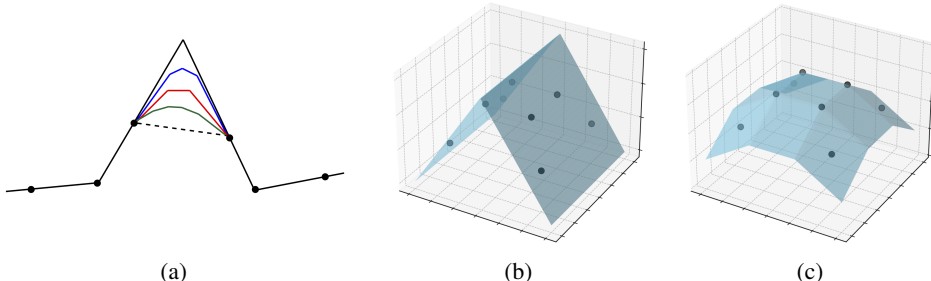

(a)    (b)    (c)

Figure 1: Fig. 1a shows several univariate min-$\ell^1$ path norm interpolants of a given dataset. Such solutions are generally non-unique, and always include at least one sparsest interpolant (black), but also include arbitrarily non-sparse interpolants (blue, red, green). Figs. 1b and 1c: two different ReLU network interpolants of the same 2D dataset with different numbers of active neurons and parameters. Fig. 1b has 5 nonzero input weight/bias parameters (its $\ell^0$ path norm as in (9)), while Fig. 1c has 16.

**$\ell^1$ path norm regularization in single-hidden-layer ReLU networks:**   Neyshabur et al. (2015) showed that the $\ell^1$ path norm of a single-hidden-layer ReLU network controls its Rademacher complexity and thus its generalization gap, but do not directly address the question of sparsity. In the context of *infinite-width* ReLU networks, the problem of minimum-$\ell^1$ path norm interpolation is known to have solutions with no more active neurons than the number of data points (Parhi and Nowak (2021, 2022); Shenouda et al. (2024)).[3] However, solutions to that problem are known to be non-unique, and generally include interpolating ReLU networks with arbitrarily many active neurons (Hanin (2022); Debarre et al. (2022)). Nakhleh et al. (2024) show that a variant of $\ell^1$ path norm minimization applied to univariate-input, multi-output networks always yields a solution with no more than $N$ active neurons, but this solution rarely coincides with the sparsest solution unless the dataset is of a very particular form. Therefore, $\ell^1$ path norm regularization applied to single-hidden-layer ReLU networks is *not* generally guaranteed to produce sparsest solutions.

**Empirical methods for training sparse neural networks:**   A large body of research has been dedicated to sparsity-promoting neural network training schemes. Here we briefly summarize some of the most well-known strategies as well as some which resemble our proposed regularization approach; our list is by no means comprehensive. Earlier works suggested using $\ell^1$ and $\ell^2$ penalties to encourage small network weights (Ng (2004); Hinton and Van Camp (1993)) or applying post-training pruning approaches (LeCun et al. (1989); Hassibi et al. (1993)). More recent pruning schemes incorporate pruning iteratively into training (Han et al. (2015); Guo et al. (2016); Frankle and Carbin (2018); Zhang et al. (2018); Zhou et al. (2019)). Group lasso-type penalties to induce structured sparsity over neurons or channels have also been suggested (Wen et al. (2016); Scardapane et al. (2017)). Other approaches include $\ell^0$ approximation using explicit gating mechanisms (Louizos et al. (2018); Srinivas and Babu (2017)) and variational dropout (Molchanov et al. (2017)). Another line of research uses reparameterization tricks to replace non-smooth sparsifying objectives with smooth versions that share the same local and global minimizers (Ziyin and Wang (2023); Kolb et al. (2023, 2025)). Finally, a number of different algorithms for $\ell^p$-type regularization ($p < 1$) in neural networks have been proposed and evaluated experimentally (Wu et al. (2014); Khan et al. (2018); Tang et al. (2023); Outmezguine and Levi (2024); Ji et al. (2025)). While these methods have demonstrated empirical success in training sparse networks, existing theory does not guarantee that any of them will find sparsest solutions. Moreover, these approaches often require complex multi-stage pipelines and are computationally costly to implement.

**Provable sparsest-recovery in specialized neural network settings:**   In the 1D input case, Boursier and Flammarion (2023) show that, under certain assumptions on the data—namely, that the data contains no more than three consecutive points on which the straight-line interpolant is strictly convex or concave—interpolation using a bias-penalized $\ell^1$ path norm regularizer will select a sparsest interpolant of the dataset. As we will see in Section 3, this assumption on the data is rather restrictive,

---

[3]For input dimension greater than one, the $\ell^1$ path norm $\sum_{k=1}^{K} |v_k| \|\boldsymbol{w}_k\|_2$ studied in those works differs from the one we consider in (8), which is equivalent to $\sum_{k=1}^{K} |v_k| \|\boldsymbol{w}_k\|_1$ for $p = 1$.

and our analysis does not require it. Their proof is also not readily extendable to multivariate inputs. Debarre et al. (2022) characterize the *sparsest* min-$\ell^1$ path norm interpolants in the univariate case and provide an algorithm for explicitly constructing one such solution. Ergen and Pilanci (2021) show that $\ell^1$ path norm minimization yields solutions with a minimal number of active neurons *if* the data dimension is greater than the number of samples (precluding the univariate-input case) and the data satisfy special assumptions, such as whitened data. Fridovich-Keil and Pilanci (2025) show that an iterative hard thresholding algorithm applied to shallow ReLU networks recovers sparsest solutions with high probability if the data is Gaussian. In contrast, our sparsity results do not require any assumptions on the data, and provide exact sparsity guarantees in arbitrary input dimension.

# 3  Univariate $\ell^p$-regularized neural networks

Here we consider single-hidden-layer $\mathbb{R} \to \mathbb{R}$ ReLU neural networks of the form

$$f_{\boldsymbol{\theta}}(x) := \sum_{k=1}^{K} v_k(w_k x + b_k)_+ + ax + c \tag{1}$$

where $(\cdot)_+ := \max\{0, \cdot\}$ is the ReLU function, $\boldsymbol{\theta} := \left\{\{w_k, b_k, v_k\}_{k=1}^{K}, a, c\right\}$ is the collection of network parameters, and all parameters are $\mathbb{R}$-valued. For a given dataset $(x_1, y_1), \ldots, (x_N, y_N) \in \mathbb{R} \times \mathbb{R}$, a fixed $p \in (0, 1]$, and a fixed width $K \geq N$,[4] consider the following problem:

$$\arg\min_{\boldsymbol{\theta}} \sum_{k=1}^{K} |w_k v_k|^p , \text{ subject to } f_{\boldsymbol{\theta}}(x_i) = y_i, \ i = 1, \ldots, N \tag{2}$$

We will refer to the quantity being minimized in (2) as the network's $\ell^p$ *path norm.* Additionally, consider the "sparsifying" problem

$$\arg\min_{\boldsymbol{\theta}} \sum_{k=1}^{K} \mathbb{1}_{w_k v_k \neq 0} , \text{ subject to } f_{\boldsymbol{\theta}}(x_i) = y_i, \ i = 1, \ldots, N \tag{3}$$

where the $\ell^0$ path norm $\sum_{k=1}^{K} \mathbb{1}_{w_k v_k \neq 0}$—which is equivalent to the limit of the $\ell^p$ path norm as $p \downarrow 0$—counts the number of active neurons in the network.

In this section, we will analyze the relationship between solutions of (2) and (3) in terms of their represented functions, and show that these functions can be explicitly described in terms of the geometry of the data points. This characterization (Theorem 3.1) shows that solutions to (2) for any $0 < p < 1$ are necessarily also solutions for $p = 1$, immediately implying data-dependent bounds on the network's parameters and Lipschitz constant. This description also allows problem (2) to be reduced to a minimization of a continuous, strictly concave function over a polytope. From there, we show in Theorem 3.2 that solutions to (2) are unique (in terms of their represented functions) for Lebesgue-almost every $0 < p < 1$ and that, for small enough $p$, this unique optimal function is also a *sparsest* interpolant of the data (i.e., a solution to (3)). Furthermore, if the data meets certain easily-verifiable geometric assumptions, solutions to (2) for *any* $0 < p < 1$ are solutions to the sparsest-interpolation problem (3).

## 3.1  Variational reformulation of (2) and (3)

We begin by showing that problems (2) and (3) can be equivalently expressed as a type of variational problem over the set of continuous piecewise linear (CPWL) functions which interpolate the data. This equivalence is critical for the analysis in this section, since it allows solutions to (2) and (3) to be characterized geometrically in terms of their represented functions and their local behavior around data points. Here, we let $S_{\boldsymbol{\theta},p}^*$ (resp. $S_{\boldsymbol{\theta},0}^*$) denote the set of parameters of optimal neural networks which solve (2) (resp. (3)) for a given dataset, and let

$$S_p^* := \{f : \mathbb{R} \to \mathbb{R} \mid f = f_{\boldsymbol{\theta}}, \ \boldsymbol{\theta} \in S_{\boldsymbol{\theta},p}^*\} \tag{4}$$

---

[4]Here and in Section 4 we fix $K \geq N$ because interpolation in any dimension is possible with $K = N$ neurons (Bubeck et al. (2020), Proposition 2). We will show that solution sets of the $\ell^p$ and $\ell^0$ path norm minimization problems for any input dimension are invariant to the selection of $K$ as long as $K \geq N$ (Corollary 3.1.1 and Proposition 4.1).

be the set of functions represented by neural networks with optimal parameters in $S^*_{\boldsymbol{\theta},p}$, for any $0 \le p \le 1$.

**Proposition 3.1.** *For any $0 \le p \le 1$, the set $S^*_p$ is exactly the solution set of*

$$\arg\min_f \; V_p(f) \,, \; \text{subject to } f(x_i) = y_i, \, i = 1, \dots, N \tag{5}$$

*where the optimization in (5) is taken over all $f : \mathbb{R} \to \mathbb{R}$ which are continuous piecewise linear (CPWL) with at most $K$ knots. For such CPWL functions $f$, we define*

$$V_p(f) := \begin{cases} \sup_{\mathcal{P}} \sum_{i=0}^{n_{\mathcal{P}}-1} |Df(x_{i+1}) - Df(x_i)|^p = \sup_{\pi} \sum_{A \in \pi} |D^2 f(A)|^p, & \text{if } 0 < p \le 1 \\ \text{number of knots of } f, & \text{if } p = 0 \end{cases} \tag{6}$$

*with the first* sup *taken over all partitions $\mathcal{P} = \{x_0 < \cdots < x_{n_{\mathcal{P}}}\}$ of $\mathbb{R}$, and the second* sup *taken over partitions $\pi$ of $\mathbb{R}$ into countably many disjoint (Borel) measurable subsets. In particular, $S^*_0$ is non-empty.*

**Remark 1.** *For $p \in (0, 1]$, $V_p(f)$ is the p-variation (Dudley and Norvaiša (2006), Part II.2) of the distributional derivative $Df$ (in the sense of functions), or equivalently of the second distributional derivative $D^2 f$ (in the sense of measures). In particular, for a CPWL function $f$ with knots at $u_1, \dots, u_K$ and corresponding slope changes $c_1, \dots, c_K$ at those knots, so that $D^2 f = \sum_{k=1}^{K} c_k \delta_{u_k}$, we have*

$$V_p(f) = \sum_{k=1}^{K} |c_k|^p$$

*In the case $p = 1$, $V_1(f)$ is exactly the total variation of $Df$ (in the sense of functions) and of $D^2 f$ (in the sense of measures), and the reformulation in Proposition 3.1 is equivalent to that of Savarese et al. (2019). For a neural network where no two neurons "activate" at the same location (i.e., $b_k / w_k = b_{k'} / w_{k'}$ for $k \ne k'$), $V_p(f)$ is exactly the $\ell^p$ path norm of $f$ as defined above.*

The proof is in Appendix A.1.1. Proposition 3.1 says that the set $S^*_p$ of functions represented by solutions to (2) is exactly the set of CPWL functions $f$ which interpolate the data with minimal sum of absolute slope changes, each taken to the $p^{\text{th}}$ power. In the case $p = 0$, solutions to (3) represent CPWL functions which interpolate the data with the fewest possible knots. This reformulation also shows that problem (3) is invariant to the choice of network width $K$, as long as $K$ is large enough to allow interpolation. As a consequence of Theorem 3.1, we will see that this same width-invariance holds for problem (2).

## 3.2 Geometric characterization of solutions to (5)

Next, in Theorem 3.1, we describe a set of geometric characteristics which any optimal network function $f \in S^*_p$ for $0 < p < 1$ must satisfy, and which at least one $f \in S^*_0$ satisfies. This characterization depends on the slopes $s_i := \frac{y_{i+1} - y_i}{x_{i+1} - x_i}$ of the straight lines $\ell_i$ connecting $(x_i, y_i)$ and $(x_{i+1}, y_{i+1})$. The *discrete curvature* at a data point $x_i$ refers to $\epsilon_i := \text{sgn}(s_i - s_{i-1})$, which is positive if the slope of the straight lines between consecutive data points increases at $x_i$, and negative if this slope decreases (with $\text{sgn}(0) = 0$).

In words, Theorem 3.1 says that the behavior of any $f \in S^*_p$ for $0 < p < 1$ is uniquely determined everywhere *except* around sequences of more than three consecutive data points $x_i, \dots, x_{i+m}$ with the same discrete curvature. On these "constant-curvature" regions of potential ambiguity, solutions must be convex (resp. concave) if the curvature of the data is positive (resp. negative), and can have at most $m$ knots on any such region. Additionally, Theorem 3.1 says that solutions to (5) for $0 < p < 1$ have at most $N - 2$ knots. Therefore, as in the case $p = 0$, we see that problem (2) is invariant (in terms of represented functions) to the choice of network width $K$, as long as $K \ge N - 2$.

**Theorem 3.1.** *For $0 < p < 1$, solutions exist to (5) (hence to (2)). For any such solution, its represented function $f \in S^*_p$ is CPWL and obeys the following:*

1. *$f$ is linear before $x_2$ and after $x_N$; between any three or more consecutive collinear data points; and between any two consecutive points $x_i$ and $x_{i+1}$ with opposite discrete curvature $\epsilon_i \ne \epsilon_{i+1}$.*

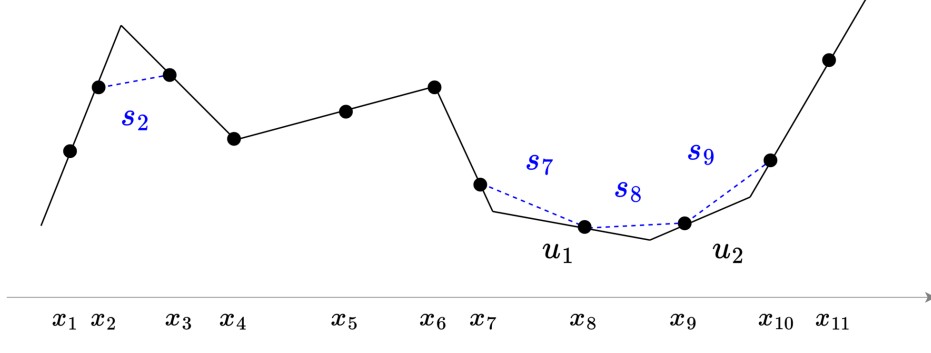

(a) A function satisfying the description in Theorem 3.1.

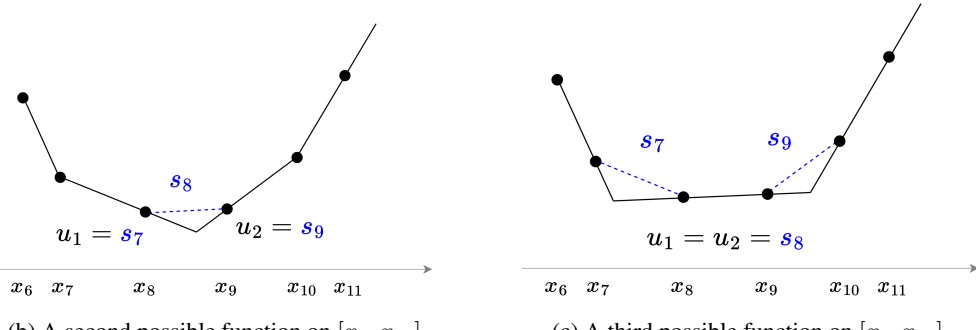

(b) A second possible function on $[x_7, x_{10}]$.

(c) A third possible function on $[x_7, x_{10}]$.

Figure 2: Illustration of Theorem 3.1. By Theorem 3.1,1, any $f \in S_p^*$ for $0 < p < 1$ must agree with the function in Fig. 2a on $(-\infty, x_7]$ and $[x_{10}, \infty)$. The only possible ambiguity occurs between $x_7$ and $x_{10}$, where all points have the same discrete curvature. Here the function behavior is described by Theorem 3.1,2b. Figs. 2b and 2c show two other functions whose behavior on $[x_7, x_{10}]$ also concurs with Theorem 3.1,2b.

2. *On any maximal set of $m$ consecutive data points $x_i, \ldots, x_{i+m}$ with the same discrete curvature (i.e., $\epsilon_{i-1} \neq \epsilon_i = \epsilon_{i+1} = \cdots = \epsilon_{i+m} \neq \epsilon_{i+m+1}$):*

    (a) *If $m = 1$, then $f$ has a single knot between $x_i$ and $x_{i+1}$, with incoming/outgoing slopes $s_{i-1}$ at $x_i$ and $s_{i+1}$ at $x_{i+1}$.*

    (b) *If $m \geq 2$, then $f$ has incoming slope $s_{i-1}$ at $x_i$ and outgoing slope $s_{i+m}$ at $x_{i+m}$. Between $x_i$ and $x_{i+m}$, $f$ takes on at most $m - 1$ slopes $u_1, \ldots, u_{m-1}$ distinct from $s_{i-1}$ and $s_{i+m}$. Each $u_j$ is between $s_{i+j-1}$ and $s_{i+j}$, inclusive, and its corresponding segment passes through $(x_{i+j}, y_{i+j})$.*

*Furthermore, there is always some $f \in S_0^*$ which obeys the above description. (See illustration in Fig. 2.)*

**Corollary 3.1.1.** *Any minimum $\ell^p$ path norm interpolant of the data for $0 < p < 1$ is also a minimum $\ell^1$ path norm interpolant, and can be represented by a network with no more than $N - 2$ neurons.*

The set $S_1^*$ of optimal neural network functions for $p = 1$ has been fully characterized in previous work (Hanin (2022); Debarre et al. (2022)), which showed that any interpolant $f$ obeying the description in Theorem 3.1 is in $S_1^*$. Therefore, Theorem 3.1 shows that any solution to (5) (hence to (2)) for $0 < p < 1$ is also a solution for $p = 1$. This result is interesting because, as our proof of Theorem 3.1 shows, problem (5) generally has multiple solutions for $p = 0$, many of which are *not* solutions for $p = 1$ and may have arbitrarily large slope changes which cannot be bounded in terms of the data. Intuitively, the latter fact is unsurprising, since the objective $V_0(f)$ depends only on the number of knots of $f$, not on the magnitudes of the corresponding slope changes. One might therefore expect that penalizing $V_p$ for sufficiently small $p$ could also produce solutions with arbitrarily large slope changes (corresponding to networks with arbitrarily large weights), particularly in light of the equivalence between $V_p$ and $V_0$ penalization for sufficiently small $p$, as we demonstrate in Section 3.3.

However, Theorem 3.1 says that this is not the case. Therefore, in conjunction with Theorem 3.2, Theorem 3.1 says that for sufficiently small $p$, penalizing $V_p$ effectively penalizes both $V_0$ *and* $V_1$ simultaneously: i.e., it selects a solution with the fewest possible knots (corresponding to a network with the fewest possible neurons), and whose weights are small in the sense that $\sum_{k=1}^{K} |v_k w_k|$ is minimal. In fact, Theorem 3.1 immediately implies the following data-dependent bounds on the parameters and on the network function's Lipschitz constant:

**Corollary 3.1.2.** *Any solution $\theta$ to* (2) *for $0 < p < 1$ has* $\max_{k=1,\dots,K} |v_k w_k| \leq \sum_{k=1}^{K} |v_k w_k| = \sum_{i=1}^{N-2} |s_{i+1} - s_i|$, *and Lipschitz constant $L \leq \max_{i=1,\dots,N-1} |s_i|$.*

Regarding the $N-2$ neuron bound in Corollary 3.1.1, we note that this bound applies to *any* minimum $\ell^p$ path norm solution for any $0 < p < 1$. In contrast, there exist minimum $\ell^1$ path norm solutions with $N-2$ knots, but also solutions with arbitrarily many knots (Hanin (2022); Debarre et al. (2022)); see Fig. 1a. Solutions for $0 < p < 1$ are thus guaranteed a certain level of sparsity which is *not* enforced by $p = 1$ minimization alone. Sparsest (minimum $\ell^0$) solutions—which we soon show will coincide with an $\ell^p$ path norm solution for small enough $p$—are known to have as many as $N-2$ active neurons and as few as $O(N/2)$ neurons, depending on the structure of data (Debarre et al. (2022)).

The proof of Theorem 3.1 hinges mainly on two auxiliary results, detailed in Appendix A.1.2, which describe the local behavior of any optimal $f \in S_p^*$ between consecutive data points in terms of $f$'s incoming and outgoing slopes at those points. This allows us to characterize when a knot can be removed from any interpolating function while maintaining interpolation and reducing its regularization cost $V_p$. The full proof is in Appendix A.1.3.

### 3.3   Uniqueness and sparsity of solutions to (5) for $0 < p < 1$

Using Theorem 3.1, we show that solutions to (5) are unique for almost every $0 < p < 1$, and for sufficiently small $0 < p < 1$, correspond with globally sparsest interpolants (i.e., interpolants with the fewest total knots). Additionally, Theorem 3.1 shows that under an easily-verifiable condition on the data, penalizing $V_p$ for *any* $0 < p < 1$ yields a sparsest interpolant. In conjunction with Theorem 3.1, this result tells us that for univariate data, $\ell^p$ path norm minimization for sufficiently small $p > 0$ simultaneously minimizes both the $\ell^1$ and $\ell^0$ path norms, producing a unique solution which is both maximally sparse and controlled in terms of its parameters' magnitudes. We note that almost-everywhere uniqueness of solutions to (5) occurs *only* in the $0 < p < 1$ case. In contrast, solutions for both $p = 0$ and $p = 1$ are non-unique in general, and for $p = 1$, they may have infinitely many knots/neurons (Debarre et al. (2022), Hanin (2022)).

**Theorem 3.2.** *For all but finitely many $0 < p < 1$, the solution to* (5) *is unique.*[5] *Furthermore, there is some data-dependent $p^*$ such that the unique solution to* (5) *for any $0 < p < p^*$ is a solution for $p = 0$. If the data contains no more than two consecutive points with the same discrete curvature, then the solution to* (5) *for any $0 < p < 1$ is also a solution for $p = 0$.*

The proof of Theorem 3.2 is in Appendix A.1.4. It relies on Theorem 3.1 in combination with the *Bauer maximum principle* (Aliprantis and Border (2006), Theorem 4.104), which states that any continuous, strictly concave function over a closed, convex set attains a minimum at an extreme point of that set. The main idea is that, using Theorem 3.1, we can recast the problem of finding the minimum-$V_p$ interpolant $f \in S$ (where $S$ denotes the set of functions which meet the description in Theorem 3.1) as a minimization of a continuous, strictly concave function over the hypercube $[0,1]^{m-1}$. This reformulation is possible because, by Theorem 3.1, the only place where these interpolants $f \in S$ may differ is around sequences of points $x_i, \dots, x_{i+m}$ (for $m \geq 2$) which all have the same nonzero discrete curvature. Using the description in Theorem 3.1,2b, the slopes $u_1, \dots, u_{m-1}$ of any $f \in S$ on such an interval $[x_i, x_{i+m}]$ can be expressed as convex combinations $u_j := (1 - \alpha_j) s_{i+j-1} + \alpha_j s_{i+j}$, and any such solution $f \in S$ can be fully identified with its corresponding vector of the parameters $[\alpha_1, \dots, \alpha_{j-1}]^\top \in [0,1]^{m-1}$. Expressed in terms of these parameters $[\alpha_1, \dots, \alpha_{j-1}]^\top \in [0,1]^{m-1}$, the cost $V_p$ is strictly concave. Therefore, by the Bauer

---

[5]Uniqueness here and in the remainder of the discussion only in terms of functions which interpolate the data with the same set of absolute slope changes. If the data contains special symmetries, it may admit multiple distinct interpolating functions which have the same set of absolute slope changes (corresponding to interpolating networks with the same weights).

maximum principle, any $f \in S$ with minimal $V_p$ for $0 < p < 1$ must correspond to one of the finitely many vertices of the cube $[0,1]^{m-1}$. Having restricted the set of possible candidate solutions to this finite set (which can be shown to include at least one sparsest solution), the theorem statement follows from standard analysis arguments.

In the next section, we will show that this general line of reasoning—recast the neural network optimization as a minimization of a strictly concave function over a polytope, and apply the Bauer maximum principle—can also be used to characterize the sparsity of $\ell^p$-regularized multivariate-input ReLU networks, although the machinery underlying the argument is very different.

# 4 Multivariate $\ell^p$-regularized neural networks

Here we consider single-hidden-layer $\mathbb{R}^d \to \mathbb{R}$ ReLU neural networks of the form

$$f_{\boldsymbol{\theta}}(\boldsymbol{x}) := \sum_{k=1}^{K} v_k (\boldsymbol{w}_k^\top \overline{\boldsymbol{x}})_+ \tag{7}$$

with output weights $v_k \in \mathbb{R}$, input weights $\boldsymbol{w}_k \in \mathbb{R}^{d+1}$, and $\overline{\boldsymbol{x}} := [\boldsymbol{x}^\top, 1]^\top$ augments the dimension of the input $\boldsymbol{x}$ to account for a bias term. As before, $\boldsymbol{\theta} := \{v_k, \boldsymbol{w}_k\}_{k=1}^{K}$ is the collection of network parameters. For a given dataset $(\boldsymbol{x}_1, y_1), \ldots, (\boldsymbol{x}_N, y_N) \in \mathbb{R}^d \times \mathbb{R}$, and fixed constants $K \geq N$ and $0 < p < 1$, consider the minimum $\ell^p$ path norm interpolation problem

$$\arg\min_{\boldsymbol{\theta}} \sum_{k=1}^{K} \|v_k \boldsymbol{w}_k\|_p^p \,, \text{ subject to } f_{\boldsymbol{\theta}}(\boldsymbol{x}_i) = y_i, \ i = 1, \ldots, N \tag{8}$$

We will prove that, for small enough $p$, any solution to (8) also solves the "sparsifying" problem

$$\arg\min_{\boldsymbol{\theta}} \sum_{k=1}^{K} \|v_k \boldsymbol{w}_k\|_0 \,, \text{ subject to } f_{\boldsymbol{\theta}}(\boldsymbol{x}_i) = y_i, \ i = 1, \ldots, N \tag{9}$$

The multivariate $\ell^0$ path norm objective in (9) counts the number of nonzero input weight/bias parameters of the *active* neurons[6] in the network. We begin by upper bounding the sparsity of solutions to (9) and showing that, as in the univariate case, problems (8) and (9) are invariant to the selection of width $K$ as long as $K \geq N$.

**Proposition 4.1.** *For any $K \geq N$ and any $0 < p < 1$, solutions to (8) exist, and any such solution has at most $N$ active neurons. The same holds for (9). Furthermore, if the data $\boldsymbol{x}_1, \ldots, \boldsymbol{x}_N$ are in general position,[7] then any solution to (9) has $\sum_{k=1}^{K} \|v_k \boldsymbol{w}_k\|_0 = O(N)$.*

See proof in Appendix A.2.1 for explicit constants in various cases.

To show the equivalence of problems (8) and (9) for sufficiently small $p$, we first show that both problems can be recast as finite- (albeit high-) dimensional optimizations over a linear constraint set. This reformulation is heavily inspired by Theorem 1 in Pilanci and Ergen (2020). Here we denote element-wise inequality for vectors $\boldsymbol{a}, \boldsymbol{b}$ as $\boldsymbol{a} \leq \boldsymbol{b}$. Let $\overline{\boldsymbol{X}} := [\overline{\boldsymbol{x}}_1, \ldots, \overline{\boldsymbol{x}}_N]^\top \in \mathbb{R}^{N \times (d+1)}$ be the matrix of augmented data points $\overline{\boldsymbol{x}}_i := [\boldsymbol{x}_i^\top, 1]^\top$, $\boldsymbol{y} := [y_1, \ldots, y_N]^\top \in \mathbb{R}^N$ be the vector of labels, and $\{\boldsymbol{D}_j\}_{j=1}^{J}$ be the collection of all $N \times N$ binary matrices of the form $\text{diag}(\mathbb{1}[\overline{\boldsymbol{X}}\boldsymbol{u} \geq \boldsymbol{0}])$ for some $\boldsymbol{u} \in \mathbb{R}^{d+1}$. It is known that $J \leq 2 \sum_{k=0}^{d} \binom{N-1}{k}$ (Cover (2006)).

**Lemma 4.1.** *For any $0 < p < 1$ and any $K \geq N$, let $\boldsymbol{\theta} = \{v_k, \boldsymbol{w}_k\}_{k=1}^{K}$ be a solution to (8). Then there is another solution $\boldsymbol{\theta}' = \{v_k', \boldsymbol{w}_k'\}_{k=1}^{K}$ to (8), which is reconstructed from a solution*

---

[6]A neuron $\boldsymbol{x} \mapsto v_k (\boldsymbol{w}_k^\top \overline{\boldsymbol{x}})_+$ is *active* if $v_k \boldsymbol{w}_k \neq \boldsymbol{0}$; i.e., that neuron has a nonzero contribution to the network function.

[7]A set of points $\boldsymbol{x}_1, \ldots, \boldsymbol{x}_N \in \mathbb{R}^d$ are in *general (linear) position* if no $k$ of them lie in a $k-2$ dimensional affine subspace, for $k = 2, 3, \ldots, d+1$. If $N \geq d+1$, this is equivalent to the statement that no hyperplane contains more than $d$ points.

$\{\boldsymbol{\nu}'_j, \boldsymbol{\omega}'_j\}_{j=1}^J$—which necessarily satisfies $|\{j \mid \boldsymbol{\nu}_j \neq 0\}| + |\{j \mid \boldsymbol{\omega}_j \neq 0\}| \leq N$—to the problem

$$\underset{\{\boldsymbol{\nu}_j, \boldsymbol{\omega}_j\}_{j=1}^J \subset \mathbb{R}^{d+1}}{\arg\min} \sum_{j=1}^J \|\boldsymbol{\nu}_j\|_p^p + \|\boldsymbol{\omega}_j\|_p^p, \text{ subject to } \sum_{j=1}^J \boldsymbol{D}_j \overline{\boldsymbol{X}}(\boldsymbol{\nu}_j - \boldsymbol{\omega}_j) = \boldsymbol{y}, \tag{10}$$

$$(2\boldsymbol{D}_j - \boldsymbol{I})\overline{\boldsymbol{X}}\boldsymbol{\nu}_j \geq \boldsymbol{0}, \ (2\boldsymbol{D}_j - \boldsymbol{I})\overline{\boldsymbol{X}}\boldsymbol{\omega}_j \geq \boldsymbol{0}, \ j = 1, \ldots, J \tag{11}$$

as

$$\{\boldsymbol{w}'_k\}_{k=1}^K = \{\boldsymbol{\nu}'_j / \alpha_j \mid \boldsymbol{\nu}'_j \neq 0\} \cup \{\boldsymbol{\omega}'_j / \beta_j \mid \boldsymbol{\omega}'_j \neq 0\} \tag{12}$$

$$\{v'_k\}_{k=1}^K = \{\alpha_j \mid \boldsymbol{\nu}'_j \neq 0\} \cup \{-\beta_j \mid \boldsymbol{\omega}'_j \neq 0\} \tag{13}$$

for any choice of $\alpha_1, \beta_1, \ldots, \alpha_J, \beta_J > 0$. Both solutions satisfy $\sum_{k=1}^K \|v_k \boldsymbol{w}_k\|_0 = \sum_{k=1}^K \|v'_k \boldsymbol{w}'_k\|_0$ as well as $\sum_{k=1}^K \|v_k \boldsymbol{w}_k\|_q^q = \sum_{k=1}^K \|v'_k \boldsymbol{w}'_k\|_q^q$ for any $0 < q < 1$. The same statement holds for solutions $\boldsymbol{\theta}$ to (9), with the objective in (10) replaced by $\sum_{j=1}^J \|\boldsymbol{\nu}_j\|_0 + \|\boldsymbol{\omega}_j\|_0$.

The proof of Lemma 4.1 is in Appendix A.2.2. The main idea is that although there are uncountably many ways to choose the neurons' parameters, there are only $J$ possible binary activation patterns, i.e., vectors representing whether a given neuron is active on each data point. By combining all neurons which induce the same activation pattern into two neurons (one with positive output weight and one with negative output weight), the network's output and $\ell^p$ path norm can be expressed as a sum over these $J$ activation patterns. The equality constraint in (10) reflects the data-fitting requirement, and the inequality constraints ensure that each $\boldsymbol{\nu}_j, \boldsymbol{\omega}_j$ correspond appropriately to the activation pattern $\boldsymbol{D}_j$ in order for the reconstruction formula (12) to hold. With this reformulation in hand, we are ready for the main result of this section:

**Theorem 4.1.** *For any dataset, there is some data-dependent $p^*$ such that any solution to (8) for any $0 < p < p^*$ is a solution to (9).*

The proof, presented in Appendix A.2.3, follows the approaches of Yang et al. (2022a); Peng et al. (2015) in analyzing sparsity of solutions to $\min_{\boldsymbol{z}:\boldsymbol{A}\boldsymbol{z}=\boldsymbol{y}} \|\boldsymbol{z}\|_p^p$ for an underdetermined linear system $\boldsymbol{A}\boldsymbol{z} = \boldsymbol{y}$, with modifications to account for the linear inequality constraints in (10). The fundamental observation is that the linear constraints in (10) determine a polytope, and the map $\boldsymbol{z} \mapsto \|\boldsymbol{z}\|_p^p$ is strictly concave on each individual orthant and invariant to absolute values of vector elements. By projecting the constraint set of (10) into the nonnegative orthant, the problem turns into a minimization of a continuous, strictly concave function over a polytope. By the Bauer maximum principle, any solution to this problem occurs at one of the finitely many vertices of that polytope, and by appropriately normalizing the vertices of this polytope, we are able to demonstrate the desired result.

Although Theorem 4.1 applies to any input dimension, thus recovering part of the result of Theorem 3.2, our multivariate analysis does not immediately recover the univariate results on functional structure or uniqueness of solutions; nor does it demonstrate that solutions for $0 < p < 1$ are always solutions for $p = 1$, as was shown in the univariate case (Corollary 3.1.1). Thus, although Theorem 4.1 guarantees exact sparsest recovery for sufficiently small $p$ in arbitrary input dimensions, the multivariate problem leaves many interesting open questions, which we save for future work.

## 5  Experiments

We perform several simple experiments on synthetic data which suggest that our proposed $\ell^p$ path norm lends itself to practical application, recovering far sparser solutions more quickly than unregularized or weight decay-regularized gradient-based training. To implement our regularizer, we use a proximal gradient algorithm based on the iteratively reweighted $\ell^1$ method of Candes et al. (2008); Figueiredo et al. (2007), the details of which are summarized in Appendix A.3.1. Fig. 3 shows the sparsity over time of networks trained with our reweighted $\ell^1$ algorithm for three different values of $p \in \{0.4, 0.7, 1\}$, as well as with unregularized Adam and AdamW weight decay, on two different synthetic datasets. For all values of $p$, the $\ell^p$-regularized networks are much sparser much earlier in training than the unregularized or weight decay regularized networks, with the $p = 0.4$ networks being the sparsest. For the univariate synthetic dataset, the $p = 0.4$ regularized network recovers

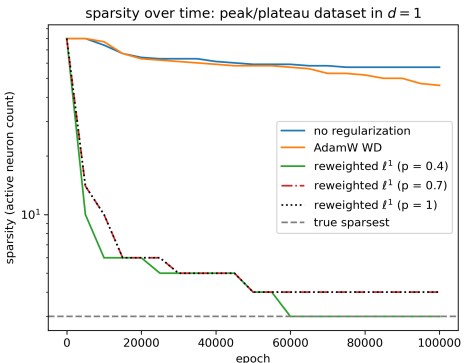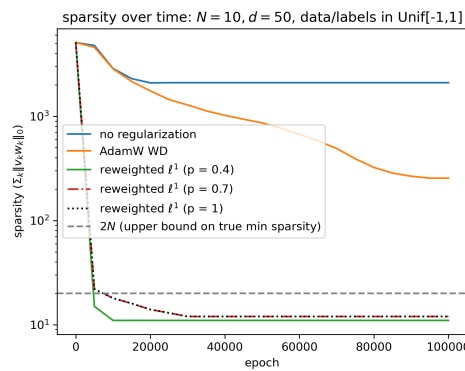

Figure 3: Sparsity over time of networks trained to interpolation with a reweighted $\ell^1$ algorithm (see Appendix A.3.1) for $\ell^p$ path norm regularization, $p \in \{0.4, 0.7, 1\}$, and of unregularized and weight decay-regularized networks. Results on the left are for a synthetic univariate "peak/plateau" dataset, and results on the right are for a high-dimensional set of random data and labels. The gray dashed lines reflect the true minimal sparsity (in the univariate case, left) and the upper bound on the minimal sparsity guaranteed by Proposition 4.1 in the multivariate case (right). For further details, results, and discussion, see Appendix A.3.2.

the true sparsest solution, and for the multivariate synthetic dataset, all $\ell^p$ regularized networks recover solutions which obey the sparsity upper bound guaranteed by Proposition 4.1. For further details, results, and discussion, see Appendix A.3.2. Code for these experiments is available at `https://github.com/julianakhleh/sparse_nns_lp`.

## 6   Conclusion and Discussion

We have introduced a smooth, $\ell^p$ path norm ($0 < p < 1$) regularization framework whose global minimizers provably coincide with the sparsest ReLU network interpolants for sufficiently small $p$, thus recasting the combinatorial $\ell^0$ minimization problem as a differentiable objective compatible with gradient descent. In the univariate case, we showed minimum $\ell^p$ path norm interpolants are unique for almost every $0 < p < 1$; never require more than $N - 2$ neurons; and are also $\ell^1$ minimizers, yielding explicit data-dependent parameter and Lipschitz bounds. In arbitrary dimensions, we demonstrate a similar $\ell^p$-$\ell^0$ equivalence for sufficiently small $p$. Our proposed regularization objective offers a principled, gradient-based alternative to heuristic pruning methods for training truly sparse neural networks.

While we demonstrate the existence of $p$ small enough for $\ell^p/\ell^0$ minimization equivalence, our proofs do not yield an efficient way to compute the "critical threshold" $p^*$, although they do demonstrate that estimating this $p^*$ is in theory possible by enumerating an exponential number of vertices of a data-dependent polytope. Whether or not $p^*$ can be computed or estimated *efficiently* is an open question of interest for future work. Other possible directions of interest are to extend our results here to multi-output and deep architectures and to other notions of sparsity (such as sparsity over entire neurons vs. parameters in the multi-dimensional case).

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

# A Proofs of main results

## A.1 Univariate results

### A.1.1 Proof of Proposition 3.1

*Proof.* By homogeneity of the ReLU—meaning that $(\alpha x)_+ = \alpha(x)_+$ for any $\alpha > 0$—any ReLU neural network of the form (1) can have its parameters rescaled as $v_k \mapsto |w_k| v_k$, $(w_k, b_k) \mapsto |w_k|^{-1}(w_k, b_k)$ without changing the network's represented function or its $\ell^p$ path norm. Therefore, any $f \in S_p^*$ can be expressed as a neural network of the form (1) with $|w_k| = 1$ for all $k = 1, \ldots, K$. Additionally, any $f \in S_p^*$ can be expressed as a network where no two neurons "activate" at the same location, i.e., $b_k/w_k \neq b_{k'}/w_{k'}$ whenever $k \neq k'$. To see this, consider a neural network $f_{\boldsymbol{\theta}}$ with unit-norm input weights which contains two distinct neurons $k, k'$ with $b_k/w_k = b_{k'}/w_{k'}$. The sum of these neurons can be rewritten as

$$v_k(w_k x + b_k)_+ + v_{k'}(w_{k'} x + b_{k'})_+ = (v_k + v_{k'})(w_k x + b_k)_+ \tag{14}$$

if $w_k = w_{k'}$, or as

$$v_k(w_k x + b_k)_+ + v_{k'}(w_{k'} x + b_{k'})_+ = (v_k + v_{k'})(w_k x + b_k)_+ - v_{k'}(w_k x + b_k) \tag{15}$$

if $w_k = -w_{k'}$. (The latter uses the identity $x = (x)_+ - (-x)_+$.) In either case, we see that the original two neurons $k, k'$ can be replaced with a single neuron and, in the latter case, an additive affine term. Because the affine term does not contribute to $\ell^p$ path norm, and because $|v_k + v_{k'}|^p \leq |v_k|^p + |v_{k'}|^p$ for $p \in (0, 1]$, the resulting network represents the same function as the original one with no greater regularization cost.

Furthermore, any neural network of the form (1) with unit-norm input weights and $K$ active neurons, where no two active neurons activate at the same location, is a CPWL function with $K$ knots, where knot $k$ is located at $-b_k/w_k$, and the slope change of the function at knot $k$ is $v_k$. Conversely, any $\mathbb{R} \to \mathbb{R}$ CPWL function $f$ with $K$ knots at locations $u_1 < \cdots < u_K$ and corresponding slope changes $v_1, \ldots, v_K$ can be expressed as

$$f(x) = f(u_0) + f'(u_0)(x - u_0) + \sum_{k=1}^{K} v_k(x - u_k)_+ \tag{16}$$

for some arbitrary point $u_0 < u_1$. Any such $f$ has $D^2 f = \sum_{k=1}^{K} v_k \delta_{u_k}$, so that $V_p(f) = \sum_{k=1}^{K} |v_k|^p$, and $V_0(f) = \sum_{k=1}^{K} \mathbb{1}_{v_k \neq 0} = K$.

These facts are sufficient to establish the equivalence of problems (2) and (5). Indeed, let $\overline{S}_{\boldsymbol{\theta},p}^*$ denote the set of optimal parameters for a modified version of problem (2) which imposes the additional constraints that each $|w_k| = 1$ and that $b_k/w_k \neq b_{k'}/w_{k'}$ whenever $k \neq k'$. For some $\boldsymbol{\theta}^* \in \overline{S}_{\boldsymbol{\theta},p}^*$, let $C^*$ denote its $\ell^p$ path norm. We have shown that $S_p^*$ can be equivalently expressed as

$$S_p^* = \{f : \mathbb{R} \to \mathbb{R} \mid f = f_{\boldsymbol{\theta}}, \boldsymbol{\theta} \in \overline{S}_{\boldsymbol{\theta},p}^*\} \tag{17}$$
$$= \{f : \mathbb{R} \to \mathbb{R} \mid f \text{ is CPWL with } \leq K \text{ knots}, V_p(f) = C^*, f(x_i) = y_i, i = 1, \ldots, N\} \tag{18}$$

which is exactly the set of minimizers of (5). Non-emptiness of $S_{0,\boldsymbol{\theta}}^*$ (and thus of $S_0^*$) follows from non-emptiness of the feasible set $\Theta$ of (3) when $K \geq N$, and the fact that the objective values of members of the feasible set lie in $\{1, \ldots, K\}$, on which a minimum is achieved. $\square$

### A.1.2 Auxiliary lemmas: local behavior of $f$ around same/opposite sign slope changes

Our proof of Theorem 3.1 relies strongly on the following two auxiliary lemmas, which describe the local behavior of any $f \in S_p^*$ for $0 \leq p < 1$ between consecutive data points. Here we denote the incoming and outgoing slopes of any interpolant $f$ at a data point $x_i$ as $s_{\text{in}}(f, x_i)$ and $s_{\text{out}}(f, x_i)$, respectively (sometimes dropping the explicit reference to $f$ if it is clear from context). First, we show in Lemma A.1 that for any optimal network function $f \in S_p^*$, $0 \leq p < 1$, if the signs of $s_i - s_{\text{in}}(f, x_i)$ and $s_{\text{out}}(f, x_{i+1}) - s_i$ agree, then $f$ connects $(x_i, y_i)$ and $(x_{i+1}, y_{i+1})$ in a single "peak" (see Fig. 4a).

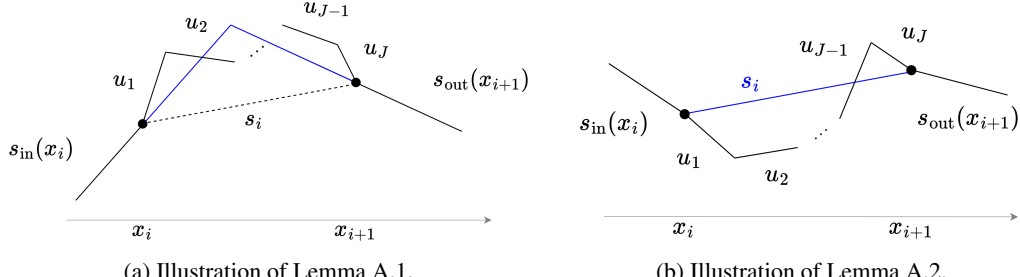

(a) Illustration of Lemma A.1.                    (b) Illustration of Lemma A.2.

Figure 4: Left: Illustration of the case $\mathrm{sgn}\left(s_i - s_{\mathrm{in}}(f, x_i)\right) = \mathrm{sgn}\left(s_{\mathrm{out}}(f, x_{i+1}) - s_i\right)$ addressed in Lemma A.1. Right: illustration of the case $\mathrm{sgn}\left(s_i - s_{\mathrm{in}}(f, x_i)\right) \neq \mathrm{sgn}\left(s_{\mathrm{out}}(f, x_{i+1}) - s_i\right)$ addressed in Lemma A.2. In both cases, the functions in black have strictly greater $V_p$ for $0 \le p < 1$ than the functions in blue.

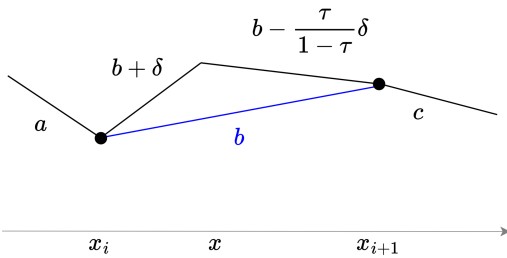

Figure 5: Base case of Lemma A.2, where we consider the possibility that $f \in S_p^*$ for some $0 \le p < 1$ has a single knot at some $x \in (x_i, x_{i+1})$ where $\mathrm{sgn}(a - b) \neq \mathrm{sgn}(b - c)$. Here $\tau := \frac{x - x_i}{x_{i+1} - x_i}$.

**Lemma A.1** (Behavior of $f \in S_p^*$ around same-sign slope changes)**.** *For $0 \le p < 1$, suppose that $f \in S_p^*$ has $\mathrm{sgn}\left(s_i - s_{\mathrm{in}}(f, x_i)\right) = \mathrm{sgn}\left(s_{\mathrm{out}}(f, x_{i+1}) - s_i\right)$ at consecutive data points $x_i, x_{i+1}$. If both signs are zero, then $f$ is linear on the interval $I := [x_i - \delta, x_{i+1} + \delta]$ surrounding $[x_i, x_{i+1}]$, for small $\delta > 0$. Otherwise, $f$ has a single knot on $I$, between $x_i$ and $x_{i+1}$. (See Fig. 4a.)*

*Proof.* If both signs are zero, then $f$ must be linear on $I$, since anything else would have nonzero $V_p(f|_I)$ for $0 \le p < 1$. If both signs are nonzero, observe that

$$|s_{\mathrm{out}}(f, x_{i+1}) - s_{\mathrm{in}}(f, x_i)|^p < |s_{\mathrm{out}}(f, x_{i+1}) - u_J|^p + |u_J - u_{J-1}|^p + \cdots + |u_2 - u_1|^p + |u_1 - s_{\mathrm{in}}(f, x_i)|^p$$

for any $u_1, \ldots, u_J$ which are all distinct from each other and from $s_{\mathrm{in}}(f, x_i)$ and $s_{\mathrm{out}}(f, x_{i+1})$. This is a simple consequence of the inequality $|a + b|^p \le |a|^p + |b|^p$, which holds for any $a, b \in \mathbb{R}$ and any $0 < p < 1$ and is strict unless $a = 0$ or $b = 0$. Since any interpolant with more than one knot on $I$ has one or more intermediate slopes $u_1, \ldots, u_J$ between $x_i$ and $x_{i+1}$, the result follows. $\quad\square$

Next, Lemma A.2 says that if the signs of $s_i - s_{\mathrm{in}}(f, x_i)$ and $s_{\mathrm{out}}(f, x_{i+1}) - s_i$ of an optimal $f \in S_p^*$, $0 < p < 1$ disagree, then $f$ is linear between $x_i$ and $x_{i+1}$.

**Lemma A.2** (Behavior of $f \in S_p^*$ around opposite-sign slope changes)**.** *For $0 \le p < 1$, suppose that $f \in S_p^*$ has $\mathrm{sgn}\left(s_i - s_{\mathrm{in}}(f, x_i)\right) \neq \mathrm{sgn}\left(s_{\mathrm{out}}(f, x_{i+1}) - s_i\right)$ at consecutive data points $x_i, x_{i+1}$. If $0 < p < 1$, then $f$ is linear between $x_i$ and $x_{i+1}$. If $p = 0$, then either $f$ is linear between $x_i$ and $x_{i+1}$, or it agrees outside of $[x_i, x_{i+1}]$ with some $g \in S_0^*$ which is linear between $x_i$ and $x_{i+1}$. (See illustration in Fig. 4b.)*

*Proof.* First consider the base case illustrated in Fig. 5, where we suppose that $f \in S_p^*$ for some $0 \le p < 1$ has a single knot at some $x \in (x_i, x_{i+1})$. To simplify the notation, we denote $a := s_{\mathrm{in}}(f, x_i)$, $b := s_i$, $c := s_{\mathrm{out}}(f, x_{i+1})$ and $\tau := \frac{x - x_i}{x_{i+1} - x_i}$ and assume that $\mathrm{sgn}(a - b) \neq \mathrm{sgn}(b - c)$. The intermediate slopes $u_1$ and $u_2$ can be parameterized as $u_1 = b + \delta$ and $u_2 = b - \frac{\tau}{1 - \tau}\delta$ for some $\delta \in \mathbb{R}$. Consider the cost $V_p(f|_I)$ of $f$ on the interval $I := (x_i - \epsilon, x_{i+1} + \epsilon)$ (for some arbitrary

$\epsilon > 0$) as a function $C(\delta)$ of the parameter $\delta$. If $p = 0$, then clearly $C(0) = 2 \leq C(\delta) \in \{2, 3\}$ for $\delta \neq 0$. This shows that the function $g$ whose slope is $b$ on $[x_i, x_{i+1}]$ has no greater cost than $f$, and thus $g \in S_0^*$. In the case $0 < p < 1$, we have

$$C(\delta) := |\delta + b - a|^p + \frac{1}{(1-\tau)^p}|\delta|^p + \left|c - b + \frac{\tau}{1-\tau}\delta\right|^p \tag{19}$$

and we will show that $C(0) < C(\delta)$ for $\delta \neq 0$, contradicting the assumption that $f \in S_p^*$.

Note that $C$ is coercive and continuous on $\delta \in \mathbb{R}$, so it attains a minimizer (this follows from the Weierstrass extreme value theorem as applied to the compact sublevel sets of $C$). By Fermat's theorem, any minimizer of $C$ must occur at critical points, i.e., points where the derivative $C'$ is zero or undefined. The three points where $C'$ is undefined are $\delta_1 = a - b$, $\delta_2 = 0$, and $\delta_3 = \frac{1-\tau}{\tau}(b - c)$. Assuming without loss of generality that $\delta_1 < \delta_2 < \delta_3$, note that $C$ is strictly concave on the intervals $(-\infty, \delta_1)$, $(\delta_1, \delta_2)$, $(\delta_2, \delta_3)$, and $(\delta_3, \infty)$. This is because compositions of concave and affine functions are concave, and the function $x \mapsto |x|^p$ for $p \in (0, 1]$ is concave on any subinterval of $\mathbb{R}$ over which $x$ does not change sign. Therefore, any point at which $C' = 0$ will be a local maximum rather than a minimum, and hence any minimum of $C$ can only occur at the critical points $\delta_1, \delta_2, \delta_3$. We have

$$C(\delta_1) = \frac{1}{(1-\tau)^p}|a - b|^p + \left|c + \frac{\tau}{1-\tau}a - \frac{1}{1-\tau}b\right|^p \tag{20}$$

$$C(\delta_2) = |b - a|^p + |c - b|^p \tag{21}$$

$$C(\delta_3) = \left|\frac{1}{\tau}b - \frac{1-\tau}{\tau}c - a\right|^p + \frac{1}{\tau^p}|b - c|^p \tag{22}$$

Now, for the variable $t \in [0, 1)$, define

$$h_1(t) := \frac{1}{(1-t)^p}|a - b|^p + \left|c + \frac{t}{1-t}a - \frac{1}{1-t}b\right|^p \tag{23}$$

and observe that $h_1(0) = C(\delta_2)$ and $h_1(\tau) = C(\delta_1)$. Its derivative is

$$h_1'(t) = \frac{p}{(1-t)^{p+1}}|a - b|^p + p\left|c + \frac{t}{1-t}a - \frac{1}{1-t}b\right|^{p-1}\text{sgn}\left(c + \frac{t}{1-t}a - \frac{1}{1-t}b\right)\frac{a - b}{(1-t)^2} \tag{24}$$

$$= \frac{p}{(1-t)^{p+1}}|a - b|^p + p\left|c + \frac{t}{1-t}a - \frac{1}{1-t}b\right|^{p-1}\text{sgn}\left(\frac{(1-t)(c - b) + t(a - b)}{1-t}\right)\frac{a - b}{(1-t)^2} \tag{25}$$

Assuming that $\text{sgn}(a - b) \neq \text{sgn}(b - c)$ with $a \neq b$ (and thus $\delta_1 \neq \delta_2$), we see that $h_1'(t) > 0$ for all $t \in [0, 1)$. This is because the term inside the sgn above is positive if $a > b$ (so that $b \leq c$) and negative if $a < b$ (so that $b \geq c$). This shows that $h_1(0) = C(\delta_2) < h_1(\tau) = C(\delta_1)$. Similarly, define

$$h_2(t) := \left|\frac{1}{t}b - \frac{1-t}{t}c - a\right|^p + \frac{1}{t^p}|b - c|^p \tag{26}$$

for $t \in (0, 1]$, so that $h_2(\tau) = C(\delta_3)$ and $h_2(1) = C(\delta_2)$. Its derivative is

$$h_2'(t) = p\left|\frac{1}{t}b - \frac{1-t}{t}c - a\right|^{p-1}\text{sgn}\left(\frac{1}{t}b - \frac{1-t}{t}c - a\right)\frac{c - b}{t^2} - \frac{p}{t^{p+1}}|b - c|^p \tag{27}$$

$$= p\left|\frac{1}{t}b - \frac{1-t}{t}c - a\right|^{p-1}\text{sgn}\left(\frac{t(b - a) + (1-t)(b - c)}{t}\right)\frac{c - b}{t^2} - \frac{p}{t^{p+1}}|b - c|^p \tag{28}$$

Assuming that $\text{sgn}(a - b) \neq \text{sgn}(b - c)$ with $b \neq c$ (and thus $\delta_2 \neq \delta_3$), we see that $h_2'(t) > 0$ for all $t \in (0, 1]$. This is because the term inside the sgn above is positive if $b > c$ (so that $a \leq b$) and negative if $b < c$ (so that $a \geq b$). This shows that $h_2(\tau) = C(\delta_3) < h_2(1) = C(\delta_2)$. Therefore, $C(0) < C(\delta)$ for $\delta \neq 0$, as desired.

Next, consider the general case, where we assume by contradiction that $f \in S_p^*$ for $0 \leq p < 1$ may have multiple knots inside $(x_i, x_{i+1})$. As before, in the case $p = 0$, $f$ can't have fewer knots than the function $g$ whose slope is $b$ on $[x_i, x_{i+1}]$; the only way for $f$ to be in $S_0^*$ is if it has a single knot

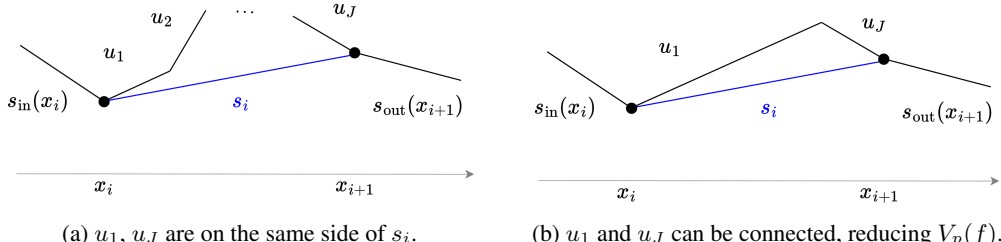

(a) $u_1$, $u_J$ are on the same side of $s_i$.

(b) $u_1$ and $u_J$ can be connected, reducing $V_p(f)$.

Figure 6: General case of Lemma A.2, where the outgoing line segment at $x_i$ and the incoming line segment at $x_{i+1}$ both lie on the same side of the straight line between $(x_i, y_i)$ and $(x_{i+1}, y_{i+1})$. We can apply the argument in the proof of Lemma A.1 to connect these two segments in a single knot inside $(x_i, x_{i+1})$ and strictly reduce $V_p(f)$.

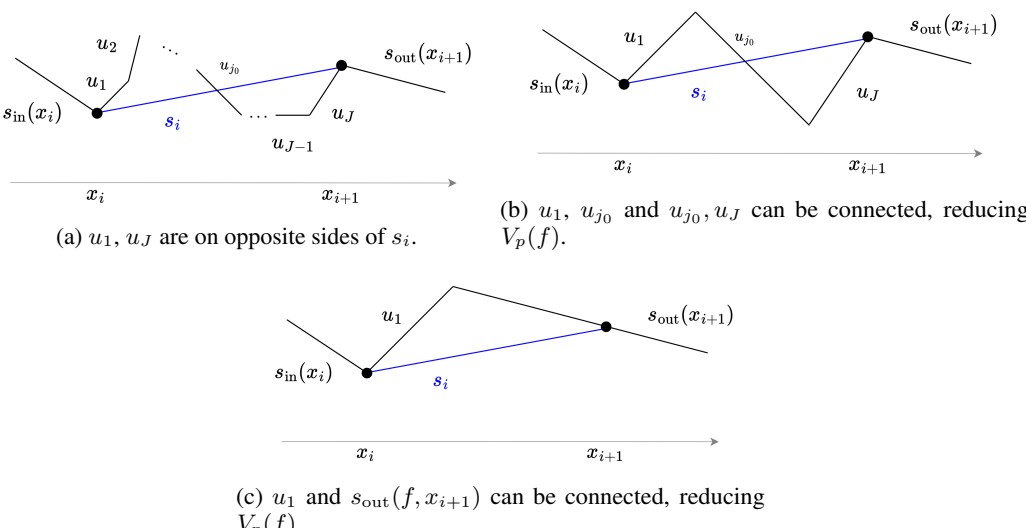

(a) $u_1$, $u_J$ are on opposite sides of $s_i$.

(b) $u_1$, $u_{j_0}$ and $u_{j_0}, u_J$ can be connected, reducing $V_p(f)$.

(c) $u_1$ and $s_{\text{out}}(f, x_{i+1})$ can be connected, reducing $V_p(f)$.

Figure 7: General case of Lemma A.2, where the outgoing line segment at $x_i$ and the incoming line segment at $x_{i+1}$ lie on opposite sides of the straight line between $(x_i, y_i)$ and $(x_{i+1}, y_{i+1})$. We can apply the argument in the proof of Lemma A.1 to connect the segments $u_1$ and $u_{j_0}$ and $u_{j_0}$ and $u_J$, resulting in a function with two knots inside $(x_i, x_{i+1})$ and strictly reducing $V_p(f)$. By the same argument, we can further reduce $V_p(f)$ by connecting $u_1$ and $s_{\text{out}}(f, x_{i+1})$, resulting in a single knot inside $(x_i, x_{i+1})$.

inside $(x_i, x_{i+1})$ and a single knot at either $x_i$ or $x_{i+1}$, in which case we also have $g \in S_0^*$. In the case $0 < p < 1$, let $u_1, \ldots, u_J$ denote the slopes of $f$ on $[x_i, x_{i+1}]$. If the line segments with slopes $u_1$ and $u_J$ lie on the same side of the line segment with slope $s_i$, then we can apply the argument in the proof of Lemma A.1 to remove the segments with slopes $u_2, \ldots, u_{J-1}$ and connect the segments with $u_1$ and $u_J$ in a single knot inside $(x_i, x_{i+1})$; this strictly reduces $V_p(f)$, contradicting $f \in S_p^*$. (See Fig. 6.) If the line segments with slopes $u_1$ and $u_J$ lie on opposite sides of the line segment with slope $s_i$, then either one of the intermediate segments, whose slope we call $u_{j_0}$, crosses the segment with slope $s_i$, or else one of the intermediate segments (again call its slope $u_{j_0}$) lies on one side of $s_i$, and $u_{j_0+1}$ lies on the other side. In either case, the segments $u_1$ and $u_{j_0}$ can be connected and the segments between them removed, as can the segments $u_{j_0}$ (or $u_{j_0+1}$) and $u_J$. (See Fig. 7.) Again, by the logic in the proof of Lemma A.1, this strictly reduces $V_p(f)$, contradicting $f \in S_p^*$. If $f$ is already of the form in Fig. 7b, with only two knots inside $(x_i, x_{i+1})$ on opposite sides of the line $s_i$, then the second knot can be removed by directly connecting $u_1$ and $s_{\text{out}}(f, x_{i+1})$ (see Fig. 7c). By the same logic, this strictly reduces $V_p(f)$, contradicting $f \in S_p^*$. $\qquad\square$

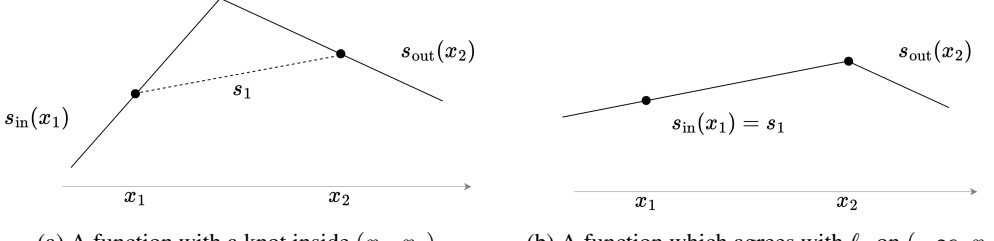

(a) A function with a knot inside $(x_1, x_2)$.    (b) A function which agrees with $\ell_1$ on $(-\infty, x_2]$.

Figure 8: Behavior of $f \in S_p^*$ before $x_2$ and after $x_N$. A knot inside $(x_1, x_2)$ can be moved to $x_2$, maintaining the same outgoing slope at $x_2$, which strictly decreases the magnitude of the slope change at the knot.

### A.1.3    Proof of Theorem 3.1

*Proof.* We first use Theorem 3.1 and Lemmas A.1 and A.2 to show that any $f \in S_p^*$ for $0 < p < 1$ must obey the description in Theorem 3.1, and that there is always some $f \in S_0^*$ which fits this description. Using this result, we argue non-emptiness of $S_p^*$. We break the proof into the following sections.

**Linearity before $x_2$ and after $x_{N-1}$.**    We will prove the statement for $(-\infty, x_2]$; the proof for $[x_{N-1}, \infty)$ is analogous. No $f \in S_p^*$ for $0 \le p \le 1$ can have a knot at or before $x_1$ as this would strictly increase the cost $V_p(f)$ without affecting the ability of $f$ to interpolate the data points. In the case $0 < p < 1$, assume by contradiction that some $f \in S_p^*$ has a knot at some $x \in (x_1, x_2)$. By Lemma A.2, it must be the case that $\text{sgn}(s_1 - s_{\text{in}}(f, x_1)) = \text{sgn}(s_{\text{out}}(f, x_2) - s_1)$, and by Lemma A.1, this knot is the only one inside $(x_1, x_2)$, with $s_{\text{in}}(f, x_1) = s_{\text{out}}(f, x_1)$ and $s_{\text{in}}(f, x_2) = s_{\text{out}}(f, x_2)$. (See Fig. 8a.) Assuming without loss of generality that $\text{sgn}(s_1 - s_{\text{in}}(f, x_1)) = \text{sgn}(s_{\text{out}}(f, x_2) - s_1) = -1$, we have $s_{\text{in}}(f, x_1) > s_1 > s_{\text{out}}(f, x_2)$, and therefore $|s_{\text{out}}(f, x_2) - s_{\text{in}}(f, x_1)| > |s_{\text{out}}(f, x_2) - s_1|$. But this shows that $V_p(f) > V_p(g)$, where $g = \ell_1$ on $(-\infty, x_2]$ and is otherwise identical to $f$. (See Fig. 8b.) This contradicts $f \in S_p^*$.

In the case $p = 0$, fix some $f \in S_0^*$. As argued above, $f$ has no knots on $(-\infty, x_1]$. If $\text{sgn}(s_1 - s_{\text{in}}(f, x_1)) \ne \text{sgn}(s_{\text{out}}(f, x_2) - s_1)$, then by Lemma A.2, either $f = \ell_1$ on $[x_1, x_2]$ (hence it also must agree with $\ell_1$ on $(-\infty, x_1]$), or there is some $g \in S_0^*$ which agrees with $\ell_1$ on $[x_1, x_2]$ (hence also on $(-\infty, x_1]$, since $g$ must also not have any knots on $(-\infty, x_1]$). If $\text{sgn}(s_1 - s_{\text{in}}(f, x_1)) = \text{sgn}(s_{\text{out}}(f, x_2) - s_1) = 0$, then by Lemma A.1, $f = \ell_1$ on $[x_1, x_2]$ and thus also on $(-\infty, x_1]$. If $\text{sgn}(s_1 - s_{\text{in}}(f, x_1)) = \text{sgn}(s_{\text{out}}(f, x_2) - s_1)$ are both nonzero, then by Lemma A.1, $f$ has a single knot inside $(x_1, x_2)$ with $s_{\text{in}}(f, x_1) = s_{\text{out}}(f, x_1)$ and $s_{\text{in}}(f, x_2) = s_{\text{out}}(f, x_2)$, as in Fig. 8a. Then function depicted in Fig. 8, which agrees with $\ell_1$ on $(-\infty, x_2]$ and with $f$ on $[x_2, \infty)$, has the same number of knots as $f$, so $g \in S_0^*$.

**Linearity between data points of opposite curvature.**    For $0 < p < 1$, assume by contradiction that some $f \in S_p^*$ does not agree with $\ell_i$ on an interval $[x_i, x_{i+1}]$ where $\text{sgn}(s_i - s_{i-1}) \ne \text{sgn}(s_{i+1} - s_i)$. By Lemmas A.1 and A.2, it must be the case that $\text{sgn}(s_i - s_{\text{in}}(f, x_i)) = \text{sgn}(s_{\text{out}}(f, x_{i+1}) - s_i)$ are both nonzero, and that $s_{\text{in}}(f, x_i) = s_{\text{out}}(f, x_i)$ and $s_{\text{in}}(f, x_{i+1}) = s_{\text{out}}(f, x_{i+1})$ and $f$ has a single knot inside $(x_i, x_{i+1})$ where the incoming line at $x_i$ and the outgoing line at $x_{i+1}$ meet. It must be the case that $\text{sgn}(s_i - s_{i-1}) \ne \text{sgn}(s_i - s_{\text{in}}(f, x_i))$ and/or that $\text{sgn}(s_{i+1} - s_i) \ne \text{sgn}(s_{\text{out}}(f, x_{i+1}) - s_i)$. Assume without loss of generality that $\text{sgn}(s_{i+1} - s_i) \ne \text{sgn}(s_{\text{out}}(f, x_{i+1}) - s_i) = 1$, so that $s_{i+1} \le s_i < s_{\text{out}}(f, x_{i+1}) = s_{\text{in}}(f, x_{i+1})$. Then clearly $s_{i+1} \ne s_{\text{out}}(f, x_{i+1})$ (in other words, $f$ does not agree with $\ell_{i+1}$ on all of $[x_{i+1}, x_{i+2}]$), so by Lemma A.1 and Lemma A.2, it must be the case that $-1 = \text{sgn}(s_{i+1} - s_{\text{in}}(f, x_{i+1})) = \text{sgn}(s_{\text{out}}(f, x_{i+2}) - s_{i+1})$, that $f$ has a single knot inside $(x_{i+1}, x_{i+2})$, and that $s_{\text{in}}(f, x_{i+2}) = s_{\text{out}}(f, x_{i+2})$. (See Fig. 9a.) Therefore, $s_{\text{in}}(f, x_{i+2}) = s_{\text{out}}(f, x_{i+2}) < s_{i+1} \le s_i < s_{\text{out}}(f, x_{i+1}) = s_{\text{in}}(f, x_{i+1})$. Furthermore, because $1 = \text{sgn}(s_{\text{out}}(f, x_{i+1}) - s_i) = \text{sgn}(s_i - s_{\text{in}}(f, x_i))$, we have $s_{\text{in}}(f, x_i) < s_i < s_{\text{out}}(x_{i+1})$. On

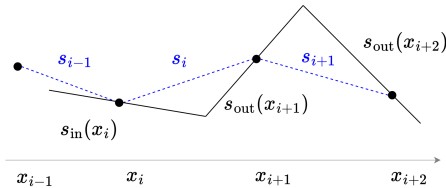

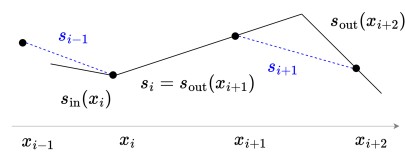

(a) A function with knots inside $(x_i, x_{i+1})$ and $(x_{i+1}, x_{i+2})$.

(b) A function which agrees with $\ell_i$ on $[x_i, x_{i+1}]$.

Figure 9: Behavior of $f \in S_p^*$ between data points of opposite curvature. The knot inside $(x_i, x_{i+1})$ on the left can be moved to $x_i$, and the knot inside $(x_{i+1}, x_{i+2})$ can be adjusted accordingly (right); this reduces the magnitudes of the slope changes of both knots.

$I := [x_{i-1} - \epsilon, x_{i+2} + \epsilon]$ for small $\epsilon > 0$, we thus have

$$V_p(f\big|_I) = |s_{\text{out}}(f, x_{i+1}) - s_{\text{in}}(f, x_i)|^p + |s_{\text{out}}(f, x_{i+2}) - s_{\text{out}}(f, x_{i+1})|^p \tag{29}$$

$$> |s_i - s_{\text{in}}(f, x_i)|^p + |s_{\text{out}}(f, x_{i+2}) - s_i|^p = V_p(g\big|_I) \tag{30}$$

where $g$ agrees with $f$ outside of $[x_i, x_{i+2}]$, agrees with $\ell_i$ on $[x_i, x_{i+1}]$, and has a single knot inside $[x_{i+1}, x_{i+2}]$ with $s_{\text{out}}(g, x_{i+1}) = s_i$ and $s_{\text{in}}(g, x_{i+2}) = s_{\text{out}}(g, x_{i+2}) = s_{\text{out}}(f, x_{i+2})$. (See Fig. 9b.) This contradicts $f \in S_p^*$.

For $p = 0$, consider some $f \in S_0^*$. If $\text{sgn}(s_i - s_{\text{in}}(f, x_i)) \neq \text{sgn}(s_{\text{out}}(f, x_{i+1}) - s_i)$, then by Lemma A.2, there is some $g \in S_0^*$ which agrees with $f$ outside of $[x_i, x_{i+1}]$ and agrees with $\ell_i$ on $[x_i, x_{i+1}]$. By Lemma A.1, if $\text{sgn}(s_i - s_{\text{in}}(f, x_i)) = \text{sgn}(s_{\text{out}}(f, x_{i+1}) - s_i) = 0$, then $f = \ell_i$ on $[x_i, x_{i+1}]$. If $\text{sgn}(s_i - s_{\text{in}}(f, x_i)) = \text{sgn}(s_{\text{out}}(f, x_{i+1}) - s_i)$ are both nonzero, then by Lemma A.1, $s_{\text{in}}(f, x_i) = s_{\text{out}}(f, x_i)$ and $s_{\text{in}}(f, x_{i+1}) = s_{\text{out}}(f, x_{i+1})$, and $f$ has a single knot inside $(x_i, x_{i+1})$ where the incoming line at $x_i$ and the outgoing line at $x_{i+1}$ meet. As before, it must be the case that $\text{sgn}(s_i - s_{i-1}) \neq \text{sgn}(s_i - s_{\text{in}}(f, x_i))$ and/or that $\text{sgn}(s_{i+1} - s_i) \neq \text{sgn}(s_{\text{out}}(f, x_{i+1}) - s_i)$. Assume without loss of generality that $\text{sgn}(s_{i+1} - s_i) \neq \text{sgn}(s_{\text{out}}(f, x_{i+1}) - s_i) = 1$, so that $s_{i+1} \leq s_i < s_{\text{out}}(f, x_{i+1}) = s_{\text{in}}(f, x_{i+1})$. Because $1 = \text{sgn}(s_{\text{out}}(f, x_{i+1}) - s_i) = \text{sgn}(s_i - s_{\text{in}}(f, x_i))$, we also have $s_{\text{in}}(f, x_i) < s_i < s_{\text{out}}(f, x_{i+1})$. If $\text{sgn}(s_{\text{out}}(f, x_{i+2}) - s_{i+1}) \neq \text{sgn}(s_{i+1} - s_{\text{in}}(f, x_{i+1})) = -1$, then by Lemma A.2, there is some $g \in S_0^*$ which agrees with $f$ outside $[x_{i+1}, x_{i+2}]$ and agrees with $\ell_{i+1}$ on $[x_{i+1}, x_{i+2}]$. Then this $g$ has $s_{\text{out}}(g, x_{i+1}) = s_i$ and $s_{\text{in}}(g, x_i) = s_{\text{in}}(f, x_i)$, so $\text{sgn}(s_{\text{out}}(x_{i+1}) - s_i) \in \{-1, 0\}$, and $\text{sgn}(s_i - s_{\text{in}}(g, x_i)) = 1$; hence by Lemma A.2, there is some $h \in S_0^*$ which agrees with $g$ outside of $[x_i, x_{i+1}]$ and agrees with $\ell_i$ on $[x_i, x_{i+1}]$. On the other hand, if $\text{sgn}(s_{\text{out}}(f, x_{i+2}) - s_{i+1}) = \text{sgn}(s_{i+1} - s_{\text{in}}(f, x_{i+1})) = -1$, then by Lemma A.1, $f$ has a single knot inside $(x_{i+1}, x_{i+2})$, and $s_{\text{in}}(f, x_{i+2}) = s_{\text{out}}(f, x_{i+2})$, as in Fig. 9a. This function has two knots on $I := [x_{i-1} - \epsilon, x_{i+2} + \epsilon]$ (for small $\epsilon > 0$). The function $g$ depicted in Fig. 9b, which agrees with $f$ outside of $[x_i, x_{i+2}]$, agrees with $\ell_i$ on $[x_i, x_{i+1}]$, and has a single knot inside $[x_{i+1}, x_{i+2}]$ with $s_{\text{out}}(g, x_{i+1}) = s_i$ and $s_{\text{in}}(g, x_{i+2}) = s_{\text{out}}(g, x_{i+2}) = s_{\text{out}}(f, x_{i+2})$, also has two knots on $I$. Therefore $g \in S_0^*$.

**Linearity between collinear data points.** For $0 < p < 1$, fix $f \in S_p^*$. If $s_{\text{in}}(f, x_i) = s_i = s_{i+1} = \cdots = s_{i+m-1} = s_{\text{out}}(f, x_{i+m})$, then $f$ must agree with $\ell_i = \cdots = \ell_{i+m-1}$ on $[x_i, x_{i+m}]$, since any other function $g$ would have $V_p(g\big|_I) > 0 = V_p(f\big|_I)$ on $I := [x_i - \epsilon, x_{i+m} + \epsilon]$ for small $\epsilon > 0$. If $\text{sgn}(s_i - s_{\text{in}}(f, x_i)) \neq \text{sgn}(s_{\text{out}}(f, x_{i+m}) - s_i)$, then the argument in the proof of Lemma A.2 shows that $f$ must agree with $\ell_i = \cdots = \ell_{i+m-1}$ on $[x_i, x_{i+m}]$. So we need only consider the case where $\text{sgn}(s_i - s_{\text{in}}(f, x_i)) = \text{sgn}(s_{\text{out}}(f, x_{i+m}) - s_i)$ are both nonzero; say without loss of generality that they both equal 1, so that $s_{\text{in}}(f, x_i) < s_i < s_{\text{out}}(f, x_{i+m})$. If $f = \ell_i$ on both $[x_i, x_{i+1}]$ and $[x_{i+m-1}, x_{i+m}]$, then it also must agree with $\ell_i$ on $[x_{i+1}, x_{i+m-1}]$ (otherwise it would have $V_p(f\big|_{[x_i, x_{i+m}]}) > 0$), so assume by contradiction that $f \neq \ell_i$ on at least one of these intervals, say without loss of generality on $[x_i, x_{i+1}]$. Then by Lemmas A.1 and A.2, it must be the case that $f$ has a single knot inside $(x_i, x_{i+1})$ and that $s_{\text{in}}(f, x_i) = s_{\text{out}}(f, x_i) < s_i < s_{\text{in}}(f, x_{i+1}) = s_{\text{out}}(f, x_{i+1})$. This implies that $f$ also disagrees with $\ell_i$ on $[x_i, x_{i+1}]$, so again by Lemmas A.1 and A.2, $f$ must have a single knot inside $(x_{i+1}, x_{i+2})$ with $s_{\text{in}}(f, x_{i+1}) = s_{\text{out}}(f, x_{i+1}) > s_{i+1} > s_{\text{in}}(f, x_{i+2}) = s_{\text{out}}(f, x_{i+2})$. The same logic applies on the remaining intervals up to and including $[x_{i+m-1}, x_{i+m}]$

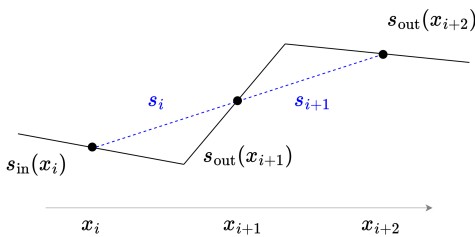

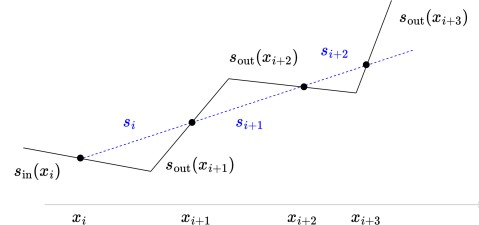

(a) A nonlinear function between $m + 1$ collinear points, $m$-even.

(b) A nonlinear function between $m + 1$ collinear points, $m$-odd.

Figure 10: Behavior of $f \in S_p^*$ between collinear points. If $f \in S_p^*$ is *not* a straight line between collinear points $(x_i, y_i), \ldots, (x_{i+m}, y_{i+m})$, it must look like Fig. 10a (if $m$ is even) or Fig. 10b (if $m$ is odd). In both cases, the sum of absolute slope changes of these functions is greater than the sum of absolute slope changes of the function $g$ which agrees with $f$ outside of $[x_i, x_{i+m}]$ and connects $(x_i, y_i), \ldots, (x_{i+m}, y_{i+m})$ with a straight line. Such a $g$ has two knots, whereas functions of the form $f$ depicted here have $m \geq 2$ knots.

(see Fig. 10). Note that if $m$ is even, we will have $s_{\text{in}}(f, x_{i+m-1}) = s_{\text{out}}(f, x_{i+m-1}) > s_{i+m-1} = s_i > s_{\text{in}}(f, x_{i+m}) = s_{\text{out}}(f, x_{i+m})$, contradicting the assumption that $\text{sgn}(s_{\text{out}}(f, x_{i+m}) - s_i) = 1$ (see Fig. 10a). If $m$ is odd, as in Fig. 10b, we have

$$V_p(f|_I) = |s_{\text{out}}(f, x_{i+1}) - s_{\text{in}}(f, x_i)|^p + |s_{\text{out}}(f, x_{i+2}) - s_{\text{out}}(f, x_{i+1})|^p \tag{31}$$

$$+ \cdots + |s_{\text{out}}(f, x_{i+m}) - s_{\text{out}}(f, x_{i+m-1})|^p \tag{32}$$

$$> |s_i - s_{\text{in}}(f, x_i)|^p + |s_{\text{out}}(f, x_{i+m}) - s_{i+m-1}|^p = V_p(g|_I) \tag{33}$$

where $g$ is the function which agrees with $f$ outside of $[x_i, x_{i+m}]$ and agrees with $\ell_i = \cdots = \ell_{i+m-1}$ on $[x_i, x_{i+m}]$; this contradicts $f \in S_p^*$.

In the case $p = 0$, fix $f \in S_0^*$. If $s_{\text{in}}(f, x_i) = s_i = \cdots = s_{i+m-1} = s_{\text{out}}(f, x_{i+m})$, then $f$ must agree with $\ell_i = \cdots = \ell_{i+m-1}$ on $[x_i, x_{i+m}]$ and if $\text{sgn}(s_i - s_{\text{in}}(f, x_i)) \neq \text{sgn}(s_{\text{out}}(f, x_{i+m}) - s_i)$, then the proof of Lemma A.2 shows that there is some $g \in S_0^*$ which agrees with $f$ outside of $[x_i, x_{i+m}]$ and agrees with $\ell_i$ on $[x_i, x_{i+m}]$. If $\text{sgn}(s_i - s_{\text{in}}(f, x_i)) = \text{sgn}(s_{\text{out}}(f, x_{i+m}) - s_i)$ are both nonzero, then there must be at least one knot on $[x_i, x_{i+m}]$ in order for the slope to change from $s_{\text{in}}(f, x_i)$ to $s_{\text{out}}(f, x_{i+m})$. It is impossible for $f$ to interpolate the data with a single knot on $[x_i, x_{i+m}]$ where the slope changes from $s_{\text{in}}(f, x_i)$ to $s_{\text{out}}(f, x_{i+m})$, since this would require at least two of the points $(x_i, y_i), \ldots, (x_{i+m}, y_{i+m})$ to both lie on either the incoming line at $x_i$ or the outgoing point at $x_{i+m}$, but this is impossible because $s_i \neq s_{\text{in}}(f, x_i)$ and $s_i \neq s_{\text{out}}(f, x_{i+m})$. Therefore, $f$ must have at least two knots on $[x_i, x_{i+m}]$. The function $g$ which agrees with $\ell_i$ on $[x_i, x_{i+m}]$ and has $s_{\text{in}}(g, x_i) = s_{\text{in}}(f, x_i)$ and $s_{\text{out}}(g, x_{i+m}) = s_{\text{out}}(f, x_{i+m})$ interpolates the points $(x_i, y_i), \ldots, (x_{i+m}, y_{i+m})$ with exactly two knots on $[x_i - \epsilon, x_{i+m} + \epsilon]$, and thus $g \in S_0^*$.

**Single knot between two data points with the same curvature.** For $0 < p < 1$, fix $f \in S_p^*$. If $i = 2$, then $f = \ell_1$ on $(-\infty, x_2]$ by Theorem 3.1,1. If $i > 2$, then by assumption, $\text{sgn}(s_{i-1} - s_{i-2}) \neq \text{sgn}(s_i - s_{i-1})$, so by Theorem 3.1,1, $f = \ell_{i-1}$ on $[x_{i-1}, x_i]$. In either case, we have $s_{\text{in}}(f, x_i) = s_{i-1}$. An analogous argument shows that $s_{\text{out}}(f, x_{i+1}) = s_{i+1}$. Similarly, Theorem 3.1,1 says that there is some $g \in S_0^*$ for which $s_{\text{in}}(g, x_i) = s_{i-1}$ and $s_{\text{out}}(g, x_{i+1}) = s_{i+1}$. In both cases, the conclusion then follows from Lemma A.2.

**Characterization around $\geq 2$ points with the same curvature.** For $0 < p < 1$, fix some $f \in S_p^*$. As in the proof of Theorem 3.1,2a above, the assumptions guarantee that $s_{i-1} = s_{\text{in}}(f, x_i)$ and $s_{i+m} = s_{\text{out}}(f, x_{i+m})$. Using this fact, we will proceed by (strong) induction, assuming without loss of generality that $\text{sgn}(s_i - s_{i-1}) = \text{sgn}(s_{i+1} - s_i) = \cdots = \text{sgn}(s_{i+m} - s_{i+m-1}) = 1$.

In the base case $m = 2$, first suppose that $\text{sgn}(s_i - s_{\text{in}}(f, x_i)) \neq \text{sgn}(s_{\text{out}}(f, x_{i+1}) - s_i)$. Since $s_{\text{in}}(f, x_i) = s_{i-1} < s_i$ by assumption, it must be the case that $\text{sgn}(s_{\text{out}}(f, x_{i+1}) - s_i) \in \{0, -1\}$. If $\text{sgn}(s_{\text{out}}(f, x_{i+1}) - s_i) = -1$, Lemma A.2 implies that $f = \ell_i$ on $[x_i, x_{i+1}]$, and thus $s_{\text{in}}(f, x_{i+1}) = s_i$. But then we have $s_{\text{in}}(f, x_{i+1}) = s_i < s_{i+1} < s_{\text{out}}(f, x_{i+2}) = s_{i+2}$, so by Lemma A.1, it must be the case that $s_{\text{in}}(f, x_{i+1}) = s_{\text{out}}(f, x_{i+1})$, contradicting $\text{sgn}(s_{\text{out}}(f, x_{i+1}) - s_i) = -1$ (see Fig. 12b).

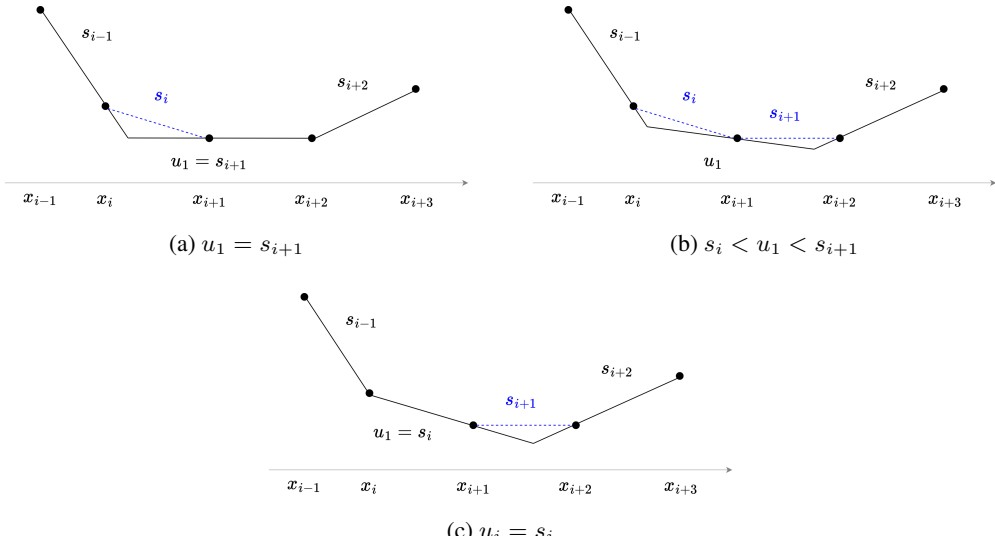

(a) $u_1 = s_{i+1}$

(b) $s_i < u_1 < s_{i+1}$

(c) $u_i = s_i$

Figure 11: Possible behavior of $f \in S_p^*$ between three consecutive data points of the same discrete curvature. All possibilities satisfy $s_i \leq u_1 \leq s_{i+1}$.

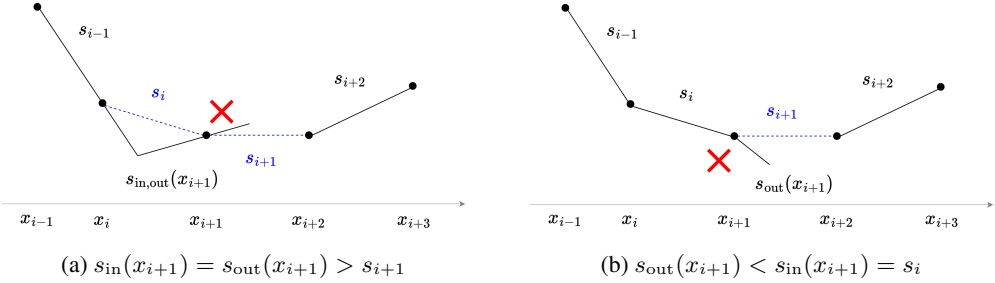

(a) $s_{\text{in}}(x_{i+1}) = s_{\text{out}}(x_{i+1}) > s_{i+1}$

(b) $s_{\text{out}}(x_{i+1}) < s_{\text{in}}(x_{i+1}) = s_i$

Figure 12: Behaviors which $f \in S_p^*$ for $0 < p < 1$ *cannot* exhibit around three consecutive points of the same discrete curvature. The case on the left violates Lemma A.2, and the case on the right violates Lemma A.1.

If $\text{sgn}(s_{\text{out}}(f, x_{i+1}) - s_i) = 0$, then Lemma A.2 implies that $f = \ell_i$ on $[x_i, x_{i+1}]$, and therefore $s_{\text{in}}(f, x_{i+1}) = s_{\text{out}}(f, x_{i+1}) = s_i$. Then $s_{\text{in}}(f, x_{i+1}) = s_i < s_{i+1} < s_{\text{out}}(f, x_{i+2}) = s_{i+2}$, so by Lemma A.1, $f$ has a single knot inside $[x_{i+1}, x_{i+2}]$, with $s_{\text{in}}(f, x_{i+1}) = s_{\text{out}}(f, x_{i+1}) = s_i$ (as we already know) and $s_{\text{in}}(f, x_{i+2}) = s_{\text{out}}(f, x_{i+2}) = s_{i+2}$. The conclusion then holds with $u_1 := s_i$ (see Fig. 11c).

On the other hand, still for the base case $m = 2$, suppose that $\text{sgn}(s_i - s_{\text{in}}(f, x_i)) = \text{sgn}(s_{\text{out}}(f, x_{i+1}) - s_i)$. Then by Lemma A.1, there is a single knot inside $[x_i, x_{i+1}]$, with $s_{i-1} = s_{\text{in}}(f, x_i) = s_{\text{out}}(f, x_i)$ and $s_{\text{in}}(f, x_{i+1}) = s_{\text{out}}(f, x_{i+1})$. It cannot be the case that $s_{\text{out}}(f, x_{i+1}) > s_{i+1}$, because if this were true, we would have $-1 = \text{sgn}(s_{i+1} - s_{\text{in}}(f, x_{i+1})) \neq \text{sgn}(s_{\text{out}}(f, x_{i+2}) - s_{i+1}) = 1$, and that would imply by Lemma A.2 that $f = \ell_{i+1}$ on $[x_{i+1}, x_{i+2}]$, contradicting $s_{\text{out}}(f, x_{i+1}) > s_{i+1}$ (see Fig. 12a). Therefore, we must have $s_{\text{out}}(f, x_{i+1}) \leq s_{i+1}$. If $s_{\text{out}}(f, x_{i+1}) < s_{i+1}$, then by Lemma A.1, there is a single knot on $[x_{i+1}, x_{i+2}]$, with $s_{\text{in}}(f, x_{i+1}) = s_{\text{out}}(f, x_{i+1})$ (as we already knew) and $s_{\text{in}}(f, x_{i+2}) = s_{\text{out}}(f, x_{i+2}) = s_{i+2}$. The conclusion then holds with $u_1 := s_{\text{in}}(f, x_{i+1}) = s_{\text{out}}(f, x_{i+1})$ (see Fig. 11b). If $s_{\text{out}}(f, x_{i+1}) = s_{i+1}$, then $0 = \text{sgn}(s_{i+1} - s_{\text{in}}(f, x_{i+1})) \neq \text{sgn}(s_{\text{out}}(f, x_{i+2}) - s_{i+1}) = 1$, so by Lemma A.2, $f = \ell_{i+1}$ on $[x_{i+1}, x_{i+2}]$. The conclusion then holds with $u_1 := s_{i+1}$ (see Fig. 11a).

Next, for the (strong) inductive step, fix some integer $m \geq 4$ and assume the conclusion holds for all integers $2, \ldots, m - 1$. First suppose that $s_{\text{out}}(f, x_{i+m-1}) > s_{i+m-2}$. Then by the inductive hypothesis, $f$ has slopes $u_1, \ldots, u_{m-2}$—some of which may be equal to each other, but all of which are distinct from $s_{\text{in}}(f, x_i) = s_{i-1}$ and $s_{\text{out}}(f, x_{i+m-1})$—on $[x_i, x_{i+m-1}]$ satisfying $s_{i+j-1} \leq u_j \leq s_{i+j}$ for all $j = 1, \ldots, m - 2$. It cannot be the case that $s_{\text{out}}(f, x_{i+m-1}) > s_{i+m-1}$, because

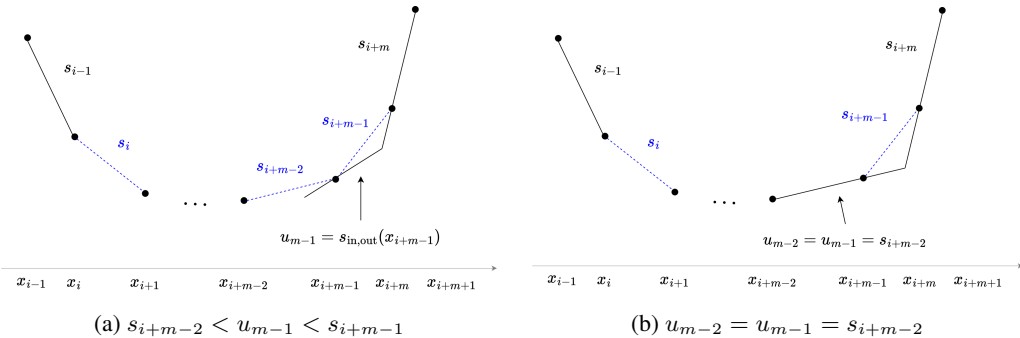

(a) $s_{i+m-2} < u_{m-1} < s_{i+m-1}$         (b) $u_{m-2} = u_{m-1} = s_{i+m-2}$

Figure 13: Possible behavior of $f \in S_p^*$ around $m$ consecutive data points of the same discrete curvature. Assuming inductively that Theorem 3.1,2b holds for $2, \ldots, m-1$, both satisfy $s_{i+j-1} \leq u_j \leq s_{i+j}$ for $j = 1, \ldots, m-1$.

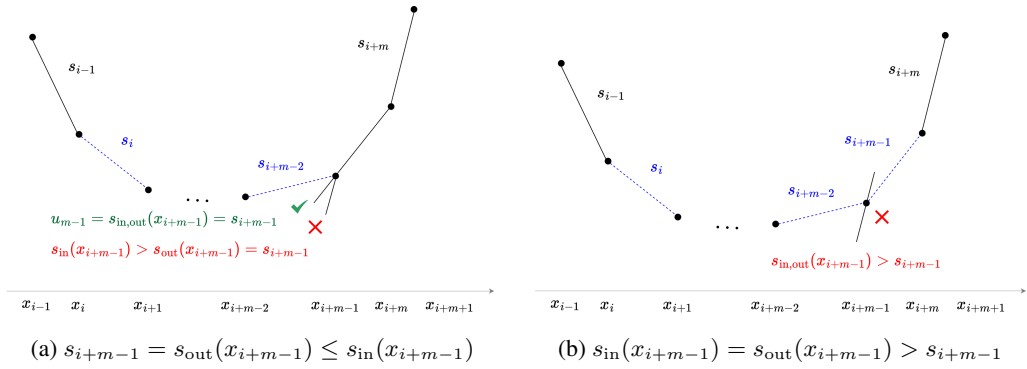

(a) $s_{i+m-1} = s_{\text{out}}(x_{i+m-1}) \leq s_{\text{in}}(x_{i+m-1})$     (b) $s_{\text{in}}(x_{i+m-1}) = s_{\text{out}}(x_{i+m-1}) > s_{i+m-1}$

Figure 14: Behaviors which $f \in S_p^*$ can and cannot exhibit between $m$ consecutive points of the same discrete curvature. Assuming inductively that Theorem 3.1,2b holds for $2, \ldots, m-1$, the case with the green check mark on the left satisfies $s_{i+j-1} \leq u_j \leq s_{i+j}$ for $j = 1, \ldots, m-1$. The case with the red $x$ on the left violates Lemma A.1, and the case on the right violates Lemma A.2.

if this were true, we would have $-1 = \text{sgn}(s_{i+m-1} - s_{\text{in}}(f, x_{i+m-1})) \neq \text{sgn}(s_{\text{out}}(f, x_{i+m}) - s_{i+m-1}) = 1$, and thus Lemma A.2 would imply that $f = \ell_{i+m-1}$ on $[x_{i+m-1}, x_{i+m}]$, contradicting $s_{\text{out}}(f, x_{i+m-1}) > s_{i+m-1}$ (see Fig. 12b). Therefore, we must have $s_{\text{out}}(f, x_{i+m-1}) \leq s_{i+m-1}$. If $s_{\text{out}}(f, x_{i+m-1}) < s_{i+m-1}$, then by Lemma A.1, there is a single knot inside $[x_{i+m-1}, x_{i+m}]$ and $s_{\text{in}}(f, x_{i+m-1}) = s_{\text{out}}(f, x_{i+m-1})$ and $s_{\text{in}}(f, x_{i+m}) = s_{\text{out}}(f, x_{i+m}) = s_{i+m}$. The conclusion then holds for $m$ with $u_{m-1} := s_{\text{out}}(f, x_{i+m-1})$ (see Fig. 13a). If $s_{\text{out}}(f, x_{i+m-1}) = s_{i+m-1}$, then by Lemmas A.1 and A.2, it must be the case that $\{0, -1\} \ni \text{sgn}(s_{i+m-1} - s_{\text{in}}(f, x_{i+m-1})) \neq \text{sgn}(s_{\text{out}}(f, x_{i+m}) - s_{i+m-1}) = 1$. It is impossible that $\text{sgn}(s_{i+m-1} - s_{\text{in}}(f, x_{i+m-1})) = -1$ because by Lemmas A.1 and A.2, for $f$ to disagree with $\ell_{i+m-2}$ on $[x_{i+m-2}, x_{i+m-1}]$, it must be the case that $s_{\text{in}}(f, x_{i+m-1}) = s_{\text{out}}(f, x_{i+m-1})$, contradicting $s_{\text{in}}(f, x_{i+m-1}) < s_{\text{out}}(f, x_{i+m-1}) = s_{i+m-1}$ (see Fig. 14a, red). Therefore, in this case we have $s_{\text{in}}(f, x_{i+m-1}) = s_{\text{out}}(f, x_{i+m-1}) = s_{i+m-1}$, and the conclusion holds for $m$ with $u_{m-1} := s_{i+m-1}$ (see Fig. 14a, green).

On the other hand, still for the (strong) inductive step, suppose that $s_{\text{out}}(f, x_{i+m-1}) \leq s_{i+m-2}$. If $s_{\text{out}}(f, x_{i+m-1}) = s_{i+m-2}$, then by Lemmas A.1 and A.2, $f$ has a single knot inside $[x_{i+m-1}, x_{i+m}]$ with $s_{\text{in}}(f, x_{i+m-1}) = s_{\text{out}}(f, x_{i+m-1}) = s_{i+m-2}$ and $s_{\text{in}}(f, x_{i+m}) = s_{\text{out}}(f, x_{i+m}) = s_{i+m}$. This implies, again by Lemmas A.1 and A.2, that $f = \ell_{i+m-2}$ on $[x_{i+m-2}, x_{i+m-1}]$. By the (strong) inductive hypothesis, $f$ has slopes $u_1, \ldots, u_{m-3}$ on $[x_i, x_{i+m-2}]$, all distinct from $s_{\text{in}}(f, x_i) = s_{i-1}$ and $s_{\text{out}}(f, x_{i+m-2}) = s_{i+m-2}$, which satisfy $s_{i+j-1} \leq u_j \leq s_{i+j}$ for $j = 2, \ldots, m-3$. The conclusion then holds for $m$ with $u_{m-2} = u_{m-1} := s_{i+m-2}$ (see Fig. 13b). It remains only to consider the case $s_{\text{out}}(f, x_{i+m-1}) < s_{i+m-2}$, and show that this is impossible for $f \in S_p^*$. If $s_{\text{out}}(f, x_{i+m-1}) < s_{i+m-2}$, then by Lemmas A.1 and A.2, there is a single knot inside $[x_{i+m-1} - x_{i+m}]$ and $s_{\text{in}}(f, x_{i+m-1}) = s_{\text{out}}(f, x_{i+m-1})$ and $s_{\text{in}}(f, x_{i+m}) = s_{\text{out}}(f, x_{i+m}) = s_{i+m}$. This in turn implies, again by Lemmas A.1 and A.2, that there is a single knot inside $[x_{i+m-2}, x_{i+m-1}]$ and

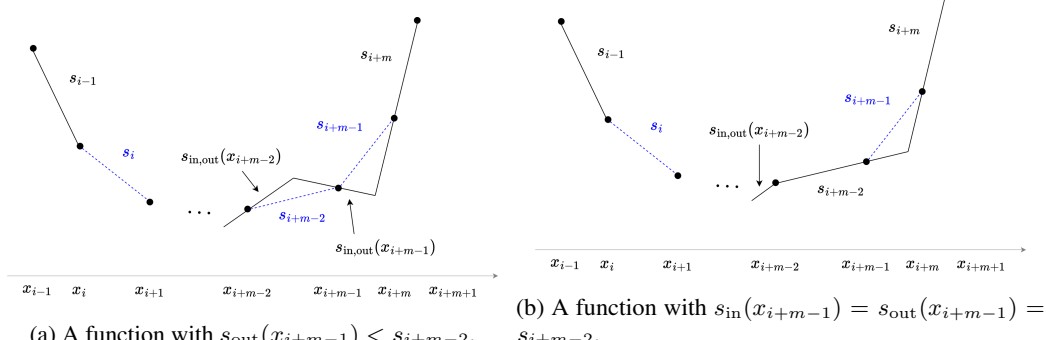

(a) A function with $s_{\text{out}}(x_{i+m-1}) < s_{i+m-2}$.

(b) A function with $s_{\text{in}}(x_{i+m-1}) = s_{\text{out}}(x_{i+m-1}) = s_{i+m-2}$.

Figure 15: Possible behavior of $f \in S_p^*$ around $m$ consecutive slope changes of the same discrete curvature. The magnitude of slope change at each knot of the function $f$ on the left, which has $s_{\text{out}}(f, x_{i+m-1}) < s_{i+m-2}$, is greater than that of the corresponding knot in the function $g$ on the right, which has $s_{\text{in}}(x_{i+m-1}) = s_{\text{out}}(x_{i+m-1}) = s_{i+m-2}$.

$s_{\text{in}}(f, x_{i+m-2}) = s_{\text{out}}(f, x_{i+m-2})$. (See Fig. 15a.) On the interval $I := [x_{i+m-2} - \epsilon, x_{i+m} + \epsilon]$ for small $\epsilon > 0$, we thus have

$$V_p(f\big|_I) = |s_{\text{out}}(f, x_{i+m-1}) - s_{\text{in}}(f, x_{i+m-2})|^p + |s_{i+m} - s_{\text{out}}(f, x_{i+m-1})|^p \tag{34}$$

$$> |s_{i+m-2} - s_{\text{in}}(f, x_{i+m-2})|^p + |s_{i+m} - s_{i+m-2}|^p \tag{35}$$

where the inequality holds because $s_{\text{out}}(f, x_{i+m-1}) < s_{i+m-2} < s_{i+m}$ and $s_{\text{in}}(f, x_{i+m-2}) > s_{i+m-2} > s_{\text{out}}(f, x_{i+m-1})$. The latter is exactly $V_p(g\big|_I)$, where $g$ is the function which agrees with $f$ outside of $[x_{i+m-2}, x_{i+m}]$, agrees with $\ell_{i+m-2}$ on $[x_{i+m-2}, x_{i+m-1}]$, and has a single knot in $[x_{i+m-1}, x_{i+m}]$ with $s_{\text{in}}(f, x_{i+m-1}) = s_{\text{out}}(f, x_{i+m-1}) = s_{i+m-2}$ and $s_{\text{in}}(f, x_{i+m}) = s_{\text{out}}(f, x_{i+m}) = s_{i+m}$. (See Fig. 15b.) This contradicts $f \in S_p^*$.

For the case $p = 0$: again, as in the proof of Theorem 3.1,2a, the assumptions guarantee that there is some $f \in S_0^*$ for which $s_{\text{in}}(f, x_i) = s_{i-1}$ and $s_{\text{out}}(f, x_{i+1}) = s_{i+1}$. The inductive argument above for $0 < p < 1$ also shows the desired result in the $p = 0$ case, with each reference to Lemma A.2 as well as the last portion of the inductive step instead justifying the existence of some $g \in S_0^*$ which exhibits the desired local behavior and agrees with $f$ elsewhere.

**Non-emptiness of $S_p^*$ for $0 < p < 1$.** As noted in Appendix A.1.1, restricting the input weights to $|w_k| = 1$ in optimization (2) recovers the same set of optimal functions $S_p^*$. The geometric characterization proved above shows that any solution to this modified (2) must have no knots outside of $[x_2, x_{N-1}]$, and thus its biases satisfy $|b_k| \leq B := \max\{|x_2|, |x_{N-1}|\}$. Additionally, any such solution has slopes absolutely bounded by $C := \max_{i=1,\dots,N-1} |s_i|$, so that each $|v_k w_k| = |v_k| \leq 2C$, and thus its skip connection parameters can be bounded as

$$|a| - \left|\sum_{w_k > 0} v_k\right| \leq \left|a + \sum_{w_k > 0} v_k\right| = |f'(x_N + 1)| \leq C \implies |a| \leq A := C + \sum_{w_k > 0} |v| \leq C + 2KC \tag{36}$$

and

$$c = y_1 - \sum_{k=1}^K v_k(w_k x_1 - b_k)_+ - a x_1 \implies |c| \leq |y_1| + \sum_{k=1}^K |v_k|(|x_1| + |b_k|) + |a x_1| \tag{37}$$

$$\leq C_0 := |y_1| + 2KC(|x_1| + B) + |x_1|(C + 2KC) \tag{38}$$

Therefore, any $f \in S_p^*$ is recovered by a restricted version of (2) which requires that $|w_k| = 1, |b_k| \leq B, |v_k| \leq 2C, |a| \leq A, |c| \leq C_0$. For any fixed choice of $w_1, \dots, w_K \in \{-1, 1\}^K$, this modified optimization (in the remaining variables) constitutes a minimization of a continuous function over a compact set, so by the Weierstrass extreme value theorem, a solution exists. Taking the minimum over all such solutions for all possible choices of $w_1, \dots, w_K \in \{-1, 1\}^K$ proves the result. $\qquad\square$

### A.1.4 Proof of Theorem 3.2

*Proof.* If the data contain no more than two consecutive points with the same discrete curvature, there is only one interpolant $f$ which fits the description in Theorem 3.1. By Theorem 4 in Debarre et al. (2022), this $f \in S_0^*$. Otherwise, if the data do contain some $x_i, \ldots, x_{i+m}$ with the same discrete curvature for $m \geq 2$, the slopes $u_1, \ldots, u_{m-1}$ of any interpolant satisfying the description in Theorem 3.1,2b have $s_{i+j-1} \leq u_j \leq s_{i+j}$ for each $j = 1, \ldots, m-1$. Indeed, any choice of $u_1, \ldots, u_{m-1}$ satisfying $s_{i+j-1} \leq u_j \leq s_{i+j}$ for each $j$ defines an CPWL interpolant of the data, given by the pointwise maximum of $\ell_{i-1}, \ell_{i+m}$, and the lines $L_j$, each of which has slope $u_j$ and passes through $(x_{i+j}, y_{i+j})$. Therefore, the set $S$ of functions described by Theorem 3.1,2b on any such $x_i, \ldots, x_{i+m}$ can be fully associated with the set of numbers $u_1, \ldots, u_m$ satisfying $s_{i+j-1} \leq u_j \leq s_{i+j}$ for each $j$. Since any such $u_j = (1 - \alpha_j)s_{i+j-1} + \alpha_j s_{i+j}$ for a unique $\alpha_j \in [0, 1]$, we can equivalently identify $S$ with the unit cube $[0, 1]^{m-1}$.

Viewed as a function of its corresponding $\boldsymbol{\alpha} = [\alpha_1, \ldots, \alpha_{m-1}]^\top \in [0, 1]^{m-1}$, the regularization cost $V_p(f|_I)$ (for $0 < p < 1$) of any $f \in S$ on $I := [x_{i-1} - \delta, x_{i+m+1} + \delta]$ for small $\delta > 0$ is

$$V_p(\boldsymbol{\alpha}) = |u_1 - s_{i-1}|^p + \sum_{j=2}^{m-1} |u_j - u_{j-1}|^p + |s_{i+m} - u_{m-1}|^p = \|\boldsymbol{A}\boldsymbol{\alpha} + \boldsymbol{c}\|_p^p \qquad (39)$$

where the rows $\boldsymbol{a}_1, \ldots, \boldsymbol{a}_m$ of $\boldsymbol{A} \in \mathbb{R}^{m \times (m-1)}$ and entries $c_1, \ldots, c_m$ of $\boldsymbol{c} \in \mathbb{R}^m$ are

$$\boldsymbol{a}_1 = [s_{i+1} - s_i, 0, \ldots, 0]^\top, \qquad c_1 = s_i - s_{i-1} \qquad (40)$$

$$\boldsymbol{a}_m = [0, \ldots, 0, s_{i+m-1} - s_{i+m}]^\top, \qquad c_1 = s_{i+m} - s_{i+m-1} \qquad (41)$$

and

$$\boldsymbol{a}_j = [0, \ldots, 0, -(s_{i+j-1} - s_{i+j-2}), s_{i+j} - s_{i+j-1}, 0, \ldots, 0]^\top, \qquad c_j = s_{i+j-1} - s_{i+j-2} \quad (42)$$

for $j = 2, \ldots, m-1$, with the nonzero entries of $\boldsymbol{a}_j$ in positions $j-1$ and $j$. By the assumption that $\epsilon_i = \cdots = \epsilon_{i+m}$ are all nonzero, the rows $\boldsymbol{a}_1, \ldots, \boldsymbol{a}_m$ of $\boldsymbol{A}$ span $\mathbb{R}^{m-1}$, and thus $\boldsymbol{\alpha} \mapsto \boldsymbol{A}\boldsymbol{\alpha} + \boldsymbol{c}$ is injective. For any distinct $\boldsymbol{\alpha}_1, \boldsymbol{\alpha}_2 \in [0, 1]^{m-1}$, we thus have $\boldsymbol{A}\boldsymbol{\alpha}_1 + \boldsymbol{c} \neq \boldsymbol{A}\boldsymbol{\alpha}_2 + \boldsymbol{c}$, and therefore

$$V_p(t\boldsymbol{\alpha}_1 + (1-t)\boldsymbol{\alpha}_2) = \|t(\boldsymbol{A}\boldsymbol{\alpha}_1 + \boldsymbol{c}) + (1-t)(\boldsymbol{A}\boldsymbol{\alpha}_2 + \boldsymbol{c})\|_p^p > t\|\boldsymbol{A}\boldsymbol{\alpha}_1 + \boldsymbol{c}\|_p^p + (1-t)\|\boldsymbol{A}\boldsymbol{\alpha}_2 + \boldsymbol{c}\|_p^p \tag{43}$$

for any $t \in (0, 1)$ by strict concavity of $\|\cdot\|_p^p$ on $[0, 1]^{m-1}$. This shows that $V_p$ is strictly concave on $[0, 1]^{m-1}$. By the Bauer maximum principle (Aliprantis and Border (2006), Theorem 4.104), $V_p(\boldsymbol{\alpha})$ thus attains a minimum on $[0, 1]^{m-1}$ at an extreme point of $[0, 1]^{m-1}$. Moreover, by strict concavity of $V_p(\boldsymbol{\alpha})$, *any* minimum of $V_p(\boldsymbol{\alpha})$ over $[0, 1]^{m-1}$ must occur at an extreme point. Therefore, when searching for an $f \in S$ with minimal $V_p$, we may restrict our attention to those $f$ corresponding to the $2^{m-1}$ vertices $\{0, 1\}^{m-1}$ of the cube $[0, 1]^{m-1}$.

Among these $2^{m-1}$ vertices, there is at least one corresponding to a sparsest solution $f \in S_0^* \cap S$. This is because, by Theorem 4 in Debarre et al. (2022), any $f \in S_0^* \cap S$ has $\lceil \frac{m+1}{2} \rceil$ knots on $I$, and there is one such $f$ if $m$ is odd, or uncountably many if $m$ is even. If $m$ is odd, this unique $f$ corresponds to the vertex $[1, 0, \ldots, 1, 0]^\top \in \{0, 1\}^{m-1}$; i.e., this $f$ has $u_j = s_{i+j}$ for odd $j$ and $u_j = s_{i+j-1}$ for even $j$. If $m$ is even, there are multiple vertices $\boldsymbol{\alpha} \in \{0, 1\}^{m-1}$ which attain the minimal number $\lceil \frac{m+1}{2} \rceil$ of knots on $I$: two examples are $[1, 0, \ldots, 1, 0, 1]^\top \in \{0, 1\}^{m-1}$ (see Fig. 16b) and $[0, 1, \ldots, 0, 1, 0]^\top \in \{0, 1\}^{m-1}$ (see Fig. 16a).

For each of the $2^{m-1}$ functions $f \in S$ corresponding to the vertices $\boldsymbol{\alpha} \in \{0, 1\}^{m-1}$, consider the associated "cost curves" $C_f(p) := V_p(f|_I)$, which is simply the regularization cost $V_p(f|_I)$ for that individual $f$ over $I$, viewed as a function of the variable $p \in [0, 1]$. Each $C_f(p)$ is a *generalized Dirichlet polynomial*[8] of the variable $p$. By the generalized Descartes rule of signs for Dirichlet polynomials (Jameson (2006), Theorem 3.1), any two cost curves $C_f(p), C_g(p)$ for distinct $f, g$ can only intersect at finitely many $p \in [0, 1]$. Therefore, for any given $p \in [0, 1]$ outside of that finite set (which has Lebesgue measure zero), a unique one of these $2^{m-1}$ candidate solutions $f$ has smaller

---

[8]*Generalized Dirichlet polynomials* are functions of the form $f(x) = \sum_{i=1}^n a_i b_i^x$, where $a_i, x \in \mathbb{R}$ and $b_1 \geq \cdots \geq b_n > 0$.

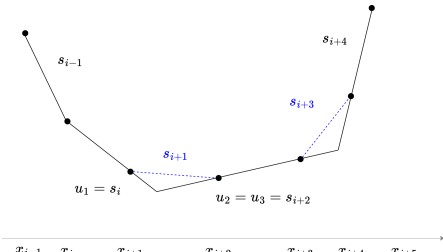 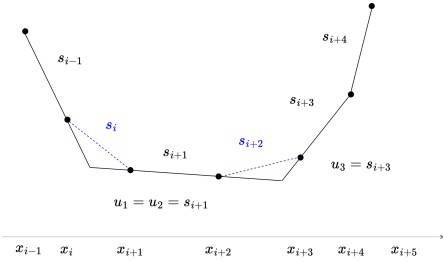

(a) One sparsest interpolant, corresponding to $\boldsymbol{\alpha} = [0, 1, 0]$.

(b) Another sparsest interpolant, corresponding to $\boldsymbol{\alpha} = [1, 0, 1]$.

Figure 16: Illustration of two sparsest interpolants in the scenario of Theorem 3.1,2b with $m = 4$. Both have $\lceil \frac{m+1}{2} \rceil = 3$ knots on $[x_i - 1, x_{i+m} + 1]$, consistent with Theorem 4 of Debarre et al. (2022).

cost $C_f(p) = V_p(f|_I)$ than the others. Furthermore, the sparsest of these $2^{m-1}$ functions (i.e., the ones in $S \cap S_0^*$) will necessarily have smaller $C_f(0) = V_0(f|_I)$ than the rest, and because all of the cost curves $C_f(p)$ are continuous, a unique one of these sparsest solutions will have smaller cost $C_f(p)$ than the others for all $p$ between $0$ and $p^*$, which is the location of the first intersection of any two of these $2^{m-1}$ candidate solutions' cost curves. $\square$

## A.2 Multivariate results

### A.2.1 Proof of Proposition 4.1

*Proof.* We address the statements individually.

**Existence of solutions to** (8) **and** (9)**.** Existence of solutions to (9) is a simple consequence of the fact that interpolation is possible whenever $K \geq N$, so the feasible set of (9) is non-empty, and objective values of (9) lie in $\{1, \ldots, K(d+1)\}$ on which a minimum is necessarily achieved.

To show existence of solutions to (8), recall that by homogeneity of the ReLU, we can rescale the input and output weights of any neural network as $\boldsymbol{w}_k \mapsto \alpha_k \boldsymbol{w}_k$ and $v_k \mapsto \alpha_k^{-1} v_k$ for any $\alpha_k > 0$ without changing the network's represented function or its $\ell^p$ or $\ell^0$ path norms. Therefore, the optimal value of (8) is equal to that of

$$\arg\min_{\boldsymbol{\theta}} \sum_{k=1}^{K} |v_k|^p , \text{ subject to } f_{\boldsymbol{\theta}}(\boldsymbol{x}_i) = y_i, \ i = 1, \ldots, N, \ \|\boldsymbol{w}_k\|_p = 1, \ k = 1, \ldots, K \quad (44)$$

Picking an arbitrary feasible $\boldsymbol{\theta}'$ for (44), any solution to (44) must have

$$\max_{k=1,\ldots,K} |v_k|^p \leq \sum_{k=1}^{K} |v_k|^p \leq \sum_{k=1}^{K} |v_k'|^p \implies \max_{k=1,\ldots,K} |v_k| \leq C := \left( \sum_{k=1}^{K} |v_k'|^p \right)^{1/p} \quad (45)$$

so we can further recast (44) as

$$\arg\min_{\boldsymbol{\theta}} \sum_{k=1}^{K} |v_k|^p , \text{ subject to } f_{\boldsymbol{\theta}}(\boldsymbol{x}_i) = y_i, \ i = 1, \ldots, N, \ \|\boldsymbol{w}_k\|_p = 1, \ |v_k| \leq C, \ k = 1, \ldots, K$$

$$(46)$$

Because the sets $\{\boldsymbol{w} : \|\boldsymbol{w}\|_p = 1\}$ and $\{v : |v| \leq C\}$ are compact, so is their Cartesian product $\{\boldsymbol{\theta} : \|\boldsymbol{w}_k\|_p = 1, |v_k| \leq C, k = 1, \ldots, K\}$. Since each $\boldsymbol{\theta} \mapsto f_{\boldsymbol{\theta}}(\boldsymbol{x}_i)$ is continuous, the preimage of the singleton sets $\{y_i\}$ under those maps are closed, and so is their finite intersection. As the intersection of a closed set with a compact set, the feasible set of (46) is compact. Problem (46) is therefore a minimization of a continuous function over a compact set, so it attains a solution by the Weierstrass extreme value theorem.

**Solutions to** (8) **and** (9) **have no more than** $N$ **active neurons.** Assume by contradiction that a solution $\{v_k, \boldsymbol{w}_k\}_{k=1}^{K}$ to (8) for $K > N$ has $K_0 > N$ active neurons $\{v_k, \boldsymbol{w}_k\}_{k=1}^{K_0}$. Because $K_0 > N$, the vectors $\boldsymbol{a}_k := [(\boldsymbol{w}_k^\top \overline{\boldsymbol{x}}_1)_+, \ldots, (\boldsymbol{w}_k^\top \overline{\boldsymbol{x}}_N)_+]^\top$, where $\overline{\boldsymbol{x}}_i := [\boldsymbol{x}_i, 1]$, are linearly dependent, meaning that there are constants $c_1, \ldots, c_{K_0}$ (not all zero) for which $\sum_{k=1}^{K_0} c_k \boldsymbol{a}_k = \boldsymbol{0}$. Then for any real $t$:

$$\sum_{k=1}^{K_0} (v_k + tc_k) \boldsymbol{a}_k = \sum_{k=1}^{K_0} v_k \boldsymbol{a}_k + t \sum_{k=1}^{K_0} c_k \boldsymbol{a}_k = \sum_{k=1}^{K_0} v_k \boldsymbol{a}_k = \boldsymbol{y}$$

where $\boldsymbol{y} := [y_1, \ldots, y_N]^\top$. In other words, the network with parameters $\{v_k + tc_k, \boldsymbol{w}_k\}_{k=1}^{K_0}$ interpolates the data, for any real $t$.

In the case of (8) for any $0 < p < 1$, choose $t > 0$ small enough that $\mathrm{sgn}(v_k + tc_k) = \mathrm{sgn}(v_k)$ for each $k$, and thus $\mathrm{sgn}((v_k + tc_k)w_{k,i}) = \mathrm{sgn}(v_k w_{k,i})$ for each $k, i$. Then by strict concavity of $t \mapsto |t|^p$ on $t \in (-\infty, 0)$ and $t \in (0, \infty)$ we have

$$|v_k w_{k,i}|^p = \left| \frac{(v_k + tc_k)w_{k,i} + (v_k - tc_k)w_{k,i}}{2} \right|^p > \frac{|(v_k + tc_k)w_{k,i}|^p + |(v_k - tc_k)w_{k,i}|^p}{2} \quad (47)$$

for each $i$ and each $k$ with $c_k \neq 0$ (if $c_k = 0$ the above holds with equality). Since $c_k \neq 0$ for at least one $k$, this implies that

$$\sum_{k=1}^{K_0} \|v_k \boldsymbol{w}_k\|_p^p > \frac{1}{2} \left( \sum_{k=1}^{K_0} \|(v_k + tc_k)\boldsymbol{w}_k\|_p^p + \sum_{k=1}^{K_0} \|(v_k - tc_k)\boldsymbol{w}_k\|_p^p \right) \quad (48)$$

but then at least one of $\sum_{k=1}^{K_0} \|(v_k + tc_k)\boldsymbol{w}_k\|_p^p$ or $\sum_{k=1}^{K_0} \|(v_k - tc_k)\boldsymbol{w}_k\|_p^p$ must be strictly less than $\sum_{k=1}^{K_0} \|v_k \boldsymbol{w}_k\|_p^p$. This contradicts optimality of $\{v_k, \boldsymbol{w}_k\}_{k=1}^{K}$.

In the case of (9), choose $t = -v_{k'}/c_{k'}$ for one of the $c_{k'} \neq 0$. We then have

$$\sum_{k=1}^{K_0} \|(v_k + tc_k)\boldsymbol{w}_k\|_0 < \sum_{k=1}^{K_0} \|v_k \boldsymbol{w}_k\|_0 \quad (49)$$

strictly, because all of the $v_k \boldsymbol{w}_k$ are nonzero, whereas at least one of the $(v_k + tc_k)\boldsymbol{w}_k$ on the left is zero (for $k = k'$), and $\|v_k \boldsymbol{w}_k\|_0 = \|(v_k + tc_k)\boldsymbol{w}_k\|_0$ whenever both $v_k \boldsymbol{w}_k$ and $(v_k + tc_k)\boldsymbol{w}_k$ are nonzero. This again contradicts optimality of $\{v_k, \boldsymbol{w}_k\}_{k=1}^{K}$.

**Sparsity bound on solutions to** (9)**.** If the data are in general position and $N \geq d+1$, then Bubeck et al. (2020) show that there exists an interpolating single-hidden-layer ReLU network with $4\lceil N/d \rceil$ neurons. Any such network clearly has at most $4(d+1)\lceil N/d \rceil \leq 4(N+1) + 4(N+1)/d \leq 8(N+1) = O(N)$ nonzero input weight/bias parameters across those $4\lceil N/d \rceil$ neurons.

If the data are in general position and $N \leq d+1$, the points $\boldsymbol{x}_1, \ldots, \boldsymbol{x}_N$ must be affinely independent, meaning that

$$\sum_{i=1}^{N} \alpha_i \boldsymbol{x}_i = \boldsymbol{0} \text{ and } \sum_{i=1}^{N} \alpha_i = 0 \implies \alpha_1 = \cdots = \alpha_N = 0 \quad (50)$$

Because this condition is equivalent to linear independence of the vectors $\overline{\boldsymbol{x}}_i := [\boldsymbol{x}_i^\top, 1]^\top$, the general position assumption ensures that augmented data matrix $\overline{\boldsymbol{X}} = [\overline{\boldsymbol{x}}_1, \ldots, \overline{\boldsymbol{x}}_N]^\top \in \mathbb{R}^{N \times (d+1)}$ has full rank $N$. Therefore, there exists a solution $\boldsymbol{w} \in \mathbb{R}^{d+1}$ to the system

$$\overline{\boldsymbol{X}} \boldsymbol{w} = \boldsymbol{y} := [y_1, \ldots, y_N]^\top \quad (51)$$

with $\|\boldsymbol{w}\|_0 = N$. (To see this, choose $N$ linearly independent columns of $\overline{\boldsymbol{X}}$, express $\boldsymbol{y}$ as a linear combination with respect to this basis, and let $\boldsymbol{w}$ be the vector of coefficients of this linear combination.) For any such $\boldsymbol{w}$, (51) says that the affine function

$$f(\boldsymbol{x}) = \boldsymbol{w}^\top \overline{\boldsymbol{x}} = (\boldsymbol{w}^\top \overline{\boldsymbol{x}})_+ - (-\boldsymbol{w}^\top \overline{\boldsymbol{x}})_+ \quad (52)$$

interpolates the data (recall $\overline{\boldsymbol{x}} := [\boldsymbol{x}^\top, 1]^\top$). The term on the right is a two-neuron ReLU network with $\ell^0$ path norm of $2\|\boldsymbol{w}\|_0 = 2N$. Also note that if the labels $y_i$ are all nonnegative (resp. nonpositive), we may discard the second (resp. first) ReLU term in (52), achieving interpolation with $\ell^0$ path norm of $\|\boldsymbol{w}\|_0 = N$. Thus in all cases, the $\ell^0$ path norm of any solution to (9) is $O(N)$. $\square$

### A.2.2 Proof of Lemma 4.1

*Proof.* We break the proof into the following steps.

**Network output on data as a sum over activation patterns.** Note that the data-fitting constraint in problems (8) and (9) can be expressed in matrix form as

$$\sum_{k=1}^{K} v_k \left(\overline{X} w_k\right)_{+} = y \tag{53}$$

recalling that $\overline{X} = [\overline{x}_1, \ldots, \overline{x}_N]^\top \in \mathbb{R}^{N \times (d+1)}$ is the matrix of augmented data points $\overline{x}_i = [x_i^\top, 1]^\top$, $y = [y_1, \ldots, y_N]^\top \in \mathbb{R}^N$ is the vector of labels, and the ReLU $(\cdot)_{+}$ is applied element-wise. Also recalling that $D_1, \ldots, D_J$ is the set of all possible $N \times N$ binary "activation pattern" matrices of the form $\mathrm{diag}(\mathbb{1}[\overline{X} u \geq 0])$ for $u \in \mathbb{R}^{d+1}$, it must be the case that the matrix $\mathrm{diag}(\mathbb{1}[\overline{X} w_k \geq 0])$ is among the $D_1, \ldots, D_J$. For any $w_k$ whose corresponding activation pattern is $D_{\mathrm{pattern}(k)}$, we have

$$(\overline{X} w_k)_{+} = D_{\mathrm{pattern}(k)} \overline{X} w_k \implies (\overline{X} w_k)_{+} v_k = D_{\mathrm{pattern}(k)} \overline{X} \tilde{w}_k$$

where $\tilde{w}_k := v_k w_k$.

For any $j = 1, \ldots, J$, let $K_j = \{k : \mathrm{pattern}(k) = j\}$ be the set of neuron indices which share the same pattern $D_j$. Then the sum of those neurons can be rewritten as

$$\sum_{k \in K_j} (\overline{X} w_k)_{+} v_k = \sum_{k \in K_j} D_j \overline{X} \tilde{w}_k = D_j \overline{X} \sum_{k \in K_j} \tilde{w}_k = D_j \overline{X} (\nu_j - \omega_j)$$

where $\nu_j$ and $\omega_j$ represent the positive and negative parts of the aggregate vector $\sum_{k \in K_j} \tilde{w}_k$, respectively, i.e.

$$\nu_j = \sum_{k \in K_j^+} v_k w_k, \qquad \omega_j = - \sum_{k \in K_j^-} v_k w_k$$

where $K_j^+ := \{k \in K_j, v_k > 0\}$ and $K_j^- := \{k \in K_j, v_k < 0\}$, so that

$$\nu_j - \omega_j = \sum_{k \in K_j^+} v_k w_k + \sum_{k \in K_j^-} v_k w_k = \sum_{k \in K_j} \tilde{w}_k$$

Therefore, the entire network output can be written as

$$\sum_{k=1}^{K} (\overline{X} w_k)_{+} v_k = \sum_{j=1}^{J} D_j \overline{X} (\nu_j - \omega_j)$$

with the understanding that, if the set $K_j$ is empty for some $j$, the vector $\nu_j - \omega_j := \sum_{k \in K_j} v_k w_k$ is the zero vector.

**Objective achieves its lower bound with two neurons per activation pattern.** Following the notation above, the objectives of (8) and (9) can be rewritten as:

$$\sum_{k=1}^{K} \|v_k w_k\|_p^p = \sum_{k=1}^{K} \|\tilde{w}_k\|_p^p = \sum_{j=1}^{J} \left( \sum_{k \in K_j^+} \|\tilde{w}_k\|_p^p + \sum_{k \in K_j^-} \|\tilde{w}_k\|_p^p \right)$$

$$\sum_{k=1}^{K} \|v_k w_k\|_0 = \sum_{k=1}^{K} \|\tilde{w}_k\|_0 = \sum_{j=1}^{J} \left( \sum_{k \in K_j^+} \|\tilde{w}_k\|_0 + \sum_{k \in K_j^-} \|\tilde{w}_k\|_0 \right)$$

Observe that:

$$\sum_{k \in K_j^+} \|\tilde{w}_k\|_p^p \geq \left\| \sum_{k \in K_j^+} \tilde{w}_k \right\|_p^p = \|\nu_j\|_p^p, \qquad \sum_{k \in K_j^-} \|\tilde{w}_k\|_p^p \geq \left\| \sum_{k \in K_j^-} \tilde{w}_k \right\|_p^p = \|\omega_j\|_p^p \tag{54}$$

$$\sum_{k \in K_j^+} \|\tilde{w}_k\|_0 \geq \left\| \sum_{k \in K_j^+} \tilde{w}_k \right\|_0 = \|\nu_j\|_0, \qquad \sum_{k \in K_j^-} \|\tilde{w}_k\|_0 \geq \left\| \sum_{k \in K_j^-} \tilde{w}_k \right\|_0 = \|\omega_j\|_0 \tag{55}$$

where in all cases, equality holds if and only if the supports of each vector in the sum (i.e., the set of indices at which each vector is nonzero) are disjoint. This follows from applying the inequality $(a + b)^p \leq a^p + b^p$—which holds for any $a, b \geq 0$ if $0 < p < 1$ and for any $a, b \in \mathbb{R}$ if $p = 0$ (defining $0^0 = 0$), and in both cases is strict unless $a = 0$ or $b = 0$—coordinate wise.

At a global minimizer of either (8) or (9), this lower bound will be achieved. To see this, note that it is always possible to replace a single one of the vectors $\tilde{w}_k$ in each group $K_j^+$ (resp. $K_j^-$) with the vector $\nu_j$ (resp. $-\omega_j$), and set the remaining vectors in each group to zero. By definition $\nu_j = \sum_{k \in K_j^+} \tilde{w}_k$ and $\omega_j = -\sum_{k \in K_j^-} \tilde{w}_k$, so clearly the network output $\sum_{j=1}^J D_j \overline{X} \left( \sum_{k \in K_j^+} \tilde{w}_k + \sum_{k \in K_j^-} \tilde{w}_k \right) = \sum_{j=1}^J D_j \overline{X} (\nu_j - \omega_j)$ on the data $\overline{X}$ remains unchanged by this modification. And with this modification, all inequalities in (54) will clearly hold with equality. This shows that, for any solution to (8) or (9), all input weight vectors $w_k$ in any individual activation pattern group $K_j^+$ or $K_j^-$ will have disjoint supports (which is the only circumstance under which the lower bounds in (54) are achieved). In any such case, the neurons in each individual positive/negative activation pattern groups can be merged into a single nonzero neuron containing their sum, without affecting either the network's ability to interpolate the data or the value of the sums $\sum_{k \in K_j^+} \|\tilde{w}_k\|_0$ or $\sum_{k \in K_j^+} \|\tilde{w}_k\|_q^q$ for any $0 < q < 1$. Note that, although this merging may alter the function represented by the neural network, it will preserve the values of $\sum_{k=1}^K \|v_k w_k\|_0$ and $\sum_{k=1}^K \|v_k w_k\|_q^q$ for any $0 < q < 1$, which is the only thing required for the statement of the lemma and its subsequent use in proving Theorem 4.1. Therefore, we may enforce that there is at most one positively-weighted neuron $v_j^+ w_j^+ = \nu_j$ and at most one negatively-weighted neuron $v_j^- w_j^- = \omega_j$ corresponding to any possible activation pattern $j$ on the data.

**Constrain the variables $\nu_j$ and $\omega_j$ to correspond to ReLU activation patterns.** In order for a particular binary pattern $D_j$ to actually correspond to an input weight/bias $w_k$, it must be the case that $(\overline{X} w_k)_i \geq 0$ wherever $(D_j)_{ii} = 1$ and $(\overline{X} w_k)_i \leq 0$ wherever $(D_j)_{ii} = 0$. This is exactly the requirement that every entry of the vector $(2D_j - I)\overline{X} w_k \in \mathbb{R}^N$ is nonnegative, since

$$((2D_j - I)\overline{X} w_k)_i = \begin{cases} (\overline{X} w_k)_i, & \text{if } (D_j)_{ii} = 1 \\ -(\overline{X} w_k)_i, & \text{if } (D_j)_{ii} = 0 \end{cases}$$

When we re-parameterize as $\tilde{w}_k = v_k w_k$ and split the neuron indices $K_j$ corroponding to activation pattern $D_j$ into the groups $K_j^+$ and $K_j^-$, the requirement that $(2D_j - I)\overline{X} w_k \geq 0$ is equivalent to requiring that $(2D_j - I)\overline{X}\tilde{w}_k \geq 0$ if $k \in K_j^+$ and $(2D_j - I)\overline{X}\tilde{w}_k \leq 0$ if $k \in K_j^-$. Because we enforce that there is at most one nonzero neuron $\tilde{w}_k = \nu_j$ (resp. $\tilde{w}_k = -\omega_j$) in each activation pattern group $K_j^+$ (resp. $K_j^-$), this condition is also clearly equivalent to $(2D_j - I)\overline{X}\nu_j \geq 0$ and $(2D_j - I)\overline{X}\omega_j \geq 0$.

**Reconstruction of solutions to (8) and (9) from solutions to (10).** By incorporating the above constraints, we have fully reparameterized the neural network problems (8) and (9) as claimed in the lemma. Because we enforce that there is at most one nonzero neuron $\tilde{w}_k = \nu_j$ (resp. $\tilde{w}_k = -\omega_j$) in each activation pattern group $K_j^+$ (resp. $K_j^-$), solutions to problem (8) can be recovered from solutions to (10) as

$$\{w_k\}_{k=1}^K = \left\{ \frac{\nu_j}{\alpha_j}, \nu_j \neq 0 \right\} \cup \left\{ \frac{\omega_j}{\beta_j}, \omega_j \neq 0 \right\} \tag{56}$$

$$\{v_k\}_{k=1}^K = \{\alpha_j, \nu_j \neq 0\} \cup \{-\beta_j, \omega_j \neq 0\} \tag{57}$$

for any constants $\alpha_1, \beta_1 \ldots, \alpha_J, \beta_J > 0$, the choice of which affects neither the network's represented function, nor its value of $\sum_{k=1}^K \|v_k w_k\|_0$ or $\sum_{k=1}^K \|v_k w_k\|_q^q$ for any $0 < q < 1$. Note that, if there were a solution to (10) with $|\{j : \nu_j \neq 0\}| + |\{j : \omega_j \neq 0\}| > N$, this would yield a solution to (8) or (9) with $K = |\{j : \nu_j \neq 0\}| + |\{j : \omega_j \neq 0\}| > N$ active neurons, contradicting Proposition 4.1. □

### A.2.3 Proof of Theorem 4.1

*Proof.* Problem (10) can be expressed more compactly in matrix form as

$$\underset{\boldsymbol{z} \in \mathbb{R}^{2J(d+1)}}{\arg\min} \|\boldsymbol{z}\|_p^p , \text{ subject to } \boldsymbol{A}\boldsymbol{z} = \boldsymbol{y}, \ \boldsymbol{G}\boldsymbol{z} \geq \boldsymbol{0} \qquad (58)$$

in the case $0 < p < 1$, or as

$$\underset{\boldsymbol{z} \in \mathbb{R}^{2J(d+1)}}{\arg\min} \|\boldsymbol{z}\|_0 , \text{ subject to } \boldsymbol{A}\boldsymbol{z} = \boldsymbol{y}, \ \boldsymbol{G}\boldsymbol{z} \geq \boldsymbol{0} \qquad (59)$$

in the case $p = 0$, where

$$\boldsymbol{z} := [\boldsymbol{\nu}_1^\top, \boldsymbol{\omega}_1^\top, \ldots, \boldsymbol{\nu}_J^\top, \boldsymbol{\omega}_J^\top]^\top \in \mathbb{R}^{2J(d+1)} \qquad (60)$$

$$\boldsymbol{A} := [\boldsymbol{D}_1\overline{\boldsymbol{X}}, -\boldsymbol{D}_1\overline{\boldsymbol{X}}, \ldots, \boldsymbol{D}_J\overline{\boldsymbol{X}}, -\boldsymbol{D}_J\overline{\boldsymbol{X}}] \in \mathbb{R}^{N \times 2J(d+1)} \qquad (61)$$

$$\boldsymbol{G} := \operatorname{diag}\left((2\boldsymbol{D}_1 - \boldsymbol{I})\overline{\boldsymbol{X}}, (2\boldsymbol{D}_1 - \boldsymbol{I})\overline{\boldsymbol{X}}, \ldots, (2\boldsymbol{D}_J - \boldsymbol{I})\overline{\boldsymbol{X}}, (2\boldsymbol{D}_J - \boldsymbol{I})\overline{\boldsymbol{X}}\right) \in \mathbb{R}^{2JN \times 2J(d+1)} \qquad (62)$$

We proceed in the following steps, which employ arguments similar to those of Yang et al. (2022a) (Theorem 2.1) and Peng et al. (2015) (Theorem 1), with minor modifications to account for the inequality constraint in (58). We note that the justification of $p$-independent $\ell^\infty$ boundedness of solutions given in Peng et al. (2015) appears to be incorrect, with Yang et al. (2022a) presenting the correct justification that we follow here.

**Solutions to (58) for any $0 < p < 1$ are contained in an $\ell^\infty$ ball of $p$-independent radius $C$.** We let $\operatorname{supp}(\boldsymbol{u})$ denote the set of nonzero indices of a vector $\boldsymbol{u}$. $\boldsymbol{M}_S$ denotes the submatrix formed by restricting its columns to an index set $S$, and $\boldsymbol{M}_{I,S}$ denotes restriction of the rows to an index set $I$ and columns to an index set $S$.

Let $\boldsymbol{z}^*$ be a solution to (58) for arbitrary $0 < p < 1$. Let $S = \operatorname{supp}(\boldsymbol{z}^*)$. Let $I = \{i : (\boldsymbol{G}\boldsymbol{z}^*)_i = 0\}$. We begin by showing that the matrix

$$\widetilde{\boldsymbol{A}}_{I,S} := \begin{bmatrix} \boldsymbol{A}_S \\ \boldsymbol{G}_{I,S} \end{bmatrix} \in \mathbb{R}^{(N+|I|) \times |S|} \qquad (63)$$

has full column rank $|S|$. Assume by contradiction that $\operatorname{rank}(\widetilde{\boldsymbol{A}}_{I,S}) < |S|$, and therefore $\widetilde{\boldsymbol{A}}_{I,S}\boldsymbol{c}_S = \boldsymbol{0}$ for some nonzero $\boldsymbol{c}_S \in \mathbb{R}^{|S|}$. Extending $\boldsymbol{c}_S$ to a vector $\boldsymbol{c} \in \mathbb{R}^{2J(d+1)}$ by zero-padding, we thus have $\boldsymbol{A}\boldsymbol{c} = \widetilde{\boldsymbol{A}}_{I,S}\boldsymbol{c}_S = \boldsymbol{0}$ and therefore $\boldsymbol{A}(\boldsymbol{z}^* \pm t\boldsymbol{c}) = \boldsymbol{A}\boldsymbol{z}^* = \boldsymbol{y}$ for any $t \in \mathbb{R}$. Similarly, $\boldsymbol{G}_I\boldsymbol{c} = \boldsymbol{G}_{I,S}\boldsymbol{c}_S = \boldsymbol{0}$, and therefore $\boldsymbol{G}_I(\boldsymbol{z}^* \pm t\boldsymbol{c}) = \boldsymbol{G}_I\boldsymbol{z}^* = \boldsymbol{0}$ for any $t \in \mathbb{R}$. If $t > 0$ is chosen small enough that $\operatorname{sgn}((\boldsymbol{G}(\boldsymbol{z}^* \pm t\boldsymbol{c}))_i) = \operatorname{sgn}((\boldsymbol{G}\boldsymbol{z}^*)_i)$ for $i \notin I$, we will thus have $\boldsymbol{G}(\boldsymbol{z}^* \pm t\boldsymbol{c}) \geq \boldsymbol{0}$, so that $\boldsymbol{z}^* \pm t\boldsymbol{c}$ are both feasible for (58).

Now choose $t$ small enough that, in addition to the previous sign requirement involving $\boldsymbol{G}$, we also have $\operatorname{sgn}(z_i^* \pm tc_i) = \operatorname{sgn}(z_i^*)$ for each $i \in S$. By strict concavity of $t \mapsto |t|^p$ on $t \in (-\infty, 0)$ and $t \in (0, \infty)$ we have

$$|z_i^*|^p > \frac{|z_i^* + tc_i|^p + |z_i^* - tc_i|^p}{2} \qquad (64)$$

for each $i \in S$ with $c_i \neq 0$ (if $c_i = 0$ the above holds with equality). Since at least one of the $c_i \neq 0$, this implies that

$$\|\boldsymbol{z}^*\|_p^p > \frac{\|\boldsymbol{z}^* + t\boldsymbol{c}\|_p^p + \|\boldsymbol{z}^* - t\boldsymbol{c}\|_p^p}{2} \qquad (65)$$

strictly. But then at least one of $\|\boldsymbol{z}^* + \boldsymbol{c}\|_p^p < \|\boldsymbol{z}^*\|_p^p$ or $\|\boldsymbol{z}^* - \boldsymbol{c}\|_p^p < \|\boldsymbol{z}^*\|_p^p$ holds strictly. Because $\boldsymbol{z}^* \pm t\boldsymbol{c}$ are both feasible for (58), this contradicts optimality of $\boldsymbol{z}^*$.

Having shown that $\widetilde{\boldsymbol{A}}_{I,S}$ is full column rank, the rank-nullity theorem implies that $\ker(\widetilde{\boldsymbol{A}}_{I,S}) = \{\boldsymbol{0}\}$. Therefore $\widetilde{\boldsymbol{A}}_{I,S}$ is injective, and thus $\boldsymbol{z}_S^*$ is the *unique* solution to

$$\widetilde{\boldsymbol{A}}_{I,S}\boldsymbol{z} = \begin{bmatrix} \boldsymbol{y} \\ \boldsymbol{0} \end{bmatrix} =: \widetilde{\boldsymbol{y}} \qquad (66)$$

This implies that $\boldsymbol{z}^*$ lies in the finite set

$$Z = \left\{ \boldsymbol{z} \mid I \subset \{1, \ldots, 2JN\}, S \subset \{1, \ldots, 2J(d+1)\}, \operatorname{rank}(\widetilde{\boldsymbol{A}}_{I,S}) = |S|, \widetilde{\boldsymbol{A}}_{I,S}\boldsymbol{z}_S = \widetilde{\boldsymbol{y}}, \boldsymbol{z}_{S^c} = \boldsymbol{0} \right\} \qquad (67)$$

which clearly depends only on $\boldsymbol{A}, \boldsymbol{G}, \boldsymbol{y}$ and not on $p$. Therefore $\|\boldsymbol{z}^*\|_\infty \leq C := \max_{\boldsymbol{z} \in Z} \|\boldsymbol{z}\|_\infty < \infty$, where $C$ is independent of $p$.

**Projection of the feasible set into the positive orthant.** Define $R_0 := \|z_0\|_\infty$ for an arbitrary solution $z_0$ to (59) with $p = 0$. Let $R := \max\{C, R_0\}$ for the $C$ defined above. The set

$$\Omega := \{z \in \mathbb{R}^{2J(d+1)} \mid Az = y, Gz \geq 0, \|z\|_\infty \leq R\} \tag{68}$$

is a polytope. As shown above, any solution to (58) for any $0 < p < 1$ is attained on $\Omega$, and by definition, at least one solution to (59) for $p = 0$ is attained on $\Omega$.

The map $z \mapsto \|z\|_p^p$ is not concave on all of $\mathbb{R}^{2J(d+1)}$, but it is strictly concave on each individual orthant, so to apply the Bauer maximum principle as in the proof of Theorem 3.2, we will relate (58) to an optimization over a projection of the polytope $\Omega$ to the nonnegative orthant $\mathbb{R}_+^{2J(d+1)}$. To do so, note that the set

$$\Psi := \left\{(z, z') \in \mathbb{R}^{2J(d+1)} \times \mathbb{R}_+^{2J(d+1)} \mid z \in \Omega, \|z'\|_\infty \leq R, |z| \leq z'\right\}, \tag{69}$$

is a polytope in the product space $\mathbb{R}^{2J(d+1)} \times \mathbb{R}_+^{2J(d+1)}$. (Here the *module vector* $|z|$ is the vector of absolute values of entries of $z$.) Because the coordinate projection of a polytope is a polytope (Goemans (2009)), the set

$$\Omega' := \left\{z' \in \mathbb{R}_+^{2J(d+1)} \mid \|z'\|_\infty \leq R, \ \exists\, z \in \Omega \text{ s.t. } |z| \leq z'\right\}, \tag{70}$$

which is given by the coordinate projection of $\Psi$ onto the $z'$ coordinate, is a polytope in $\mathbb{R}_+^{2J(d+1)}$. Furthermore, $\min_{z \in \Omega} \|z\|_p^p = \min_{z' \in \Omega'} \|z'\|_p^p$. To see this, note that for any $z \in \Omega$, its module vector $|z| \in \Omega'$, so $\min_{z \in \Omega} \|z\|_p^p \geq \min_{z' \in \Omega'} \|z'\|_p^p$. If that inequality were strict, then there would be some $z \in \Omega$ with $|z| < z_*' \ni \arg\min_{z' \in \Omega'} \|z'\|_p^p$, but this would imply that $\min_{z \in \Omega} \|z\|_p^p < \min_{z' \in \Omega} \|z'\|_p^p$.

As a polytope, $\Omega'$ is compact, convex, and has finitely many extreme points, the set of which we denote $\text{Ext}(\Omega')$. Let

$$r := \min\{z_i' > 0 \mid z' = [z_1', \ldots, z_{2J(d+1)}']^\top \in \text{Ext}(\Omega')\} \tag{71}$$

be the smallest nonzero coordinate in any of the extreme points of $\Omega'$.

Next, note that for $0 < p < 1$, the objective $z \mapsto \|z\|_p^p$ is continuous and strictly concave on the nonnegative orthant $\mathbb{R}_+^{2J(d+1)}$, and thus on $\Omega'$. Therefore, by the Bauer maximum principle (Aliprantis and Border (2006), Theorem 4.104), a solution to $\arg\min_{z' \in \Omega'} \|z'\|_p^p$ exists at an extreme point of $\Omega'$. In particular, by strict concavity of $z \mapsto \|z\|_p^p$, *any* solution to $\arg\min_{z' \in \Omega'} \|z'\|_p^p$ must be at an extreme point of $\Omega'$. (Otherwise, if such a solution had $z' = ta' + (1-t)b'$ for distinct $a', b' \in \Omega'$ and $t \in (0, 1)$, then $\|z'\|_p^p > t\|a'\|_p^p + (1-t)\|b'\|_p^p \geq t\|z'\|_p^p + (1-t)\|z'\|_p^p = \|z'\|_p^p$ which is impossible.)

**Sparse recovery result.** Putting everything together, fix an arbitrary $0 < p < 1$ and let $z_p$ be a solution to (58) for that $p$. The previous paragraph shows that $|z_p|$ is a solution to $\arg\min_{z' \in \Omega'} \|z'\|_p^p$, and therefore $|z_p| \in \text{Ext}(\Omega')$. Then:

$$\|z_p\|_0 = \|r^{-1}|z_p|\|_0 = \lim_{q \downarrow 0} \sum_{i=1}^J \left(\frac{|z_{p,i}|}{r}\right)^q \tag{72}$$

$$\leq \sum_{i=1}^J \left(\frac{|z_{p,i}|}{r}\right)^p = r^{-p} \min_{z' \in \Omega'} \|z'\|_p^p = r^{-p} \min_{z \in \Omega} \|z\|_p^p = \left(\frac{R}{r}\right)^p \min_{z \in \Omega} \|R^{-1}z\|_p^p \tag{73}$$

$$\leq \left(\frac{R}{r}\right)^p \min_{z \in \Omega} \|R^{-1}z\|_0 = \left(\frac{R}{r}\right)^p \min_{z \in \Omega} \|z\|_0 \tag{74}$$

where the inequalities come from the fact that $p \mapsto x^p$ is decreasing for $x \in (0, 1)$ and increasing for $x > 1$. Because $\|z\|_0$ is a positive integer for any $z$, and (59) attains at least one solution on $\Omega$, the above shows that $z_p$ solves (59) for any $p$ satisfying

$$\left(\frac{R}{r}\right)^p \min_{z \in \Omega} \|z\|_0 < \min_{z \in \Omega} \|z\|_0 + 1 \tag{75}$$

$$\iff p < \frac{\log(\min_{z \in \Omega} \|z\|_0 + 1) - \log(\min_{z \in \Omega} \|z\|_0)}{\log R - \log r} \tag{76}$$

if $r < R$, or for any $0 < p < 1$ if $r = R$. (Note that by definition of $\Omega'$, $r \leq R$ always.)

Let $\boldsymbol{\theta}_0$ be a solution to (9) and $\boldsymbol{\theta}_p$ be a solution to (8) for any $p$ which obeys the inequality in (76), and let $\boldsymbol{\theta}'_0$ and $\boldsymbol{\theta}'_p$ be the corresponding solutions—constructed from solutions $\boldsymbol{z}_p$ and $\boldsymbol{z}_0$ to (58) and (59), respectively—as stated in Lemma 4.1. We have shown that

$$\|\boldsymbol{\theta}_p\|_0 = \|\boldsymbol{\theta}'_p\|_0 = \|\boldsymbol{z}_p\|_0 = \|\boldsymbol{z}_0\|_0 = \|\boldsymbol{\theta}'_0\|_0 = \|\boldsymbol{\theta}_0\|_0 \tag{77}$$

which proves the result. $\qquad\square$

### A.3  Experiments

All code for the experiments can be found at `https://github.com/julianakhleh/sparse_nns_lp`.

#### A.3.1  Reweighted $\ell^1$ algorithm

To implement our proposed $\ell^p$ path norm regularizer, we use the iteratively reweighted $\ell^1$ algorithm of Candes et al. (2008); Figueiredo et al. (2007), which we summarize informally here. The principal motivation is the inequality

$$|x|^p \leq |x|p|y|^{p-1} + (1-p)|y|^p \tag{78}$$

which holds for all $x \in \mathbb{R}$, all $y \in \mathbb{R} \setminus \{0\}$, and all $0 < p \leq 1$, with equality when $p = 1$ and/or when $x = y$. Applied to $x = |v_k w_{k,i}|$, we have

$$\sum_{k=1}^{K} \|v_k \boldsymbol{w}_k\|_p^p = \sum_{k=1}^{K} \sum_{i=1}^{d} |v_k w_{k,i}|^p \leq \sum_{k=1}^{K} \sum_{i=1}^{d} \left( |v_k w_{k,i}|p|y_{k,i}|^{p-1} + (1-p)|y_{k,i}|^p \right) \tag{79}$$

for any choice of constant $y_{k,i} \in \mathbb{R} \setminus \{0\}$. The iteratively reweighted $\ell^1$ algorithm attempts to minimize the $\ell^p$ path norm objective on the left hand side of (79) by minimizing its upper bound on the right. Because the choice of $v_k, w_{k,i}$ which minimizes this upper bound is invariant to the additive constant $(1-p)|y_{k,i}|^p$ term, we can equivalently choose $v_k w_{k,i}$ at each iteration $t$ to minimize only the first term $C_{k,i}|v_k w_{k,i}|$ where $C_{k,i} := p|y_{k,i}|^{p-1}$. Because the upper bound is tighter when $y_{k,i}$ is closer to the optimal values of $v_k w_{k,i}$ for this iteration $t$, we choose the constants $y_{k,i}$ as $v_k^{(t-1)} w_{k,i}^{(t-1)}$, where $v_k^{(t-1)}, w_{k,i}^{(t-1)}$ are the previous iterates. The regularization penalty thus becomes

$$\sum_{k=1}^{K} \sum_{i=1}^{d} C_{k,i}|v_k w_{k,i}| \tag{80}$$

which is simply a separable weighted $\ell^1$ penalty with weights $C_{k,i}$. This objective lends itself to a standard $\ell^1$ proximal gradient update algorithm, with each soft-thresholding step scaled appropriately according to the individual threshold $C_{k,i}$. The full algorithm is summarized in Algorithm 1.

---

**Algorithm 1** Iteratively reweighted $\ell^1$ algorithm for $\ell^p$ path norm minimization

---

**Input**: loss function $\mathcal{L}$, sparsity parameter $0 < p \leq 1$, learning rate $\gamma > 0$, regularization parameter $\lambda > 0$, total number of iterations $T$.

 

**for** $t = 1, \ldots, T$ **do**

 Compute thresholds: $C_{k,i} \leftarrow \lambda p |v_k^{(t-1)} w_{k,i}^{(t-1)}|^{p-1}$

 Gradient update for input weights: $\tilde{w}_{k,i} \leftarrow w_{k,i}^{(t-1)} - \lambda \frac{\partial \mathcal{L}(\boldsymbol{\theta})}{\partial w_{k,i}}\big|_{w_{k,i}^{(t-1)}}$

 Gradient update for output weights: $\tilde{v}_k \leftarrow v_k^{(t-1)} - \lambda \frac{\partial \mathcal{L}(\boldsymbol{\theta})}{\partial v_k}\big|_{v_k^{(t-1)}}$

 Reweighted $\ell^1$ prox update: $u_{k,i} \leftarrow \mathrm{Prox}_{C_{k,i}|\cdot|} = \mathrm{sgn}(\tilde{v}_k \tilde{w}_{k,i})(|\tilde{v}_k \tilde{w}_{k,i}| - C_{k,i})_+$

 Update input weights: $w_{k,i}^{(t)} \leftarrow \mathrm{sgn}(\tilde{w}_{k,i}^{(t)}) \frac{u_{k,i}}{\sqrt{\|\boldsymbol{u}_k\|_2}}$

 Update output weights: $v_k^{(t)} \leftarrow \mathrm{sgn}(\tilde{v}_k^{(t)}) \sqrt{\|\boldsymbol{u}_k\|_2}$      $\triangleright$ satisfies $u_k = v_k^{(t)} w_{k,i}^{(t)}$

**end for**

---

We note that there are infinitely many ways to choose the updated input/output weights $w_{k,i}^{(t)}$ and $v_k^{(t)}$ to satisfy $u_k = v_k^{(t)} w_{k,i}^{(t)}$; due to homogeneity of the ReLU (meaning that $(\alpha x)_+ = \alpha(x)_+$ for any $\alpha \geq 0$), any choice $w_{k,i}^{(t)} \leftarrow \alpha u_{k,i}$ and $v_k^{(t)} \leftarrow 1/\alpha$ for any $\alpha > 0$ would satisfy $u_k = v_k^{(t)} w_{k,i}^{(t)}$ and produce the same neural network function. The particular choice described in Algorithm 1 additionally satisfies the *balancing* constraint $\|\boldsymbol{w}_k^{(t)}\|_2 = |v_k^{(t)}|$, and we find that this selection tends to perform best in practice. We also note that, for univariate input dimension $d = 1$ and sparsity parameter $p = 1$, Algorithm 1 is equivalent to the PathProx algorithm of Yang et al. (2022b).

### A.3.2   Setup and results

We test our algorithm on two simple synthetic datasets. The first is a univariate "peak/plateau" dataset, which consists of the data/label pairs:

$$(-2, 0), (-1, 0), (0, 1), (1, 1), (2, 0), (3, 0) \tag{81}$$

For this dataset, the theory of Debarre et al. (2022) shows that the sparsest interpolant $f$ is unique, and is represented using 3 ReLU neurons as

$$f(x) = (x + 1)_+ - 2(x - 1/2)_+ + (x - 2)_+ \tag{82}$$

Our theory in Section 3 also shows that this $f$ is a global $\ell^p$-path norm minimizer for any $0 < p \leq 1$, and is the unique such minimizer for any $0 < p < 1$.

Fig. 17 shows the sparsity over time of our reweighted $\ell^1$ algorithm for three different values of $p \in \{0.4, 0.7, 1\}$, implemented in PyTorch using the Adam optimizer, along with that of Adam-only (no regularization) and AdamW weight decay. All networks share the same random initialization and are trained with MSE loss for 100,000 epochs with learning rate $\gamma = 0.01$, regularization parameter $\lambda = 0.003$ (except for unregularized Adam-only, which uses $\lambda = 0$), and hidden layer width $K = 80$. All three values of $p$ in our reweighted $\ell^1$ algorithm produce vastly sparser solutions earlier on in training than both Adam-only and AdamW; however, only $p = 0.4$ eventually recovers the true sparsest solution $f$ with 3 ReLU neurons (see Fig. 18).

Fig. 19 shows the functions learned by all five networks throughout the course of training. We see that reweighted $\ell^1$ with $p \in \{0.4, 0.7, 1\}$ all converge quickly to near-sparsest solutions, and then the small additional kinks inside $[0, 1]$ disappear gradually throughout training, with only $p = 0.4$ eliminating them completely (the final solutions for $p \in \{0.7, 1\}$ have a single extraneous active neuron of small magnitude which activates just before $x = 1/2$).

For our second experiment, we consider $N = 10$ data points in $d = 50$ dimensions. The coordinates of each data $\boldsymbol{x}_i$ point are drawn i.i.d. from $\text{Unif}[-1, 1]$, as are the labels $y_i$. As in the univariate case, we compare the sparsity over time of our reweighted $\ell^1$ algorithm for $p \in \{0.4, 0.7, 1\}$, implemented in PyTorch using the Adam optimizer, against that of Adam-only (no explicit regularization) and AdamW weight decay. All networks are trained using MSE loss for 100,000 epochs with learning rate $\gamma = 0.01$, regularization parameter $\lambda = 0.005$ (except for unregularized Adam-only, which uses $\lambda = 0$), and hidden layer width $K = 100$. Fig. 20 shows that all values of $p$ produce much sparser solutions than Adam-only and AdamW weight decay, with $p = 0.4$ producing sparser solutions than $p \in \{0.7, 1\}$. The solutions recovered by $p \in \{0.4, 0.7, 1\}$ all obey the sparsity upper bound of $2N$ guaranteed by the proof of Proposition 4.1.


Figure 17: Sparsity over time of five networks trained to interpolation on the univariate peak-plateau dataset (81). The reweighted $\ell^1$ algorithm for $\ell^p$ path norm minimization (Algorithm 1) recovers much sparser solutions earlier in training than unregularized Adam or AdamW weight decay regularization, with the smallest value $p = 0.4$ eventually recovering the sparsest possible interpolant (82).

- The abstract and/or introduction should clearly state the claims made, including the contributions made in the paper and important assumptions and limitations. A No or NA answer to this question will not be perceived well by the reviewers.
- The claims made should match theoretical and experimental results, and reflect how much the results can be expected to generalize to other settings.
- It is fine to include aspirational goals as motivation as long as it is clear that these goals are not attained by the paper.

2. **Limitations**

Question: Does the paper discuss the limitations of the work performed by the authors?

Answer: [Yes]

Justification: we discuss limitations and possible future directions in Section 6.

Guidelines:

- The answer NA means that the paper has no limitation while the answer No means that the paper has limitations, but those are not discussed in the paper.
- The authors are encouraged to create a separate "Limitations" section in their paper.
- The paper should point out any strong assumptions and how robust the results are to violations of these assumptions (e.g., independence assumptions, noiseless settings, model well-specification, asymptotic approximations only holding locally). The authors should reflect on how these assumptions might be violated in practice and what the implications would be.
- The authors should reflect on the scope of the claims made, e.g., if the approach was only tested on a few datasets or with a few runs. In general, empirical results often depend on implicit assumptions, which should be articulated.

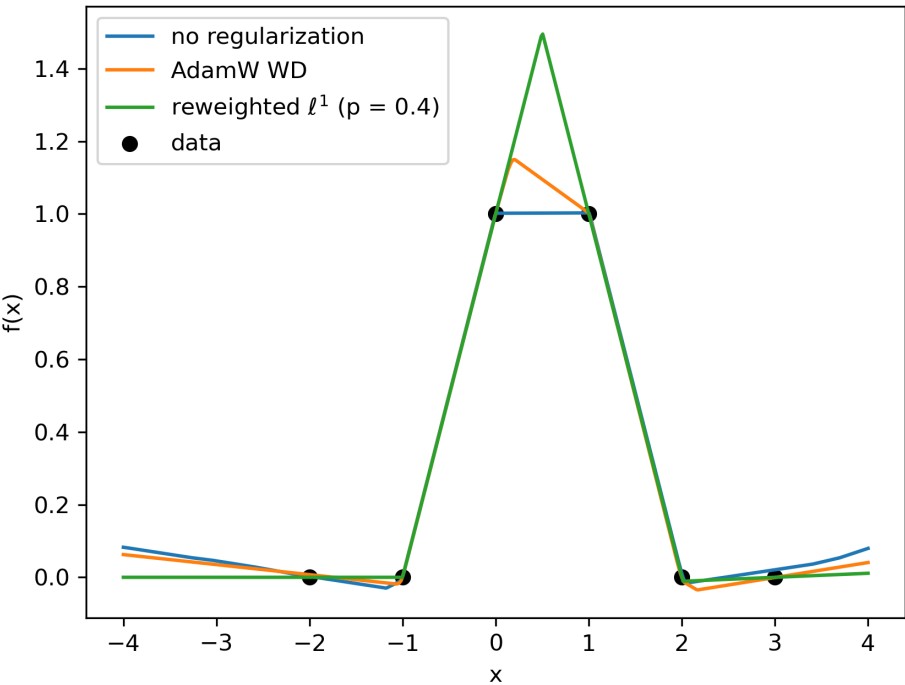

Figure 18: Three interpolants of the peak-plateau dataset, learned after 100,000 epochs using unregularized Adam, AdamW weight decay, and reweighted $\ell^1$ (Algorithm 1) with $p = 0.4$. Only the latter recovers the true sparsest interpolant (82).

- The authors should reflect on the factors that influence the performance of the approach. For example, a facial recognition algorithm may perform poorly when image resolution is low or images are taken in low lighting. Or a speech-to-text system might not be used reliably to provide closed captions for online lectures because it fails to handle technical jargon.
- The authors should discuss the computational efficiency of the proposed algorithms and how they scale with dataset size.
- If applicable, the authors should discuss possible limitations of their approach to address problems of privacy and fairness.
- While the authors might fear that complete honesty about limitations might be used by reviewers as grounds for rejection, a worse outcome might be that reviewers discover limitations that aren't acknowledged in the paper. The authors should use their best judgment and recognize that individual actions in favor of transparency play an important role in developing norms that preserve the integrity of the community. Reviewers will be specifically instructed to not penalize honesty concerning limitations.

3. **Theory assumptions and proofs**

   Question: For each theoretical result, does the paper provide the full set of assumptions and a complete (and correct) proof?

   Answer: [Yes]

   Justification: all necessary assumptions are made clear in the theorem statements, and full rigorous proofs are provided in the appendices.

   Guidelines:

   - The answer NA means that the paper does not include theoretical results.

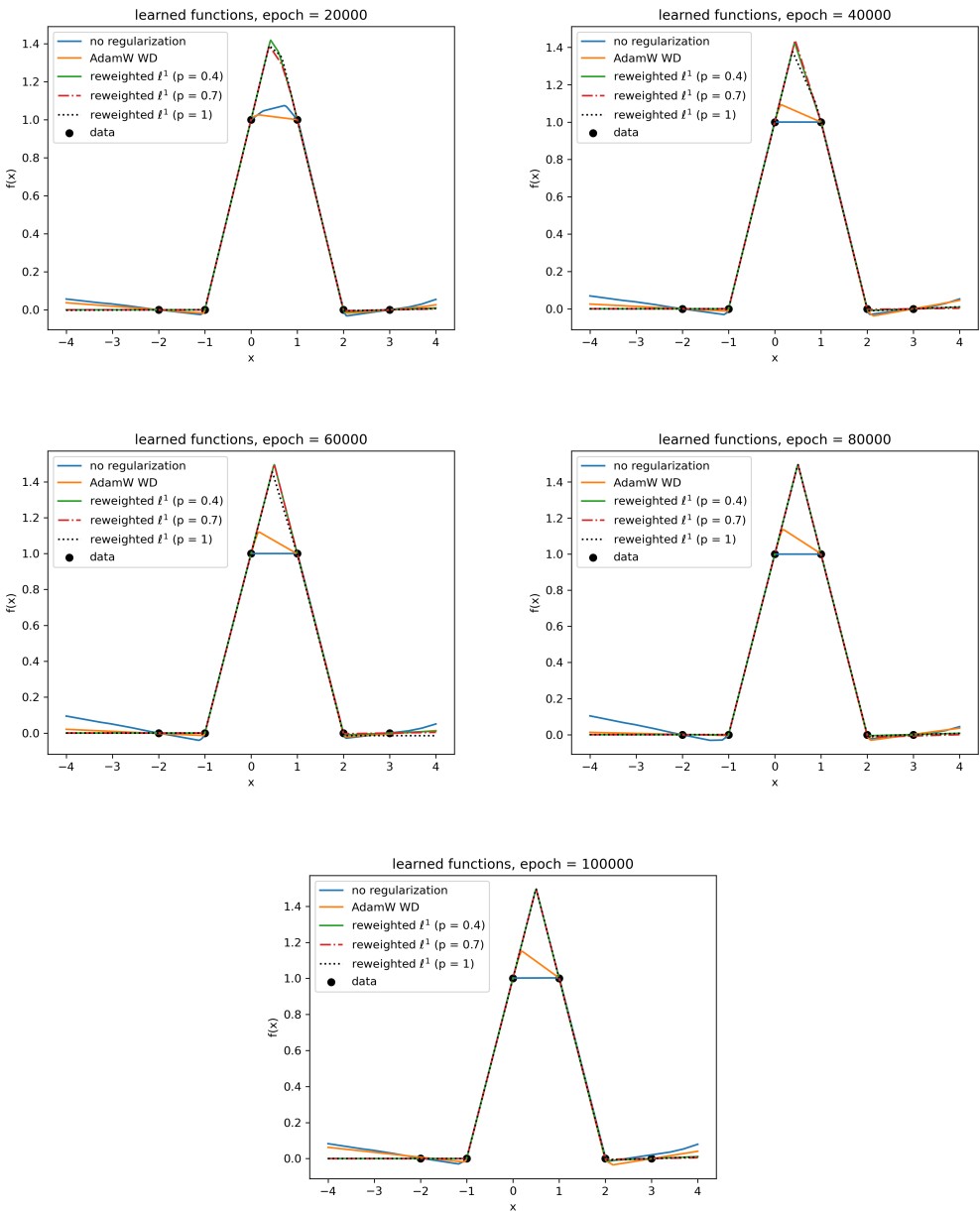

Figure 19: Learned network functions of five different algorithms throughout the course of training. Reweighted $\ell^1$ with $p \in \{0.4, 0.7, 1\}$ converge to near-sparsest solutions early on in training, with only $p = 0.4$ eventually eliminating all extraneous neurons to recover the true sparsest solution (82).

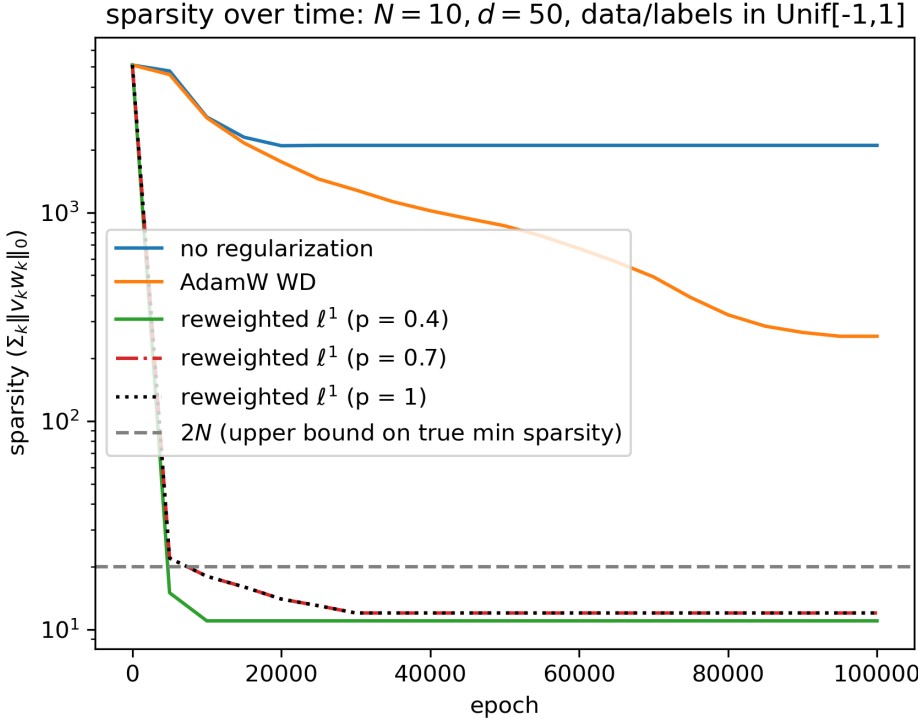

Figure 20: Sparsity over time of five networks trained to interpolation on $N = 10$ uniform random data points in $d = 50$. The solutions obtained by the $\ell^1$ algorithm (Algorithm 1) for $p \in \{0.4, 0.7, 1\}$ satisfy the sparsity upper bound of $2N$ guaranteed by the proof of Proposition 4.1.

- All the theorems, formulas, and proofs in the paper should be numbered and cross-referenced.
- All assumptions should be clearly stated or referenced in the statement of any theorems.
- The proofs can either appear in the main paper or the supplemental material, but if they appear in the supplemental material, the authors are encouraged to provide a short proof sketch to provide intuition.
- Inversely, any informal proof provided in the core of the paper should be complemented by formal proofs provided in appendix or supplemental material.
- Theorems and Lemmas that the proof relies upon should be properly referenced.

4. **Experimental result reproducibility**

Question: Does the paper fully disclose all the information needed to reproduce the main experimental results of the paper to the extent that it affects the main claims and/or conclusions of the paper (regardless of whether the code and data are provided or not)?

Answer: [NA]

Justification: no experiments.

Guidelines:

- The answer NA means that the paper does not include experiments.
- If the paper includes experiments, a No answer to this question will not be perceived well by the reviewers: Making the paper reproducible is important, regardless of whether the code and data are provided or not.
- If the contribution is a dataset and/or model, the authors should describe the steps taken to make their results reproducible or verifiable.
- Depending on the contribution, reproducibility can be accomplished in various ways. For example, if the contribution is a novel architecture, describing the architecture fully

might suffice, or if the contribution is a specific model and empirical evaluation, it may be necessary to either make it possible for others to replicate the model with the same dataset, or provide access to the model. In general. releasing code and data is often one good way to accomplish this, but reproducibility can also be provided via detailed instructions for how to replicate the results, access to a hosted model (e.g., in the case of a large language model), releasing of a model checkpoint, or other means that are appropriate to the research performed.

- While NeurIPS does not require releasing code, the conference does require all submissions to provide some reasonable avenue for reproducibility, which may depend on the nature of the contribution. For example

  (a) If the contribution is primarily a new algorithm, the paper should make it clear how to reproduce that algorithm.

  (b) If the contribution is primarily a new model architecture, the paper should describe the architecture clearly and fully.

  (c) If the contribution is a new model (e.g., a large language model), then there should either be a way to access this model for reproducing the results or a way to reproduce the model (e.g., with an open-source dataset or instructions for how to construct the dataset).

  (d) We recognize that reproducibility may be tricky in some cases, in which case authors are welcome to describe the particular way they provide for reproducibility. In the case of closed-source models, it may be that access to the model is limited in some way (e.g., to registered users), but it should be possible for other researchers to have some path to reproducing or verifying the results.

5. **Open access to data and code**

   Question: Does the paper provide open access to the data and code, with sufficient instructions to faithfully reproduce the main experimental results, as described in supplemental material?

   Answer: [Yes]

   Justification: all code for the experiments is available publicly at the aforementioned `https://github.com/julianakhleh/sparse_nns_lp`.

   Guidelines:

   - The answer NA means that paper does not include experiments requiring code.
   - Please see the NeurIPS code and data submission guidelines (`https://nips.cc/public/guides/CodeSubmissionPolicy`) for more details.
   - While we encourage the release of code and data, we understand that this might not be possible, so "No" is an acceptable answer. Papers cannot be rejected simply for not including code, unless this is central to the contribution (e.g., for a new open-source benchmark).
   - The instructions should contain the exact command and environment needed to run to reproduce the results. See the NeurIPS code and data submission guidelines (`https://nips.cc/public/guides/CodeSubmissionPolicy`) for more details.
   - The authors should provide instructions on data access and preparation, including how to access the raw data, preprocessed data, intermediate data, and generated data, etc.
   - The authors should provide scripts to reproduce all experimental results for the new proposed method and baselines. If only a subset of experiments are reproducible, they should state which ones are omitted from the script and why.
   - At submission time, to preserve anonymity, the authors should release anonymized versions (if applicable).
   - Providing as much information as possible in supplemental material (appended to the paper) is recommended, but including URLs to data and code is permitted.

6. **Experimental setting/details**

   Question: Does the paper specify all the training and test details (e.g., data splits, hyperparameters, how they were chosen, type of optimizer, etc.) necessary to understand the results?

Answer: [Yes]

Justification: details of experiments are fully described in Appendix A.3.

Guidelines:

- The answer NA means that the paper does not include experiments.
- The experimental setting should be presented in the core of the paper to a level of detail that is necessary to appreciate the results and make sense of them.
- The full details can be provided either with the code, in appendix, or as supplemental material.

7. **Experiment statistical significance**

Question: Does the paper report error bars suitably and correctly defined or other appropriate information about the statistical significance of the experiments?

Answer: [No]

Justification: our experiments are performed only on synthetic data and are meant to illustrate the feasibility of the regularizer proposed by our theory; as such, statistical significance tests are not necessary.

Guidelines:

- The answer NA means that the paper does not include experiments.
- The authors should answer "Yes" if the results are accompanied by error bars, confidence intervals, or statistical significance tests, at least for the experiments that support the main claims of the paper.
- The factors of variability that the error bars are capturing should be clearly stated (for example, train/test split, initialization, random drawing of some parameter, or overall run with given experimental conditions).
- The method for calculating the error bars should be explained (closed form formula, call to a library function, bootstrap, etc.)
- The assumptions made should be given (e.g., Normally distributed errors).
- It should be clear whether the error bar is the standard deviation or the standard error of the mean.
- It is OK to report 1-sigma error bars, but one should state it. The authors should preferably report a 2-sigma error bar than state that they have a 96% CI, if the hypothesis of Normality of errors is not verified.
- For asymmetric distributions, the authors should be careful not to show in tables or figures symmetric error bars that would yield results that are out of range (e.g. negative error rates).
- If error bars are reported in tables or plots, The authors should explain in the text how they were calculated and reference the corresponding figures or tables in the text.

8. **Experiments compute resources**

Question: For each experiment, does the paper provide sufficient information on the computer resources (type of compute workers, memory, time of execution) needed to reproduce the experiments?

Answer: [No]

Justification: our experiments are small-scale and computationally light and can easily be run on almost any computational setup, so we do not feel the need to report specifics on the compute resources.

Guidelines:

- The answer NA means that the paper does not include experiments.
- The paper should indicate the type of compute workers CPU or GPU, internal cluster, or cloud provider, including relevant memory and storage.
- The paper should provide the amount of compute required for each of the individual experimental runs as well as estimate the total compute.

- The paper should disclose whether the full research project required more compute than the experiments reported in the paper (e.g., preliminary or failed experiments that didn't make it into the paper).

9. **Code of ethics**

Question: Does the research conducted in the paper conform, in every respect, with the NeurIPS Code of Ethics https://neurips.cc/public/EthicsGuidelines?

Answer: [Yes]

Justification: Our paper has no social consequences.

Guidelines:

- The answer NA means that the authors have not reviewed the NeurIPS Code of Ethics.
- If the authors answer No, they should explain the special circumstances that require a deviation from the Code of Ethics.
- The authors should make sure to preserve anonymity (e.g., if there is a special consideration due to laws or regulations in their jurisdiction).

10. **Broader impacts**

Question: Does the paper discuss both potential positive societal impacts and negative societal impacts of the work performed?

Answer: [NA]

Justification: Our paper has no societal impacts.

Guidelines:

- The answer NA means that there is no societal impact of the work performed.
- If the authors answer NA or No, they should explain why their work has no societal impact or why the paper does not address societal impact.
- Examples of negative societal impacts include potential malicious or unintended uses (e.g., disinformation, generating fake profiles, surveillance), fairness considerations (e.g., deployment of technologies that could make decisions that unfairly impact specific groups), privacy considerations, and security considerations.
- The conference expects that many papers will be foundational research and not tied to particular applications, let alone deployments. However, if there is a direct path to any negative applications, the authors should point it out. For example, it is legitimate to point out that an improvement in the quality of generative models could be used to generate deepfakes for disinformation. On the other hand, it is not needed to point out that a generic algorithm for optimizing neural networks could enable people to train models that generate Deepfakes faster.
- The authors should consider possible harms that could arise when the technology is being used as intended and functioning correctly, harms that could arise when the technology is being used as intended but gives incorrect results, and harms following from (intentional or unintentional) misuse of the technology.
- If there are negative societal impacts, the authors could also discuss possible mitigation strategies (e.g., gated release of models, providing defenses in addition to attacks, mechanisms for monitoring misuse, mechanisms to monitor how a system learns from feedback over time, improving the efficiency and accessibility of ML).

11. **Safeguards**

Question: Does the paper describe safeguards that have been put in place for responsible release of data or models that have a high risk for misuse (e.g., pretrained language models, image generators, or scraped datasets)?

Answer: [NA]

Justification: no risks of data or model misuse.

Guidelines:

- The answer NA means that the paper poses no such risks.

- Released models that have a high risk for misuse or dual-use should be released with necessary safeguards to allow for controlled use of the model, for example by requiring that users adhere to usage guidelines or restrictions to access the model or implementing safety filters.
- Datasets that have been scraped from the Internet could pose safety risks. The authors should describe how they avoided releasing unsafe images.
- We recognize that providing effective safeguards is challenging, and many papers do not require this, but we encourage authors to take this into account and make a best faith effort.

12. **Licenses for existing assets**

Question: Are the creators or original owners of assets (e.g., code, data, models), used in the paper, properly credited and are the license and terms of use explicitly mentioned and properly respected?

Answer: [NA]

Justification: no use of existing assets.

Guidelines:

- The answer NA means that the paper does not use existing assets.
- The authors should cite the original paper that produced the code package or dataset.
- The authors should state which version of the asset is used and, if possible, include a URL.
- The name of the license (e.g., CC-BY 4.0) should be included for each asset.
- For scraped data from a particular source (e.g., website), the copyright and terms of service of that source should be provided.
- If assets are released, the license, copyright information, and terms of use in the package should be provided. For popular datasets, `paperswithcode.com/datasets` has curated licenses for some datasets. Their licensing guide can help determine the license of a dataset.
- For existing datasets that are re-packaged, both the original license and the license of the derived asset (if it has changed) should be provided.
- If this information is not available online, the authors are encouraged to reach out to the asset's creators.

13. **New assets**

Question: Are new assets introduced in the paper well documented and is the documentation provided alongside the assets?

Answer: [NA]

Justification: no new assets.

Guidelines:

- The answer NA means that the paper does not release new assets.
- Researchers should communicate the details of the dataset/code/model as part of their submissions via structured templates. This includes details about training, license, limitations, etc.
- The paper should discuss whether and how consent was obtained from people whose asset is used.
- At submission time, remember to anonymize your assets (if applicable). You can either create an anonymized URL or include an anonymized zip file.

14. **Crowdsourcing and research with human subjects**

Question: For crowdsourcing experiments and research with human subjects, does the paper include the full text of instructions given to participants and screenshots, if applicable, as well as details about compensation (if any)?

Answer: [NA]

Justification: no crowdsourcing or human subject research.

Guidelines:

- The answer NA means that the paper does not involve crowdsourcing nor research with human subjects.
- Including this information in the supplemental material is fine, but if the main contribution of the paper involves human subjects, then as much detail as possible should be included in the main paper.
- According to the NeurIPS Code of Ethics, workers involved in data collection, curation, or other labor should be paid at least the minimum wage in the country of the data collector.

15. **Institutional review board (IRB) approvals or equivalent for research with human subjects**

Question: Does the paper describe potential risks incurred by study participants, whether such risks were disclosed to the subjects, and whether Institutional Review Board (IRB) approvals (or an equivalent approval/review based on the requirements of your country or institution) were obtained?

Answer: [NA]

Justification: no human subject research.

Guidelines:

- The answer NA means that the paper does not involve crowdsourcing nor research with human subjects.
- Depending on the country in which research is conducted, IRB approval (or equivalent) may be required for any human subjects research. If you obtained IRB approval, you should clearly state this in the paper.
- We recognize that the procedures for this may vary significantly between institutions and locations, and we expect authors to adhere to the NeurIPS Code of Ethics and the guidelines for their institution.
- For initial submissions, do not include any information that would break anonymity (if applicable), such as the institution conducting the review.

16. **Declaration of LLM usage**

Question: Does the paper describe the usage of LLMs if it is an important, original, or non-standard component of the core methods in this research? Note that if the LLM is used only for writing, editing, or formatting purposes and does not impact the core methodology, scientific rigorousness, or originality of the research, declaration is not required.

Answer: [NA]

Justification: LLMs were used for writing/editing, literature review, and occasional assistance in proving theorems (mainly searching for related/relevant existing results), which we do not consider an important, original, or non-standard usage.

Guidelines:

- The answer NA means that the core method development in this research does not involve LLMs as any important, original, or non-standard components.
- Please refer to our LLM policy (https://neurips.cc/Conferences/2025/LLM) for what should or should not be described.

