# OpenReview forum: "Global Minimizers of $\ell^p$-Regularized Objectives Yield the Sparsest ReLU Neural Networks"
_NeurIPS.cc/2025/Conference — NeurIPS 2025 poster_

### Official Review · Reviewer_vc8p · 2025-06-27

**Clarity:** 2
**Significance:** 2
**Originality:** 2
**Rating:** 4
**Confidence:** 2

**Summary:**

The paper introduces a smooth ℓₚ (0<p<1) path-norm regularizer for single-hidden-layer ReLU networks whose global minimizers provably coincide with the sparsest interpolating networks—i.e., those with the fewest nonzero neurons—thereby recasting the combinatorial ℓ₀ problem as a differentiable objective . In the univariate case, it additionally establishes uniqueness (for almost every p), explicit N–2 width and parameter bounds, and Lipschitz control, while in arbitrary dimensions it shows that for sufficiently small p any minimizer recovers an ℓ₀ solution via a finite-dimensional concave-over-polytope reformulation.

**Questions:**

1. How might the theory extend to deeper ReLU architectures—do global minimizers of the ℓₚ path‐norm still recover sparsest networks beyond one hidden layer?

2. What practical guidelines can you offer for choosing “sufficiently small” p to balance sparsity recovery against numerical stability during optimization?

3. Have you tested the concave‐over‐polytope reformulation empirically in moderate‐ to high‐dimensional settings, and if so, how does its computational cost scale?

**Ethical Concerns:**

["NO or VERY MINOR ethics concerns only"]

**Final Justification:**

In the light of authors' response, my concerns has been clarified to some extent so I increased my score.

**Quality:**

2

**Strengths And Weaknesses:**

Overall Strengths

* Provable Sparsity Guarantees: Establishes that global minimizers of the smooth ℓₚ path‐norm (0<p<1) exactly recover the sparsest ReLU networks, effectively bridging continuous optimization and combinatorial ℓ₀ minimization.

* Rigorous Univariate Analysis: Provides uniqueness of minimizers (for almost every p), explicit width‐(N–2) and parameter bounds, plus Lipschitz control in the one‐dimensional case.

* General‐Dimension Extension: Shows that for sufficiently small p, any minimizer corresponds to an ℓ₀ solution via a finite‐dimensional concave‐over‐polytope reformulation, highlighting broad applicability.


Overall Weaknesses

* Architectural Limitation: Analysis is confined to single‐hidden‐layer ReLU networks; it remains unclear how these results extend to deeper or more complex architectures.

* Choice of p: Requires “sufficiently small” p < 1 for the general‐dimension result, but offers limited practical guidance on selecting p or managing numerical instability as p→0.

* Computational Scalability: The concave‐over‐polytope reformulation, while elegant, may be intractable in high dimensions without further algorithmic development.

---

> ### Author Rebuttal · Authors · 2025-07-31
>
> We thank the reviewer for their feedback, but we think there may be some important misunderstandings regarding the key elements of our paper. Below we address the questions and concerns individually.
> - **Role of the "concave-over-polytope" problem reformulation:** the reviewer seems to severely misunderstand the role that our reformulation of the multivariate neural network training problem (as a concave minimization over a linear constraint set) plays in our analysis. We do *not* propose this reformulation as part of an algorithm for $\ell^p$ path norm neural network minimization problem; nowhere in the paper do we propose any algorithm. Rather, this reformulation is a proof technique for showing  $\ell^p$ path norm minimization with neural networks recovers $\ell^0$ solutions for sufficiently small $p$. Therefore, this reformulation is not something which needs to be tested numerically.
> - **Extension to deeper networks:** we agree that extending our results to deeper networks would be interesting and desirable. We conjecture that this is probably possible, but difficult, and would require different proof techniques than the ones used in our results. Specifically, the difficulties in extending our proofs to deep networks are the following:
>     - Univariate deep networks: like shallow, these are CPWL functions, but the relationship between the knots in the function and the neurons is more complex than in the shallow case. Specifically, modifying a single neuron of a shallow univariate network only produces a local change in the represented function, but for deep univariate networks, modifying neurons in deeper layers can produce global changes in the represented function. This makes our proof approach for the univariate shallow case difficult to apply to deep networks.
>     - Multivariate deep networks: although [1] develops a reformulation of three-layer ReLU network training as a linear constraint problem, similar to the two-layer formulation that inspires our proof approach, the resulting reformulated problem minimizes an $\ell^{2,p}$ group norm (rather than an $\ell^p$ norm) subject to a set of linear constraints. Unlike the $\ell^p$ norm, the $\ell^{2,p}$ group norm is neither convex nor concave, so our arguments here also do not readily extend to the deep case.
>
>     Thus, any extension to deeper networks would likely require different proof strategies than those used in our paper. Nonetheless, we believe that our results for the shallow case are valuable, as they provide (to our knowledge) the first sparse recovery guarantee for neural networks with arbitrary data, and develop innovative proof techniques for analyzing $\ell^p$ minimization in the highly nonlinear, nonconvex neural network context.
>
> [1] Ergen and Pilanci. "Global Optimality Beyond Two Layers: Training Deep ReLU Networks via Convex Programs." 2021.
>
> - **Computation of the critical threshold $p^*$:** our proofs do highlight a possible approach for computing p*----in both the univariate and multivariate cases, this can be done by enumerating and searching over the extreme points of a polytope determined by data-dependent constraints. However, there may be an enormous number of these extreme points (exponential or super-exponential in the number of data points $N$), which makes computing this value impractical. We conjecture that this difficulty may be unavoidable, because even for the simpler linear problem $\min_x \\| x \\|_p^p \  \textrm{s.t.} \ Ax = b$ with $0 < p \leq 1$, computation of restricted isometry and null space constants (which play a similar role in determining sufficient conditions for this problem to recover $\ell^0$ minimizers) is known to be NP hard in general [2]. Nonetheless, we argue that showing the existence of this $p^*$ is a significant theoretical contribution, as it rigorously confirms the natural intuition that taking $p$ close enough to 0 will eventually recover an exact $\ell^0$ solution. In practice, a simple strategy could be to retrain several times with increasingly small (e.g. halved) values of $p$ and stop once no further sparsity gains are observed. Developing more refined approaches for selecting $p$ is of interest for future work.
>
> [2] Tillman and Pfetsch. "The Computational Complexity of the Restricted Isometry Property, the Nullspace Property, and Related Concepts in Compressed Sensing." 2013.
> - **Clarity:** the reviewer rated the clarity of our submission as "1. poor." It would be helpful if the reviewer provided specific feedback on which parts were unclear so that we can further address any confusion and make any necessary changes to the paper.

---

> ### Author Response · Authors · 2025-08-07
>
> As the discussion period is nearing its end, we would appreciate feedback from the reviewer on our rebuttal, and any other potential questions or points of confusion. Thank you!

---

> > ### Comment · Reviewer_vc8p · 2025-08-08
> > **Re:**
> >
> > Thanks for your response. Based on that, I revised my review.

---

### Official Review · Reviewer_5Uw1 · 2025-07-02

**Clarity:** 3
**Significance:** 3
**Originality:** 3
**Rating:** 5
**Confidence:** 3

**Summary:**

This work proves that for sufficiently small $p$, miniming the $p$-path norm yields the sparsest shallow neural networks that interpolate the data.

**Questions:**

The results focus on min-norm interpolants. What can be said about solutions to the regularized problem $min Loss + \lambda p\text{-path norm}$? This paper is significantly more impactful if it can be said that minimizers of the regularized problem for sufficiently small $p$ correspond to "sparsest" solutions in some sense. Similarly, can anything be said about how close (S)GD, adam, etc.. run on the regularized objective gets to finding "sparsest" networks?

**Ethical Concerns:**

["NO or VERY MINOR ethics concerns only"]

**Final Justification:**

This paper has strong theory on a problem of significant interest. My original concerns were about:

1. The application to the regularized problem, which has been addressed.
2. The lack of insight about optimization methods, which is a hard problem outside the scope of this paper
3. Discussing a few missing pieces of related work, which has been addressed.

Therefore I keep my positive score.

**Limitations:**

yes

**Quality:**

3

**Strengths And Weaknesses:**

Theory is rigorous and insightful.
Relationship to prior work is clear. It is clear that there is broad interest in finding sparse neural networks, and that prior work does not provide a way to find the *sparsest* solutions, nor does it provide a way to find sparse solutions very efficiently. This work does both.

The main weakness of this setting is the focus on exact min-norm interpolants. In practice, one would probably run an optimization method (SGD, adam, etc.) on the regularized objective to train the networks.

The univariate variational reformulation in Prop 3.1 seems closely related to "How do infinite width bounded norm networks look in function space?" by Savarese et al. (2019) and "Noisy Interpolation Learning with Shallow Univariate ReLU Networks" by Joshi et al. (2023) (though these works do not focus on finding sparsest solutions). Can you cite and compare to these papers?

*Minor comments:*
- Line 26: are purposes = our purposes
- Line 308: is eq (48) the correct reference?

---

> ### Author Rebuttal · Authors · 2025-07-31
>
> We thank the reviewer for their careful evaluation of our work and for their feedback. Below we address the questions and concerns individually.
> - **Min-norm interpolation vs. regularized loss solutions:** our sparse recovery results in the multivariate case also extend to the regularized loss problem $\min_{\\theta} \\mathcal{L}(y,f_{\\theta}(x)) + \lambda \sum_{k=1}^K \\| v_k w_k \\|\_p^p$ for any loss function $\mathcal{L}$ which is piecewise affine in its second argument. This includes the commonly-used hinge loss as well as the $L^1$ and $L^{\\infty}$ losses. This holds because the regularized loss problem can be recast as $\min_{\\theta} \sum_{k=1}^K \\| v_k w_k \\|\_p^p \\ \textrm{s.t.} \\  \\mathcal{L}(y,f_{\\theta}(x)) \leq C $ , and under the piecewise linearity assumption, constraint set $\\{ \theta: \mathcal{L}(y, f_{\theta}(x)) \leq C \\}$ will be a polytope under the reformulation discussed in the multivariate section. The rest of our multivariate argument then applies to this problem verbatim. Extensions to non-piecewise-affine losses such as MSE are less clear, although potentially attainable using other techniques. Rigorously characterizing the behavior of SGD/Adam with this $\ell^p$ objective would also be interesting, but would likely require a very different set of analytical tools, and is outside the scope of this paper. Nonetheless, it is indeed possible to relax the interpolation assumption and replace it with certain types of regularized loss; we can add this to the final version of the paper. Additionally, the interpolation regime itself is of practical relevance, since neural networks in practice very often are trained to interpolation (zero or near-zero training error).
> - **Relationship to prior work:** for p = 1, our variational reformulation in the univariate case corresponds to that of Savarese et al. (2019). We will update the paper with this reference and make the relationship more explicit. The Joshi paper studies the generalization properties of sparse univariate ReLU network interpolants (sparsity here is obtained by regularizing the biases as shown in [1], which is only proven to enforce sparsity under some assumptions on the data) in comparison to the straight-line interpolant, and shows that these sparsest interpolants may or may not demonstrate either "tempered overfitting" or "catastrophic overfitting" depending on exactly how the error is measured. Although the question of generalization is somewhat separate from our focus in this work (which is on how to obtain guaranteed sparsest solutions, if these are known to desirable a priori), we can also mention this paper in our related work.
> - **Minor comments:** we will fix the typo and the incorrect equation reference in line 308.
>
> [1] Boursier and Flammarion. "Penalising the biases in norm regularisation enforces sparsity." 2023.

---

> > ### Comment · Reviewer_5Uw1 · 2025-08-05
> >
> > Thank you for your response. My concerns are resolved and I keep my positive score.

---

### Official Review · Reviewer_k9Kj · 2025-07-03

**Clarity:** 3
**Significance:** 2
**Originality:** 3
**Rating:** 4
**Confidence:** 3

**Summary:**

This work introduces a continuous regularization objective for single-hidden-layer ReLU networks using the ℓ^p​ path norm for 0<p<1. The authors prove that global minimizers of this objective correspond to the sparsest possible interpolating networks. This work recasts the combinatorial ℓ^0​ minimization into a smooth optimization, offering a principled, gradient-based alternative to heuristic pruning methods for achieving guaranteed sparsity.

**Questions:**

- The theory relies on a data dependent threshold p. From a practical perspective, how should a user choose an appropriate p during training?
- The analysis is confined to single-hidden-layer networks. What are the primary theoretical obstacles to extending these guarantees for ℓ^p/ℓ^0 equivalence to deep ReLU networks?
-Your work focuses on creating the sparsest possible model for a given dataset, which is acceptable for model compression. But, have you also tested how well these highly sparse models perform on new, unseen data? I am wondering how their accuracy compares to models trained with more standard regularizers.

**Ethical Concerns:**

["NO or VERY MINOR ethics concerns only"]

**Final Justification:**

The score was adjusted based on the authors' rebuttal.

**Limitations:**

- The theoretical sparsity guarantees only hold for sufficiently small values of p in the ℓp norm. However, the work does not provide a practical method to compute or estimate the critical threshold p* below which sparsity is guaranteed.
- The theoretical results are derived for shallow, single-hidden-layer ReLU networks. Although some results are extended to arbitrary input dimensions, the work does not address deeper architectures, which are more common in modern deep learning practice.
- The paper focuses entirely on theoretical results, with no empirical experiments to validate the practical performance of the proposed regularization method.

**Quality:**

3

**Strengths And Weaknesses:**

The work provides a rigorous theoretical guarantee that a continuous, almost-everywhere differentiable objective can produce the sparsest interpolating ReLU network. This transforms a difficult combinatorial ℓ^0 problem into a smooth optimization task and provide a principled alternative to heuristic, multi-stage pruning algorithms.

The analysis for the univariate case is strong and establishes not only sparsity but also the uniqueness of the solution for almost every p ∈ (0,1).
The authors successfully extend the core findings to the multivariate setting. By reformulating the optimization, they prove that for any dataset there exists a threshold p below that the ℓ^p​ minimizer is an ℓ^0​ (sparsest) solution.
The theoretical guarantees apply only to single-hidden-layer networks. Extending the analysis to deeper or convolutional architectures remains an open challenge, which limits the immediate applicability of the proposed objective.

---

> ### Author Rebuttal · Authors · 2025-07-31
>
> We thank the reviewer for their careful evaluation of our work and for their feedback. Below we address the questions and concerns individually.
> - **Extension to deeper networks:** we agree that extending our results to deeper networks would be interesting and desirable. We conjecture that this is probably possible, but difficult, and would require different proof techniques than the ones used in our results. Specifically, the difficulties in extending our proofs to deep networks are the following:
>     - Univariate deep networks: like shallow, these are CPWL functions, but the relationship between the knots in the function and the neurons is more complex than in the shallow case. Specifically, modifying a single neuron of a shallow univariate network only produces a local change in the represented function, but for deep univariate networks, modifying neurons in deeper layers can produce global changes in the represented function. This makes our proof approach for the univariate shallow case difficult to apply to deep networks.
>     - Multivariate deep networks: although [1] develops a reformulation of three-layer ReLU network training as a linear constraint problem, similar to the two-layer formulation that inspires our proof approach, the resulting reformulated problem minimizes an $\ell^{2,p}$ group norm (rather than an $\ell^p$ norm) subject to a set of linear constraints. Unlike the $\ell^p$ norm, the $\ell^{2,p}$ group norm is neither convex nor concave, so our arguments here also do not readily extend to the deep case.
>
>     Thus, any extension to deeper networks would likely require different proof strategies than those used in our paper. Nonetheless, we believe that our results for the shallow case are valuable, as they provide (to our knowledge) the first sparse recovery guarantee for neural networks with arbitrary data, and develop innovative proof techniques for analyzing $\ell^p$ minimization in the highly nonlinear, nonconvex neural network context.
>
> [1] Ergen and Pilanci. "Global Optimality Beyond Two Layers: Training Deep ReLU Networks via Convex Programs." 2021.
>
> - **Computation of the critical threshold $p^*$:** our proofs do highlight a possible approach for computing p*—in both the univariate and multivariate cases, this can be done by enumerating and searching over the extreme points of a polytope determined by data-dependent constraints. However, there may be an enormous number of these extreme points (exponential or super-exponential in the number of data points $N$), which makes computing this value impractical. We conjecture that this difficulty may be unavoidable, because even for the simpler linear problem $\min_x \\| x \\|_p^p \  \textrm{s.t.} \ Ax = b$ with $0 < p \leq 1$, computation of restricted isometry and null space constants (which play a similar role in determining sufficient conditions for this problem to recover $\ell^0$ minimizers) is known to be NP hard in general [2]. Nonetheless, we argue that showing the existence of this $p^*$ is a significant theoretical contribution, as it rigorously confirms the natural intuition that taking $p$ close enough to 0 will eventually recover an exact $\ell^0$ solution. In practice, a simple strategy could be to retrain several times with increasingly small (e.g. halved) values of $p$ and stop once no further sparsity gains are observed. Developing more refined approaches for selecting $p$ is of interest for future work.
>
> [2] Tillman and Pfetsch. "The Computational Complexity of the Restricted Isometry Property, the Nullspace Property, and Related Concepts in Compressed Sensing." 2013.
>
>
>
> - **Experiments and practical performance**: We have since conducted numerical experiments (not included in our initial submission) which demonstrate that for high-dimensional data, a reweighted $\ell^1$ algorithm for minimizing our proposed $\ell^p$ objective produces solutions which are significantly sparser than those obtained with weight decay or unregularized training. In low dimensions, simply adding the $\ell^p$ regularizer to the Adam loss also produces solutions which are significantly sparser than both weight decay and unregularized training. We are unable attach these experiments here due to the new response format rules prohibiting PDFs and links: however, we can include them in the final paper. We also note that other strategies for $\ell^p$ regularization with $0 < p \leq  1$ in neural networks have also been proposed and shown to be empirically successful, including the weight factorization/reparameterization trick (see [3], Section 5.3 "Compression Benchmark"). Thus, although algorithmic implementation is not the focus of our paper, this type of regularization has shown good empirical performance, and may be even further improved with continued research into training approaches. Regarding how well these maximally-sparse solutions generalize to unseen test data as compared to other non-sparse solutions: although this is interesting, it is outside the scope of our focus here, which is on how to obtain these maximally sparse solutions if they are known a priori to be desirable for a particular problem.
>
> [3] Kolb et al. "Deep Weight Factorization: Sparse Learning through the Lens of Artificial Symmetries." 2025.

---

> > ### Comment · Reviewer_k9Kj · 2025-08-06
> >
> > Thank you for your detailed response. I have no further questions.

---

### Official Review · Reviewer_AdaJ · 2025-07-06

**Clarity:** 4
**Significance:** 3
**Originality:** 3
**Rating:** 5
**Confidence:** 3

**Summary:**

This paper demonstrates that the lp path norm (0 < p < 1) can induce the sparsest neural networks (l0 minimization). The uniqueness of the global minimizer (in the functional space) and the equivalence of lp to the l0 minimization problem when p is small enough are proved first in a one-hidden-layer and single-input-single-output ReLU network, and then generalized to a multi-input-single-output setting.

**Questions:**

Could you please provide some numerical examples for lp path norm minimization?

**Ethical Concerns:**

["NO or VERY MINOR ethics concerns only"]

**Final Justification:**

My questions about the limitations and implementations are well discussed in the rebuttal period. I believe the authors are willing to include this part in the next revision, which will benefit many readers.

**Limitations:**

The author has already mentioned some limitations in the conclusion part, including the two weaknesses I have raised.

**Quality:**

3

**Strengths And Weaknesses:**

Strength

- The mathematical analysis appears to be well done. However, I didn't verify all the details in practice.
- The theoretical implications are inspiring for designing neural networks.


Weakness

- Practitioners might look forward to more practical implications. For example, does the conclusion still hold for neural networks with ReLU networks with one hidden layer? At first glance, such neural networks also fall into the CPWL class, but I am unsure whether the technique can be generalized.
- The hardness in optimization might prevent the practical application. I think the lp path norm is another non-convex term, which introduces more concerns in optimization. Maybe some numerical examples could be there to demonstrate the practical values of the conclusions.
- Some references about empirical methods for training sparse neutral networks are missing. Specifically, I would like to mention the sparsity via the re-parametereization tricks [1] and its important follow up [2].

[1] https://arxiv.org/abs/2210.01212
[2] https://arxiv.org/abs/2307.03571

---

> ### Author Rebuttal · Authors · 2025-07-31
>
> We thank the reviewer for their careful evaluation of our work and for their feedback. Below we address the questions and concerns individually.
> - **Extension to deeper networks:** we agree that extending our results to deeper networks would be interesting and desirable. We conjecture that this is probably possible, but difficult, and would require different proof techniques than the ones used in our results. Specifically, the difficulties in extending our proofs to deep networks are the following:
>     - Univariate deep networks: like shallow, these are CPWL functions, but the relationship between the knots in the function and the neurons is more complex than in the shallow case. Specifically, modifying a single neuron of a shallow univariate network only produces a local change in the represented function, but for deep univariate networks, modifying neurons in deeper layers can produce global changes in the represented function. This makes our proof approach for the univariate shallow case difficult to apply to deep networks.
>     - Multivariate deep networks: although [1] develops a reformulation of three-layer ReLU network training as a linear constraint problem, similar to the two-layer formulation that inspires our proof approach, the resulting reformulated problem minimizes an $\ell^{2,p}$ group norm (rather than an $\ell^p$ norm) subject to a set of linear constraints. Unlike the $\ell^p$ norm, the $\ell^{2,p}$ group norm is neither convex nor concave, so our arguments here also do not readily extend to the deep case.
>
>     Thus, any extension to deeper networks would likely require different proof strategies than those used in our paper. Nonetheless, we believe that our results for the shallow case are valuable, as they provide (to our knowledge) the first sparse recovery guarantee for neural networks with arbitrary data, and develop innovative proof techniques for analyzing $\ell^p$ minimization in the highly nonlinear, nonconvex neural network context.
>
> [1] Ergen and Pilanci. "Global Optimality Beyond Two Layers: Training Deep ReLU Networks via Convex Programs." 2021.
>
> - **Hardness of optimization and numerical implementation/experiments:** we argue that in principle, non-convexity of the $\ell^p$ regularizer is not a major obstacle since the neural network training objective is already non-convex. In linear problems, effective algorithms for non-convex $\ell^p$ minimization with $0 < p \leq 1$ have been developed and are used frequently in practice (see e.g. [2]). We have since conducted numerical experiments (not included in our initial submission) which demonstrate that for high-dimensional data, a reweighted $\ell^1$ algorithm for minimizing our proposed $\ell^p$ objective produces solutions which are significantly sparser than those obtained with weight decay or unregularized training. In low dimensions, simply adding the $\ell^p$ regularizer to the Adam loss also produces solutions which are significantly sparser than both weight decay and unregularized training. We are unable attach these experiments here due to the new response format rules prohibiting PDFs and links: however, we can include them in the final paper. We also note that other strategies for $\ell^p$ regularization with $0 < p \leq  1$ in neural networks have also been proposed and shown to be empirically successful, including the weight factorization/reparameterization trick (see [3], Section 5.3 "Compression Benchmark"). Thus, although algorithmic implementation is not the focus of our paper, this type of regularization has shown good empirical performance, and may be even further improved with continued research into training approaches.
>
> [2] Lyu et al. "A comparison of typical $\ell^p$ minimization algorithms." 2013.
> [3] Kolb et al. "Deep Weight Factorization: Sparse Learning through the Lens of Artificial Symmetries." 2025.
>
> - **Additional references:** we thank the reviewer for the recommendation and will include the cited papers in our related work section.

---

> > ### Comment · Reviewer_AdaJ · 2025-08-09
> >
> > Dear authors,
> >
> > Thanks for your discussion. I fully understand the difficulty in the current stage of extending the theoretical study to a more general case. And I also appreciate your comment on the implementation side. I believe most of the readers may also benefit from this part.
> >
> > Glad to adjust my scores to acceptance.
> >
> > Best

---

### Decision · Program_Chairs · 2025-09-17

**Decision:**

Accept (poster)

**Comment:**

The main contribution of this paper is to show that minimizing $\ell_p$$ quasinorms of the weights of a 2-layer neural network (i.e. single-hidden-layer neural network) subject to an interpolation constraint corresponds to the sparsest possible interpolating network. The reviewers generally liked this insight and considered it to be a novel contribution to the mathematics of neural networks, despite some small limitations that were raised (e.g. limitation of the analysis to one hidden layer, the objective remaining nonconvex even when cast in continuous objective function form). The authors are highly recommended to include all of the detailed clarifications that they provided during the rebuttal phase in the camera-ready version.